# SBSC: Step-by-Step Coding for Improving Mathematical Olympiad Performance

**Kunal Singh, Ankan Biswas, Sayandeep Bhowmick, Pradeep Moturi, Siva Kishore Gollapalli**
Fractal AI Research
Mumbai, India
`{kunal.singh}@fractal.ai`

## Abstract

We propose Step-by-Step Coding (SBSC): a multi-turn math reasoning framework that enables Large Language Models (LLMs) to generate sequence of programs for solving Olympiad level math problems. At each step/turn, by leveraging the code execution outputs and programs of previous steps, the model generates the next sub-task and the corresponding program to solve it. This way, SBSC, sequentially navigates to reach the final answer. SBSC allows more granular, flexible and precise approach to problem-solving compared to existing methods. Extensive experiments highlight the effectiveness of SBSC in tackling competition and Olympiad-level math problems. For Claude-3.5-Sonnet, we observe SBSC (greedy decoding) surpasses existing state-of-the-art (SOTA) program generation based reasoning strategies by absolute 10.7% on AMC12, 8% on AIME and 12.6% on MathOdyssey. Given SBSC is multi-turn in nature, we also benchmark SBSC's greedy decoding against self-consistency decoding results of existing SOTA math reasoning strategies and observe performance gain by absolute 6.2% on AMC, 6.7% on AIME and 7.4% on MathOdyssey. Scripts & Data is uploaded at this link.

## 1 Introduction

Mathematical reasoning has emerged as a critical benchmark to measure the advanced reasoning and problem-solving abilities of the Large Language Models (LLMs) (Brown et al., 2020; Chowdhery et al., 2022; Achiam et al., 2023; Reid et al., 2024; Anthropic, 2023; OpenAI, June, 2024). This is due to the complex and creative nature of the numerous reasoning steps required to solve the problems.

Chain-of-Thought (Wei et al., 2022) and Scratchpad (Nye et al., 2021) prompting strategies helped LLMs to solve a problem using a step-by-step thought process. Program-Aided Language (PAL) (Gao et al., 2022) & Program-Of-Thought (POT) (Chen et al., 2022) introduced problem-solving via program generation where the answer is obtained by executing the generated program. Tool-Integrated Reasoning Agent (ToRA) (Gou et al., 2023) & Mathcoder (Wang et al., 2023a) introduced tool-integrated math problem solving format where model outputs natural language reasoning followed by program generation to solve the entire problem using a single code block and incorporates code-interpreter output for either summarizing the program output to get the final answer and terminate; or re-attempt the problem in the subsequent turn using the same format. For brevity, let's call ToRA's defined way of tool-integrated reasoning (TIR) strategy as TIR-ToRA.

The current generation of advanced LLMs such as GPT-4o (Achiam et al., 2023), Claude-3.5-Sonnet (Anthropic, 2023) and Gemini-ultra (Reid et al., 2024) have achieved high scores on elementary GSM8k (Cobbe et al., 2021) high-school level MATH (Hendrycks et al., 2021) by leveraging these reasoning strategies via in-context learning (Brown et al., 2020; Chowdhery et al., 2022). Multiple studies (Yu et al., 2023b; Yue et al., 2023; Toshniwal et al., 2024; Gou et al., 2023; Wang et al., 2023a; Mitra et al., 2024; Beeching et al., 2024; Shao et al., 2024) have tried supervised fine-tuning (SFT) approach to distill these reasoning formats using a propriety models like GPT4 (Achiam et al., 2023). These studies show significant performance improvement over GSM8K and MATH benchmarks.

## 1.1 MOTIVATION

However, recent math specific competition and Olympiad-level benchmarking on Math Odyssey (Fang et al., 2024), OlymiadBench (He et al., 2024), and the American Invitational Mathematics Examination (AIME) & the American Mathematics Competitions (AMC) (Beeching et al., 2024; DeepSeek-AI et al., 2024; Reid et al., 2024) questions show that the state-of-the-art (SOTA), both generalist and specialist, LLMs continue to struggle with advanced math reasoning. These results highlights the limitation of the existing math prompting techniques. (Tong et al., 2024) highlights the severe bias towards easy problems that exists in the SOTA SFT datasets which originates primarily due to the ineffectiveness of the current prompting strategies in complex math problem-solving. Often, multiple chains are generated via self-consistency decoding (Wang et al., 2022) and majority voting is done to boost the accuracy which is unlike how humans solve problems.

Fundamentally, both PAL & TIR-ToRA generate a single program block to solve the entire problem. Additionally, TIR-ToRA framework allows the model to re-attempt the program generation in case of execution error. These approaches show improved performance over COT on elementary & high school level math problems. However, solving olympiad-level math problem requires coming up with complex and creative solution that constitutes of numerous elaborate intermediate steps which eventually leads to the answer. Often, it is not feasible to solve a complex problem entirely using a single program block and as a result, these prompting strategies fail to systematically address each detailed step of the problem-solving process. It tends to overlook specified constraints, edge cases or necessary simplifications, which are often encountered in Olympiad-level problems.

## 1.2 OUR CONTRIBUTION

Olympiad level math problem-solving can be viewed as solving/exploring an intermediate sub-task/key-concept in depth; and discovering + solving the next critical sub-task dynamically basis the accumulated knowledge of previous sub-tasks/key-concepts explorations. To this end, we propose Step-by-Step Coding framework (SBSC) which is a multi-turn math reasoning framework that leverages existing programming (Naman Jain, 2024) and in-context learning skills (Brown et al., 2020) of the current generation of LLMs, particularly Claude-3.5-Sonnet (Anthropic, 2023) & GPT-4o (OpenAI, June, 2024). In each turn, it leverages code-interpreter results and knowledge of previous sub-tasks solutions or concept-explorations to define and programmatically solve the next sub-task. Thus it uses code generation as the reasoning strategy to solve an intermediate sub-task or explore an intermediate concept/step. Thus, providing detailed focus to each step of problem solving unlike PAL & TIR-ToRA. SBSC allows an intermediate key-step to be discovered, and be explored and refined (if needed) before being appended to the chain of steps whereas in PAL & TIR-ToRA all the intermediate steps are always stitched together.

We investigate the performance of SBSC on last 11 years of AIME & AMC-12 questions. We also benchmark on Olympiad-subset of MathOdyssey dataset along with math questions from OlympiadBench. We compare our method (greedy decoding) against greedy-decoding generation of existing reasoning strategies: COT, PAL & TIR-ToRA. We also show SBSC (greedy decoding) effectiveness by benchmarking against self-consistency decoding results of COT, PAL & TIR-ToRA. We conduct extensive ablations to understand the benefits of our approach such as sensitivity to exemplars, topic-wise analysis and measuring improvement in program refinement/debugging ability over TIR-ToRA due to the granular nature of SBSC process.

## 2 SBSC: STEP-BY-STEP CODING FRAMEWORK

Solving complex math problems, such as competition or Olympiad-level ones, involves creative thinking, applying diverse mathematical knowledge, and dynamically creating subsequent strategies as new insights emerge. One must discover sub-tasks dynamically, rigorously explore intermediate concepts, and carefully handle constraints and edge-cases. Since PAL & TIR-ToRA generates single code block (even during self-correction step; incase of TIR-ToRA) to solve a problem, they lack the flexibility or granularity to emulate this. To address this, we introduce SBSC.

SBSC is a multi-turn, code-generation based math reasoning prompting strategy where at each turn: the model generates an intermediate sub-task and corresponding program to solve that sub-task by leveraging the outputs of the previous turns. At the end of each turn, code interpreter is used to

execute the program block to generate the solution for the intermediate sub-task. The intermediate sub-task depends on the results of the previous turns and the question. The code snippet for the $i^{th}$ sub-task directly incorporates the execution results of the previous code snippets by directly defining them as variables and symbols. This way SBSC makes LLMs generate sequence of programs over multiple turns to solve complex math problems.

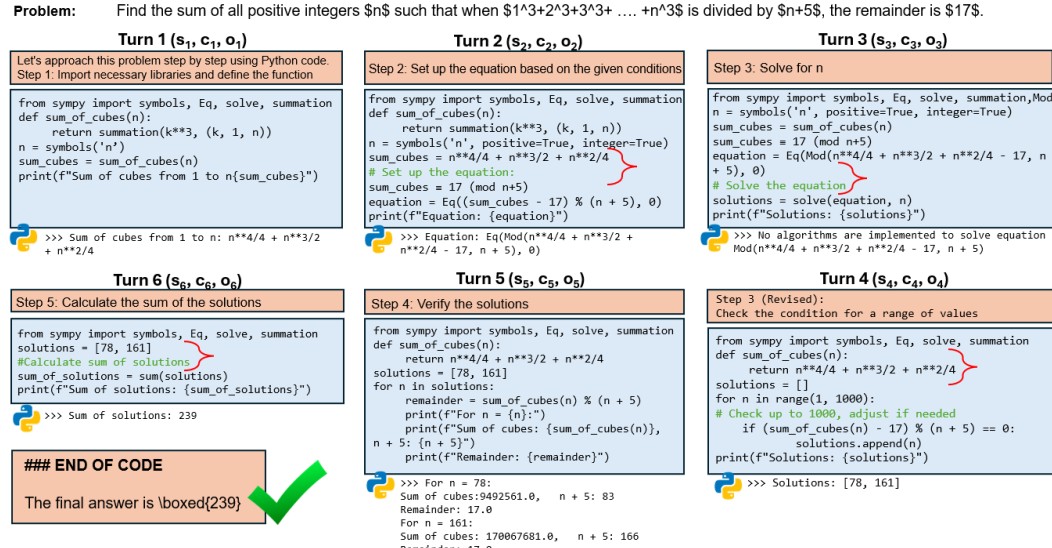

(a) Example multi-turn SBSC response for an AIME problem. Pink boxes denote the sub-task $s_i$ at the i-th step, blue boxes denote the program $c_i$ to solve $s_i$ and $>>>$ denote the corresponding execution output $o_i$. The red curly brackets indicate reusing outputs from earlier steps.

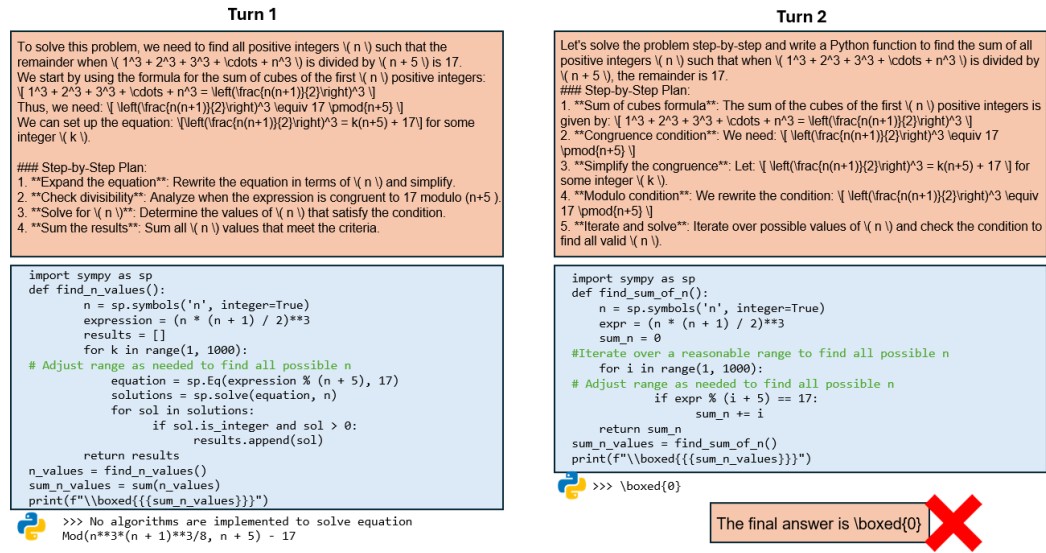

(b) Example TIR-ToRA response for the same problem, which is not solved correctly. In first turn, it tries to solves the problem at once using a rational and program. It encounters error and in second turn, tries to fix the entire approach and solve again but the solution is incorrect.

Figure 1: Comparison of SBSC and TIR-ToRA frameworks for same AIME problem

Our inference procedure is inspired by ToRA (Gou et al., 2023). Solution chain is initialized with the Prompt $p$ containing method instructions followed by exemplars and the current question $q$. At each step, LLM $G$ first outputs a subtask $s_i$. If $s_i$ generation ends with stop-word "###END OF CODE", we extract the final answer. Else, it continues to generate program code $c_i$ ending with

stop-word ""`output". We then pass $c_i$ to code interpreter and obtain the execution message or output $o_i \leftarrow E(c_i)$. The solution chain is updated by concatenating it with $s_i, c_i, o_i$ and loop continues till we get "###END OF CODE". For the $i^{th}$ turn and $\oplus$ denoting concatenation, the sequential process can be generalised as (except for the last turn where just the final answer is generated) :

$$s_i \oplus c_i \sim G(\cdot \mid p \oplus q \oplus (s_1 \oplus c_1 \oplus o_1) \oplus (s_2 \oplus c_2 \oplus o_2) \oplus ......(s_{i-1} \oplus c_{i-1} \oplus o_{i-1})) \quad (1)$$

Step-wise sequential approach of SBSC ensures that every part of the problem is addressed with exact precision, reducing the risk of errors that might arise from false assumptions or skipped steps. In case the code execution for any step results in an erroneous output, SBSC is better able to rectify that particular step. In depth understanding of SBSC (multiple examples & comparisons) at A.6.

We present example responses from both SBSC and TIR-ToRA for a problem from AIME in figures 1a and 1b respectively. As seen in case of TIR-ToRA, the initial program generated by the model runs into an execution error. At the next turn, it attempts to rectify the error and comes up with a new approach and the corresponding program. This time, the code executes correctly but due to reasoning error the final answer is wrong. On the other hand, we see that SBSC is progressing step-by-step, tackling individual sub-tasks with separate programs and utilising outputs of previous steps. In the third step, it runs into a code execution error but succeeds in rectifying it using a different approach in the very next turn. Further, we observe SBSC checking the validity of the generated solutions in the fourth step before proceeding with the final step and ultimately reaches the correct answer.

## 2.1 SBSC Exemplar Design

To enable SBSC framework in LLMs, we rely on in-context learning abilities (Brown et al., 2020) of LLMs as explored by multiple previous works such as (Chen et al., 2022; Gao et al., 2022; Gou et al., 2023) etc. We also use a system prompt similar to previous works. With respect to exemplar design, to enable program generation, we borrow learning from PAL (Gao et al., 2022) & POT (Chen et al., 2022) to have meaningful variable names in the code and using natural language comments within programs(Chen et al., 2022). To enable intermediate tool (code interpreter) usage, we leverage the use of stop words similar to in (Gou et al., 2023). Sample SBSC exemplars can be found at A.11, A.12.

## 3 Experiment

### 3.1 Benchmark datasets

We mainly use problems from 4 popular math competition datasets for benchmarking our performance: AIME, AMC, MathOdyssey (Fang et al., 2024) and OlympiadBench (He et al., 2024), covering multiple domains, mainly: Algebra, Combinatorics, Number Theory and Geometry. We use problems of last 11 years from AMC and AIME, obtaining questions and answers (Q&A) in LaTeX format from the AoPS Wiki website. MathOdyssey (Fang et al., 2024), a popular benchmark for LLM math reasoning, consists of problems of varying difficulties. We include the 148 problems belonging to olympiad-level competitions. OlympiadBench is another challenging benchmark for LLMs containing olympiad-level multilingual scientific problems. We select only math related questions, in english language.

### 3.1.1 Dataset Processing Details:

First, we filter out all questions having reference images associated. Second, we process the questions to have integer type answers if they are already not in that format. All AIME problems have a unique integer answer ranging from 0 to 999, while AMC-12 problems are of Multiple Choice Question(MCQ) format. Similar to NuminaMath (Beeching et al., 2024), we remove all the answer choices from each AMC-12 question and modify the representation of the final answer for the question, wherever necessary, to ensure an integer answer. In case of OlympiadBench and MathOdyssey, we simply modify the question as needed. For this, we prompt GPT-4o to append an additional line at the end of each problem as suitable. Following is an example for demonstration:

**Original Question:** An urn contains one red ball and one blue ball. A box of extra red and blue balls lies nearby. George performs the following operation four times: he draws a ball from the urn at random and then takes a ball of the same color from the box and returns those two matching balls to the urn. After the four iterations the urn contains six balls. What is the probability that the urn contains three balls of each color?

**Answer**: $\frac{1}{5}$

**Modified Question:** An urn contains one red ball and one blue ball. A box of extra red and blue balls lies nearby. George performs the following operation four times: he draws a ball from the urn at random and then takes a ball of the same color from the box and returns those two matching balls to the urn. After the four iterations the urn contains six balls. What is the probability that the urn contains three balls of each color? If the answer is represented as a fraction $\frac{m}{n}$ in its simplest terms, what is the value of m+n?

**Integer Answer**: 6

Final test set, contains 330 AIME, 475 AMC-12, 158 MathOdyssey & 504 OlympiadBench problems.

## 3.2 BASELINE & CONFIGURATIONS

We benchmark against three prompting/reasoning strategies: COT (Wei et al., 2022), PAL (Gao et al., 2022) TIR-ToRA (Gou et al., 2023). We use `gpt-4o-2024-05-13` and `Claude-3.5-Sonnet` as base LLMs for our experiments. For all datasets and all reasoning frameworks, we use 4-shot setting. Maximum number of turns (n) SBSC is set to 15. For greedy decoding inference, we use `temperature=0 and max_tokens=1024` and also, we run 3 times and report average. For greedy decoding of TIR-ToRA, we keep `n = 15` as well (Note: this is because although in TIR-ToRA strategy the model attempts to solve the entire problem in the single turn, in case of execution error or readjustment it tries to re-attempt in subsequent turns). We also benchmark SBSC's greedy decoding results against self-consistency (SC) (Wang et al., 2022) decoding results (majority@7) of COT, PAL & TIR-TORA. We do this primarily for two reasons: First, SBSC takes multiple turns before arriving at the final answer (on average 6-7 turns per problem , Table 3 in Appendix A.1) and Secondly, to benchmark against the reliance of the current existing prompting strategies on majority voting for boosting accuracy. For SC decoding, we use `temperature=0.7 and top_p=0.9`. Note: we experimentally observe that for n > 4, there is insignificant increase in accuracy for TIR-ToRA so we set `n=4` for TIR-ToRA during SC decoding.

Note: PAL (Gao et al., 2022) work also reports a combined approach with Least-to-Most (L2M) prompting strategy (Wang et al., 2022), L2M-PAL that is essentially two stage. We implemented it as per the reported examples in the PAL work. We benchmark it on AMC + AIME dataset. We observe that L2M-PAL at best matches PAL or TIR-ToRA scores. Detailed results available in appendix A.14. Hence for our main results, we stick to PAL & TIR-ToRA along with self-consistency decoding due to resource optimisation and wider adaption of those prompting strategies for math-problem solving. For more discussion on L2M-PAL please check A.14.

## 3.3 PROMPTING/FEW-SHOT EXEMPLARS

For both AIME and AMC, we select 90 questions each, drawn from problems of years other than those included in the evaluation datasets. These questions were prompted with COT, PAL, TIR-ToRA and SBSC to generate corresponding solutions in accurate format. For each dataset, we create a subset of 10 problems correctly solved by every method and finally select a combination of 4 exemplars among them. For MathOdyssey as well as Olympiad Bench, we use AIME exemplars as these datasets are of similar difficulty level. We provide the 4 chosen exemplars and system-prompts, used in the main experiments, for different methods in Appendix (A.9, A.10, A.11, A.12) & repository here.

## 4 RESULTS

We report the percentage accuracy of all the methods with different base LLMs and across all the benchmarking datasets in Table 1. On AMC dataset, SBSC shows an absolute improvement over TIR-ToRA (greedy decoding) by roughly 11% using Claude-3.5-Sonnet and 7% using GPT-4o. SBSC

Table 1: Benchmarking SBSC against different math reasoning methods across 3 datasets:We report the average accuracy( in percentage unit) over 3 runs. Best result in each setting is highlighted in **bold** & second best is underlined. Absolute improvement in performance by SBSC over the previous best method in each setting is indicated in subscript.

| Method | AMC | | AIME | | MathOdyssey | | Olympiad Bench | |
|---|---|---|---|---|---|---|---|---|
| | greedy | maj@7 | greedy | maj@7 | greedy | maj@7 | greedy | maj@7 |
| Claude-3.5-Sonnet | | | | | | | | |
| COT | 31.16 | 35.79 | 9.09 | 10.91 | 11.89 | 16.89 | 39.35 | 42.46 |
| PAL | 35.79 | 36.42 | 27.48 | 28.79 | 27.23 | 31.01 | 41.07 | 44.44 |
| TIR-ToRA | 38.59 | 43.16 | 24.64 | 26.67 | 27.23 | 32.43 | 47.69 | 50.60 |
| SBSC (Ours) | **49.33**$_{\uparrow 10.7}$ | $-_{\uparrow 6.2}$ | **35.45**$_{\uparrow 8}$ | $-_{\uparrow 6.7}$ | **39.86**$_{\uparrow 12.6}$ | $-_{\uparrow 7.4}$ | **53.31**$_{\uparrow 5.6}$ | $-_{\uparrow 2.7}$ |
| GPT-4o | | | | | | | | |
| COT | 35.94 | 37.47 | 10.39 | 12.12 | 13.51 | 17.57 | 41.80 | 47.22 |
| PAL | 36.48 | 38.11 | 24.63 | 26.97 | 15.74 | 20.27 | 41.67 | 46.43 |
| TIR-ToRA | 37.33 | 40.42 | 22.42 | 25.45 | 19.59 | 23.64 | 43.32 | **49.61** |
| SBSC (Ours) | **44.55**$_{\uparrow 7.2}$ | $-_{\uparrow 4.1}$ | **30.7**$_{\uparrow 6.1}$ | $-_{\uparrow 3.7}$ | **26.55**$_{\uparrow 7}$ | $-_{\uparrow 2.9}$ | **48.74**$_{\uparrow 5.4}$ | $-_{\downarrow 0.87}$ |

greedy decoding results outperforms SC decoding results of TIR-TORA by absolute 6% and 4%, for Claude-3.5-Sonnet and GPT-4o respectively. We see similar absolute improvements in accuracy on our AIME dataset too. SBSC outperforms its nearest competitor (PAL) by 8% and 6% with greedy settings and SC settings by 6.7% and 3.7%, for Claude-3.5-Sonnet and GPT-4o respectively. For MathOdyssey, SBSC improves by as much as 12.6% and 7% over TIR-ToRA while showing improvement of 7.4% and 3% over its SC variant, for Claude-3.5-Sonnet & GPT-4o respectively. On OlympiadBench, for GPT-4o, SBSC matches SC results of TIR-ToRA and is better than the second best greedy variant by more than 5%. While for Claude-3.5-Sonnet, SBSC shows an absolute improvement of nearly 6% and 3% over TIR-ToRA in greedy and SC setting respectively. Standard deviation values at A.13. We conduct additional studies, presented in appendix A.2, that show SBSC surpasses previous methods on two more additional challenging datasets JEE-Bench (Arora et al., 2023) and OmniMATH (Gao et al., 2024). We also show in A.3 that SBSC is effective for open source model as well.

## 5 ABLATIONS & ANALYSIS

### 5.1 SENSITIVITY TO EXEMPLARS

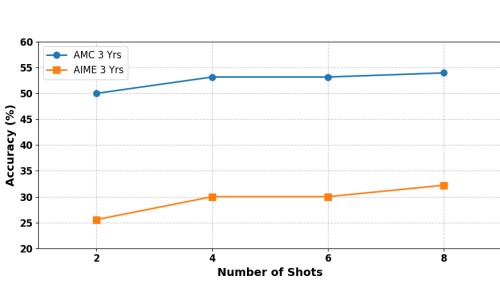

Figure 2: Effect of Number of Exemplars

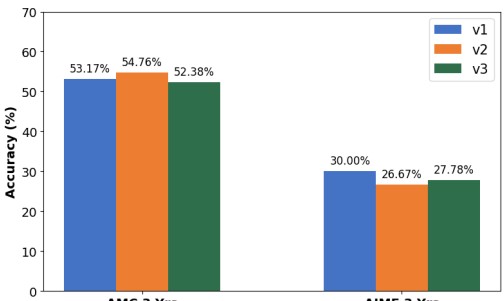

Figure 3: Sensitivity to choice of Exemplars

We study the effect of number/choice of examples in prompting on SBSC's performance using Claude-3.5-Sonnet on a subset of AIME and AMC data. As shown in Figure 2, we observe a notable increase in performance when increasing the examples from 2 to 4, which then starts to saturate as we further increase the number of examples to 6 and 8. This justifies our decision of using a 4-shot setting. To understand if the choice of exemplars affect the accuracy or not, we conduct a sensitivity analysis. We randomly sample 4 exemplars out of the already created pool of 10 exemplars three

times to create 3 variations of 4-shot prompts: v1, v2, and v3. In Figure 3, we can see that the performance remains stable irrespective of the exemplars used.

## 5.2 SBSC EXEMPLAR TUNING

Natural language comments present within a program have proven to be useful (Gao et al., 2022). So, in each of the SBSC exemplars, we provide suitable comments in natural language within the Python program for each turn to help guide the model.

Table 2: SBSC performance comparison across prompt variations using Claude-3.5-Sonnet

|            | Full Prompt | Without Comments | Without Line 1 |
|------------|-------------|------------------|----------------|
| AMC 3 Yrs  | 67          | 62               | 60             |
| AIME 3 Yrs | 27          | 19               | 16             |

For few-shot learning, apart from relevant exemplars, the LLM also benefits from a general instruction at the beginning (Zheng et al., 2024; Gou et al., 2023; Wang et al., 2023a) that provides a guideline or context about how the model should approach the task, particularly those requiring logical reasoning, multi-step operations, etc. This can be specially useful when the task requires a more nuanced understanding and when the instructions need to be followed rigorously, as is the case with SBSC. Kindly refer to A.11 and A.12 for detailed prompts.

In particular, we highlight one line from the instructions part of the prompt wherein, the model is specifically being instructed to invoke a code rectification step to ensure that the error is not propagated further, leading to a wrong answer. It also ensures the model focuses only on the intermediate step. :
**If the executed code snippet returns an error, use it to correct the current step's code snippet. DO NOT restart solving from Step 1.** 1

In Table 2, we study the importance of these two components in particular: the comments within the code snippets and line 1 mentioned above. Our findings suggest that removal of either of these components lead to a significant decrease in the performance, indicating how each of them are crucial aspects of our exemplar prompts.

## 5.3 CODE DEBUGGING ABILITY

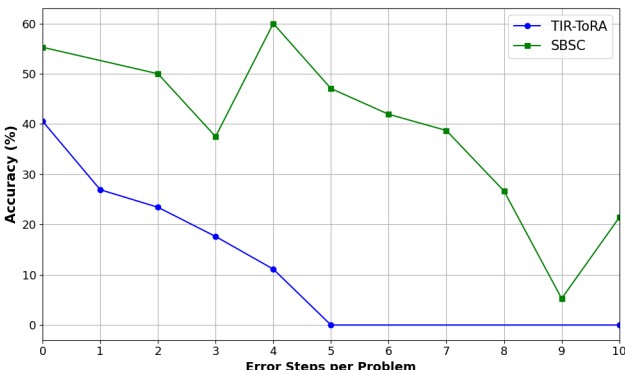

Figure 4: Comparison of Debugging Abilities

We present the superior ability of our method to resolve an error related to code execution. If at any step of the trajectory chain, the program returns an execution error, we consider that to be an error step. We visually represent this, using Claude-3.5-Sonnet responses across AMC, AIME and MathOdyssey datasets in Figure 4, where we see that SBSC is able to recover from even multiple wrong steps and reach the correct final answer quite easily when compared to TIR-ToRA whose performance drops steeply on increasing error steps. This can be attributed to the fact that SBSC, being precise and granular, tackles only a focused part of the problem and finds it easier to correct its mistakes compared to TIR-ToRA which tries to correct the program at the problem level.

## 5.4 TOPIC-WISE ANALYSIS

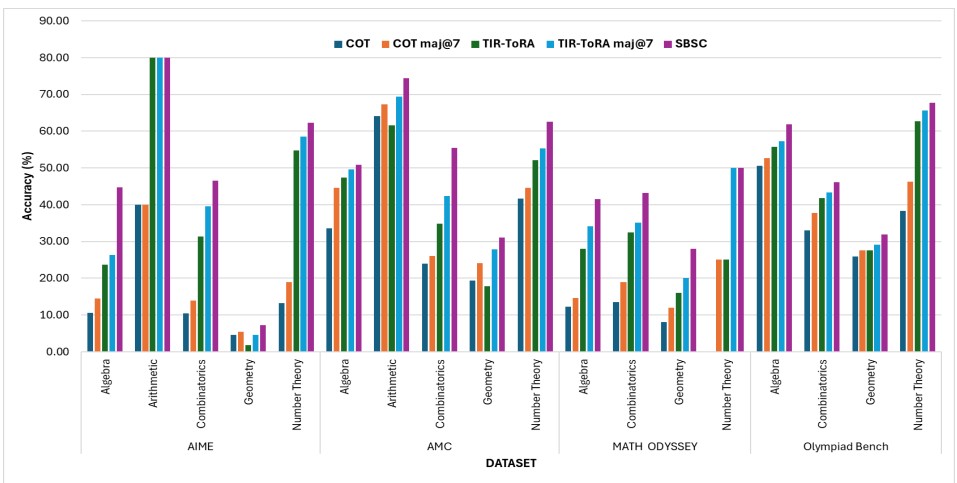

Figure 5: Topic breakdown analysis

We use GPT-4o-mini (OpenAI, June, 2024) to classify problems from AIME and AMC, while MathOdyssey and OlympiadBench already contained topic labels. Our test set primarily comprised of: Algebra, Arithmetic, Combinatorics, Number Theory and Geometry. In this study, we benchmark the solutions obtained using Claude-3.5-Sonnet. As can be seen in Figure 5, our method outperforms COT & TIR-ToRA (against both greedy and self-consistency decoding) in all the individual topics and across all the 4 datasets, thereby proving beneficial for all topics. This highlights the generalisation ability of our approach extending to different types and complexities of problems.

## 5.5 SBSC ACCURACY CORRELATION WITH CODING CAPABILITIES OF LLMS

We study the correlation of code related capabilities of the LLMs with respect to their success with SBSC. Since coding capabilities of a model is pivotal towards successfully following and executing our SBSC approach, we make a comparison involving LLMs with varying coding abilities. Figure 6 shows that the SBSC scores are correlated to the code generation abilities of the corresponding models for all cases that were evaluated on a subset of AIME and AMC data. The code-generation scores were taken from LiveCodeBench (Naman Jain, 2024) benchmark.

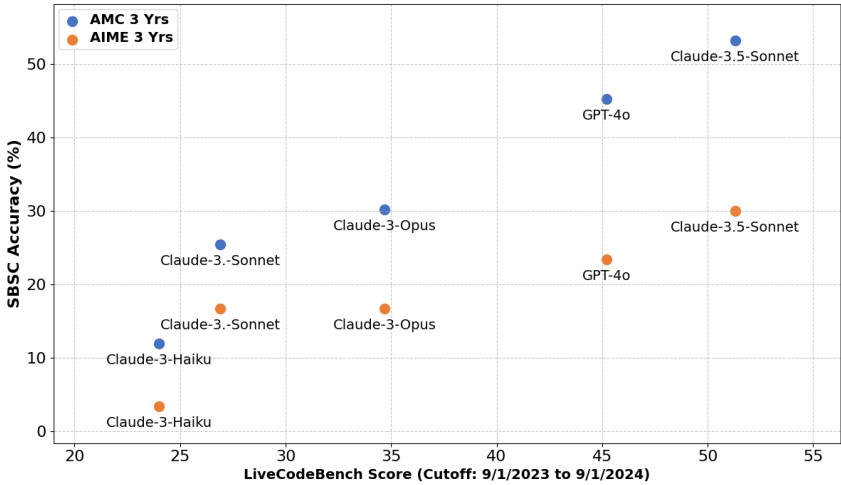

Figure 6: SBSC accuracy correlation with coding ability of LLMs

## 5.6 SBSC + SELF-CONSISTENCY

Self-consistency (SC) decoding (Wang et al., 2022) has proven to be effective in boosting accuracy via sampling multiple chains and taking a majority voting. We employ SC decoding to assess the upper bound of our approach. For this study, we use `temperature=0.7` and `top_p=0.7`.

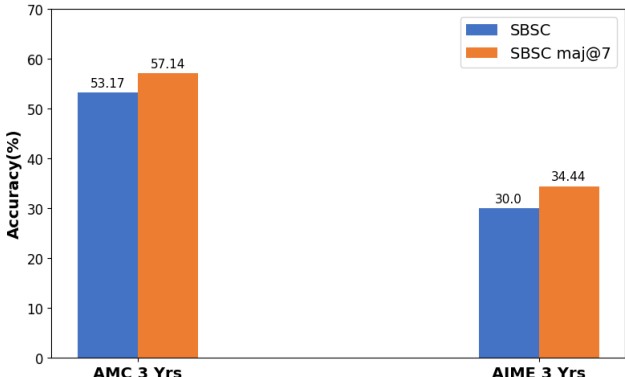

Figure 7: SBSC scores with Self-Consistency (maj@7)

We generate 7 chains using Claude-3.5-Sonnet for each problem of last 3 years of AMC and AIME; and consider the majority voted answer as the prediction to be compared against the ground truth. We notice from Figure 7 that the maj@7 accuracy is higher than that of greedy decoding, following the usual trend with other prompting approaches like COT, PAL, etc.

More Ablations in Appendix:

- We present average cost per question analysis in A.4 and show that our main results Table 1 already accounts of token normalized comparison.

- We additionally show, in A.5, that exemplar preparation for SBSC, like previous methods, did not require any manual effort apart from format inspections. We also create a larger pool of AIME exemplars show the stability and generalisability of SBSC by sampling 9 sets of 4 exemplars and reporting accuracy on last 3 years AIME and MathOdyssey questions in A.5.

- We also present a study on sympy usage by SBSC and TIR-ToRA in A.7. We show that sympy usage is consistent in both the methods, however SBSC achieves higher accuracy if we compare the accuracy among those questions.

## 6 RELATED WORK

Recently, significant advancements across various research directions have been made to enhance the mathematical capabilities of large language models (LLMs). One of the major ones has been along the prompting and thinking strategies such as Chain-of-Thought (COT) method (Wei et al., 2022; Kojima et al., 2022) that has shown to evoke multi-step thinking in LLMs before arriving at the answer. These methods struggle with complex and symbolic computations. For this, PAL (Gao et al., 2022) & POT (Chen et al., 2022) suggest making LLMs perform reasoning by writing program and offloading the computations to code interpreter. TIR-ToRA (Gou et al., 2023) does rationale generation in one go beforehand and then it codes the entire solution to get the final answer using single code block. Additionally, in case of an error, it tries to re-attempt in similar format. Another line of research has been around pre-training and supervised fine-tuning (SFT). Multiple studies (Shao et al., 2024; Ying et al., 2024; DeepSeek-AI et al., 2024; Azerbayev et al., 2023; Lewkowycz et al., 2022; Paster et al., 2023; Taylor et al., 2022) have shown pre-training LLMs on high-quality maths tokens results in increased mathematical knowledge and reasoning abilities. Recent approaches (Yu et al., 2023b; Gou et al., 2023; Yue et al., 2023; Wang et al., 2023a; Shao et al., 2024; Toshniwal et al., 2024; Mitra et al., 2024; Beeching et al., 2024; Yin et al., 2024; Tong et al., 2024) have tried query/problem augmentation along with creating synthetic reasoning paths/trajectories using a teacher

model like GPT4 (Achiam et al., 2023) for SFT. These methods showed significant improvement in the math reasoning abilities of the model. Also, some studies (Wang et al., 2023b; Yu et al., 2023a; Xi et al., 2024; Chen et al., 2024; Lightman et al., 2023b) provide an alternative to manual annotations for process supervision (Lightman et al., 2023a).

Previous program generation-based methods such as PAL, & TIR-ToRA have followed a static approach and hence struggle with complex problems. These methods typically attempt to solve the problem in a single, large code block or with minimal iterations(in case of TIR-ToRA). But they lack producing/exploring any intermediate steps. SBSC dynamically decomposes complex problems into a sequence of manageable sub-tasks. SBSC embodies exploration by discovering new sub-tasks basis the previous sub-tasks. SBSC is highly focused as these sub-tasks are explored individually via program generation to generate intermediate outputs. SBSC has superior step-level self-correction abilities. Hence due to these major advantages, SBSC achieves significant improvement over TIR-ToRA & PAL across 4 different benchmarks as can be seen in Table 1.

## 7  CONCLUSION

We introduce SBSC, a multi-turn math reasoning framework that tries to enable LLMs to solve complex math problems. SBSC pursues the solution, step-by-step with each turn dedicated to a step, and arrives at final answer via multiple turns. At each turn, an intermediate sub-task and its corresponding program solution is generated leveraging the execution outputs and solutions of all the previous sub-tasks. We show performance improvements of SBSC over TIR-ToRA, PAL & COT on challenging math problems. We also show that greedy-decoding results of SBSC outperforms self-consistency results of other prompting strategies.

## 8  FUTURE WORK

Given the detailed, dynamic and flexible step-wise nature of problem-solving along with the fact that its leverage program generation to conclude a key-intermediate step, we believe SBSC reasoning format could be highly useful for guided decoding strategies such as in Outcome-Supervised Value Model (Yu et al., 2023a), AlphaMATH (Chen et al., 2024), Q* framework (Wang et al., 2024). It would be well suited for step-wise preference optimisation for reasoning such as in (Lai et al., 2024). SBSC trajectories could be used also for imitation learning via SFT.

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

## A APPENDIX

### A.1 NUMBER OF STEPS IN SBSC

In Table 3, we present the number of turns taken per question by SBSC responses obtained using Claude-3.5-Sonnet across the different datasets.

Table 3: Table showing number of turns/steps used by SBSC

| Number of turns or steps | AMC | AIME | MathOdyssey |
|---|---|---|---|
| 2 | 21 | 12 | 8 |
| 3 | 57 | 19 | 17 |
| 4 | 101 | 47 | 19 |
| 5 | 79 | 51 | 21 |
| 6 | 63 | 43 | 28 |
| 7 | 41 | 43 | 14 |
| 8 | 42 | 31 | 10 |
| 9 | 12 | 18 | 8 |
| others | 59 | 66 | 23 |
| **Average turns or steps/Problem** | 6.0 | 6.9 | 6.4 |

### A.2 RESULTS ON JEE-BENCH AND OMNIMATH DATASETS

We present results on additional two benchmarks: JEE-Bench (Arora et al., 2023) and Omni-MATH (Gao et al., 2024). For JEE-Bench, we evaluate on 98 numerical answer type questions. For Omni-MATH, we filter out 576 questions out of that with all 26 Qs from calculus topics and 110 Qs randomly sampled from each one of the remaining topics. We use AIME exemplars for evaluation on these two datasets to also further show how generalizable SBSC is.

For JEE-Bench in Table 4, we observe significant improvements compared to other methods. For Claude 3.5 Sonnet, SBSC's greedy decoding takes an absolute lead (against greedy decoding) of 16% over TIR-ToRA, 20% over PAL and over 30% COT. Similarly benchmarking against self-consistency decoding (majority@7) of other methods, SBSC takes an absolute lead of 11% over TIR-ToRA, 16% over PAL and over 25% COT.

Table 4: Benchmarking accuracy on JEE-Bench dataset.

| Model | COT | | PAL | | TIR-ToRA | | SBSC | |
|---|---|---|---|---|---|---|---|---|
| | greedy | maj@7 | greedy | maj@7 | greedy | maj@7 | greedy | maj@7 |
| Claude 3.5 | | | | | | | | |
| Sonnet | 32.65 | 37.76 | 42.86 | 46.94 | 45.92 | 51.02 | **62.24**$_{\uparrow+16.32}$ | -$_{\uparrow+11.22}$ |
| GPT-4o | 29.59 | 33.67 | 35.71 | 38.78 | 32.65 | 37.76 | **50.00**$_{\uparrow+14.29}$ | -$_{\uparrow+11.22}$ |

On Omni-MATH dataset as seen in Table 5, SBSC(greedy decoding) shows an absolute improvement over TIR-ToRA (greedy decoding) by roughly 10% using Claude-3.5-Sonnet and 6% using GPT-4o. SBSC greedy decoding results outperforms self-consistency decoding results of TIR-TORA by absolute 6% and 4%, for Claude-3.5-Sonnet and GPT-4o respectively.

Table 5: Benchmarking accuracy on OmniMATH dataset.

| Model | COT | | PAL | | TIR-ToRA | | SBSC | |
|---|---|---|---|---|---|---|---|---|
| | greedy | maj@7 | greedy | maj@7 | greedy | maj@7 | greedy | maj@7 |
| Claude 3.5 | | | | | | | | |
| Sonnet | 23.09 | 26.22 | 32.81 | 35.94 | 33.68 | 37.15 | **43.06**$_{\uparrow+9.38}$ | -$_{\uparrow+5.91}$ |
| GPT-4o | 27.78 | 29.86 | 34.38 | 37.15 | 33.85 | 36.28 | 40.45$_{\uparrow+6.07}$ | $_{\uparrow+3.40}$ |

## A.3 BENCHMARKING OPEN SOURCE MODEL: DEEPSEEKCODER V2.5

We performed benchmarking with DeepSeekCoder 2.5 (DeepSeek-AI, 2024) which is an open source model. Note: given the resource constraint and that most of the frontier open source LLMs such as DeepSeekCoder 2.5 require multiple gpus and days to host and run the experiments respectively, we leveraged this open source model via api provided by deepseek which was highly cost-effective. We present the results on 2 test-sets: AIME and AMC as used in the main results.

We can clearly observe in Table 6 that SBSC outperforms both greedy decoding and self-consistency decoding results of TIR-ToRA, PAL & COT on both last 11 years AIME & AMC.

Table 6: DeepSeekCoder 2.5 accuracy for SBSC, TIR-ToRA, PAL & COT on last 11 years AIME & AMC test-sets.

| Model | COT | | PAL | | TIR-ToRA | | SBSC | |
|---|---|---|---|---|---|---|---|---|
| | greedy | maj@7 | greedy | maj@7 | greedy | maj@7 | greedy | maj@7 |
| AIME | 8.48 | 10.30 | 18.79 | 20.00 | 19.70 | 20.61 | **23.64**$_{\uparrow+3.94}$ | -$_{\uparrow+3.03}$ |
| AMC | 24.84 | 26.53 | 29.47 | 30.53 | 31.16 | 32.42 | **34.53**$_{\uparrow+3.37}$ | -$_{\uparrow+2.11}$ |

## A.4 COST AND TOKEN COMPARISON ACROSS SBSC, TIR-ToRA, PAL & COT

We also have done a study with Claude 3.5 Sonnet to compare the average cost per question. We have taken average over 50 AIME+AMC questions. We leveraged the prompt-caching of repeated input tokens while doing this experiment. Specifically, we cache the system prompt and exemplars for all the methods. The results are presented in the table below. As shown in the comparison in Table 7 SBSC is 3X of TIR-ToRA, 4X of COT and 7X of PAL. When comparing average cost per question even in non-caching mode, from Table 8 we observe that SBSC is 6.2X of TIR-ToRA, 7X of COT and 14.3x of PAL. Hence, Table 1 already covers fair comparison part as well even if we take

average cost per question under consideration. In Table 1, we already compared 1 chain of SBSC with 7 chains of other methods. Just as experimental detail, we would like to mention we leveraged prompt caching for our reported experiments with Claude 3.5 Sonnet.

Table 7: Average Cost per Question for different methods using Claude with Prompt Caching.

| Claude with prompt caching | COT | PAL | TIR-ToRA* | SBSC* |
|---|---|---|---|---|
| Input tokens | 110.75 | 107.75 | 479.25 | 4788.90 |
| Output tokens | 543.95 | 353.20 | 685.15 | 1070.00 |
| Cache tokens | 2754.00 | 1255.00 | 2692.20 | 23610.60 |
| Cost/Question ($) | 0.0093 | 0.0060 | 0.0125 | 0.0375 |
| Cost ratio (SBSC/Methods) | 4.00 | 6.25 | 3.00 | 1.00 |

*: SBSC & TIR-ToRA values represent the sum of all turns per question

Table 8: Average Cost per Question for different methods Using Claude without Prompt Caching.

| Claude without prompt caching | COT | PAL | TIR-ToRA* | SBSC* |
|---|---|---|---|---|
| Input tokens | 2858.75 | 1356.75 | 3565.45 | 32846.70 |
| Output tokens | 536.05 | 267.35 | 543.70 | 1161.50 |
| Cost/Question ($) | 0.0166 | 0.0081 | 0.0189 | 0.1160 |
| Cost ratio (SBSC/Methods) | 7.00 | 14.30 | 6.20 | 1.00 |

*: SBSC & TIR-ToRA values represent the sum of all turns per question

In Table 9, we present the average token count per question for exemplars and system prompt across all the methods. We would like to highlight that due to prompt caching mode available in LLMs, the entire prompt is charged only once.

Table 9: Token Usage Comparison Across Different Methods (Note: All values are in tokens)

| Dataset | Method | COT | PAL | TIR-ToRA | SBSC |
|---|---|---|---|---|---|
| AIME | system_prompt | 11 | 42 | 123 | 208 |
| | exemplars | 2347 | 1049 | 1588 | 4012 |
| AMC | system_prompt | 11 | 42 | 123 | 208 |
| | exemplars | 1557 | 1013 | 1554 | 3486 |

In case of SBSC, the last turn requires the largest context length as last turn(n) input consists of = system prompt + exemplars + question + output trajectory from 1 to n-1. We present the study in Table 10.

Table 10: Token Analysis of System Prompts and Trajectories (Note: All values are in tokens)

| Dataset | System Prompt + Exemplars | Trajectory 1 to n-1 | Total |
|---|---|---|---|
| AIME | 4,220.00 | 899.20 | 5,119.20 |
| AMC | 3,694.00 | 835.85 | 4,529.85 |

A.5 Exemplar sensitivity analysis with larger pool

A.5.1 Brief on Exemplar Curation & Crafting and Why it didn't require manual effort

To enable the multi-turn reasoning for SBSC, we use combination of system prompt and few-shot prompting (exemplars) similar to what POT, PAL, TIR-ToRA used. Similar to past works, our exemplar generation process didn't require any manual effort other than writing the system prompt and manually inspecting the final pool for format check.

To prepare the exemplars,

- We first used 0-shot prompting (system prompt) to generate responses for questions in the SBSC format. As mentioned in 3.3 "Prompting/Few-Shot Exemplars", for both AIME and AMC, We select 90 questions each, drawn from problems of years other than those included in the evaluation datasets.

- We observe that out of 90, we only had 15 questions that were solved across the methods (COT, PAL, TIR-ToRA, SBSC). This churn was particularly due to the low accuracy of COT method.

- A pool of 10 common problems and corresponding exemplars were finally selected. In this step, we ensured the diversity in the problem set (we used GPT-4o to get the question categories) and ensured that SBSC format was accurately followed in them. The distribution was: 3 Algebra, 3 Number Theory, 2 Geometry, 2 Combinatorics.

- However, the selection of 4 exemplars from the 10 is completely random.

- Additionally, as mentioned in section 2.1, we also ensure certain practices (mainly 2) for program generations in the exemplars which are inspired from PAL paper. First is to have meaningful and coherent variable names and second is to having natural language intermediate comments within the program. We observed that the base LLM followed the first point (meaningful variable names) by default. For the second point, which is the natural language comments within the code, we used GPT-4o to add comments within the generated programs for the final 10 curated exemplars. PAL paper already established (and used) these two things improve the performance. We even measured the impact of the comment based exemplar tuning in section 5.2 and Table 2.

- The only thing manual in this entire exemplar curation process was writing the system prompt and final inspection if SBSC format was correctly followed or not.

A.5.2 Additional Exemplar Sensitivity & Generalization Analysis with bigger pool

We conduct an additional study to expand the pool of exemplars for sensitivity analysis as suggested by the reviewer. We expand the pool to 30 AIME exemplars (prepared in the same manner as made initial 10 exemplars as mentioned above and in section 3.3 "Prompting/Few-Shot Exemplars") that were solved where we have 8 Algebra, 8 Number Theory, 8 Combinatorics and 6 Geometry questions. We randomly sample 4 exemplars from these 30 exemplars 9 times to create 9 sets of 4-shot prompts: v1, v2, v3, v4, v5, v6, v7, v8, v9. We test them on the same set of last 3 years AIME questions.

For further testament to our generalisation, we use AIME exemplars on MathOdyssey test-set.

We can clearly observe in Table 11 that the performance on AIME remains stable, as experienced in previous study in figure 3, with 9 exemplar-set randomly sampled from 30 AIME Qs. Also, from Table 12 we can see the performance on MathOdyssey stays stable with even 9 AIME exemplar sets.

We have utilized AIME exemplars for benchmarking SBSC on MathOdyssey 1, OlympiadBench 1, JEE-Bench 4 and Omni-MATH 5. We can observe on all these datasets, SBSC performances much better than other SOTA methods. This proves the generalization of SBSC and also proves that there has been no over fitting of exemplars done.

Table 11: Last 3 Yrs AIME Accuracy for Different Exemplar Combinations

| Exemplar Variations | Last 3 Yrs AIME Accuracy (%) |
|---|---|
| v1 | 31.11 |
| v2 | 32.22 |
| v3 | 32.22 |
| v4 | 33.33 |
| v5 | 32.22 |
| v6 | 30.00 |
| v7 | 28.88 |
| v8 | 33.33 |
| v9 | 28.88 |

Table 12: MathOdyssey Accuracy for Different Exemplar Combinations

| Exemplar Variations | MathOdyssey Accuracy (%) |
|---|---|
| v1 | 39.86 |
| v2 | 38.51 |
| v3 | 37.17 |
| v4 | 38.51 |
| v5 | 41.89 |
| v6 | 37.84 |
| v7 | 39.86 |
| v8 | 39.86 |
| v9 | 37.84 |

## A.6  UNDERSTANDING SBSC IN DETAIL

In this section, we demonstrate some scenarios where SBSC has been successful while TIR-ToRA has failed, with the help of some example questions and investigating the responses obtained from the two models.

Let's consider the question in Example 1, involving a geometric progression of numbers written in logarithmic form, which TIR-ToRA gets wrong.The method uses a binary search technique, which is not very precise when dealing with exact values required for mathematical problems, especially when fractions are involved.The solution uses a function to check whether the logarithms form a geometric progression which introduces additional complexity and potential inaccuracies because it involves comparing ratios that may not be exactly equal due to floating-point arithmetic.Also, this single-turn method tends to overlook specified constraints or necessary simplifications, which are often encountered in Olympiad level problems and instead makes false assumptions.

The question in Example 2 is an example scenario where TIR-ToRA fails because it makes an incorrect assumption. It misinterprets the Lipschitz condition and incorrectly makes a simpler assumption that the difference $f(800) - f(400)$ is equal to the maximum possible difference, which is 200. While the magnitude of the difference is bounded by 200, it does not mean that the actual difference will always be 200. Iterative solutions, as are often the only way out in single program based solutions, can sometimes lead to infinite loops, especially in cases where the stopping condition is not clearly defined or understood by the LLM.

As can be seen in Example 3, the single code is unable to take advantage of the factorization of $20^{20}$,

which is key to solving the problem efficiently and instead iterates over a very large range of potential values for $m$, leading to inefficiency. The upper bound 2020 is extremely large and the sheer number of iterations causes a timeout.

Example 4 presents a scenario where TIR-ToRA makes up an assumption about the problem and writes the code for terminating a loop accordingly, which leads to a timeout error, as the incorrect assumption leads to an infinite loop. It lacks intermediate checks that would provide insights into whether the sequence terms are of the form $\frac{t}{t+1}$, which is crucial for solving the problem and would have enabled it to chalk out the termination conditions suitably.

On the other hand, our Step-By-Step Coding method enforces a decomposition of the problem into smaller sub-task. Each sub-task is tackled independently by the LLM, which generates code to solve it and then uses the resulting output to suitably proceed to the next sub-task and this process continues till the final answer is reached. Such an approach ensures that every part of the problem is addressed with exact precision, reducing the risk of errors that might arise from skipped steps. Dividing the problem into multiple sub-tasks also allows it to make necessary simplifications that would make the future sub-tasks, and hence the entire problem, easier to solve.

Going back to the problem in Example 1, SBSC starts by defining the logarithms and setting up the equations based on the geometric progression condition. It then simplifies the equations to reduce them to a more manageable form, eliminating unnecessary complexity and allowing straightforward solving. Throughout the problem, it uses precise mathematical formulations of the problem, ensuring the solution is accurate. Since this method isn't trying to solve the entire problem at one go, it doesn't need to make any assumptions to simplify the problem statement.

For the question in Example 2, it correctly interprets the problem, keeps applying the given Lipschitz condition as it solves each sub-task and finds the correct maximum possible value of $f(f(800)) - f(f(400))$. By systematically checking for constraints and edge cases at each stage, our method guarantees that solutions are not only accurate but also valid under all possible conditions. This is particularly crucial for problems with intricate conditions or multiple solution paths. Additionally, our process explicitly defines loop conditions to ensure that termination criteria are clear, allowing for correction if issues like infinite loops arise.

For the problem in Example 3, the first sub-task correctly identified by SBSC is to factorize the large number $20^{20}$ into smaller powers, paving the path towards an efficient solution. In the subsequent steps, it takes care of constraints like ensuring that m and n are integers and avoids over-counting by properly iterating over valid powers, giving us the correct answer.

Likewise, in Example 4, we see that our method generates each term of the sequence one at a time, allowing for intermediate checks to confirm whether each term is of the desired form $\frac{t}{t+1}$. Thus, it accurately identifies the terms that meet the required condition and ensures that only the correct indices are considered. This avoids the infinite loop risk and ensures the correctness of each step before moving on. We also notice that the model runs into an error at the first sub-task itself but based on the corresponding code output, it is able to take an alternate approach and successfully solves the sub-task, moving onto later ones. This example also highlights our superior code debugging ability, as discussed earlier.

**Example Question 1:** There is a unique positive real number x such that the three numbers $\log_8 2x$, $\log_4 x$ , and $\log_2 x$ , in that order, form a geometric progression with positive common ratio. The number x can be written as $\frac{m}{n}$ , where m and n are relatively prime positive integers. Find m + n

---

**Incorrect solution by TIR-ToRA**

Let's approach this problem step-by-step:

1. Define a function to check if three numbers form a geometric progression.

2. Use binary search to find $x$ that satisfies the condition.

3. Convert $x$ to a fraction and return the sum of its numerator and denominator.

```python
import math
from fractions import Fraction
def solution():
    # Function to check if three numbers form
    #a geometric progression
```

---

```
    def is_geometric_progression(a, b, c):
        ratio1 = b / a
        ratio2 = c / b
        return math.isclose(ratio1, ratio2) and ratio1 > 1
    # Binary search to find x
    left, right = 1, 1000
    while right - left > 1e-10:
        x = (left + right) / 2
        log8_2x = math.log(2*x, 8)
        log4_x = math.log(x, 4)
        log2_x = math.log(x, 2)
        if is_geometric_progression(log8_2x, log4_x, log2_x):
            break
        elif (log4_x / log8_2x) > (log2_x / log4_x):
            left = x
        else:
            right = x
    # Convert x to a fraction
    frac = Fraction(x).limit_denominator()
    return frac.numerator + frac.denominator
result = solution()
print(f"\\boxed{{{result}}}")

>>> output:
\boxed{2}
```

The answer is $\boxed{2}$

## Correct solution by SBSC:

Let's solve this problem step by step using SymPy-based Python code.
Step 1: Define the symbols and set up the equations

```
from sympy import symbols, log, Eq, solve
# Define symbols
x, r = symbols('x r')
# Define the three logarithms
log1 = log(2*x, 8)
log2 = log(x, 4)
log3 = log(x, 2)
# Set up equations for geometric progression
eq1 = Eq(log2 / log1, r)
eq2 = Eq(log3 / log2, r)
print("Equation 1:", eq1)
print("Equation 2:", eq2)

>>> output
Equation 1: Eq(log(8)*log(x)/(log(4)*log(2*x)), r)
Equation 2: Eq(log(4)/log(2), r)
```

Let's continue solving this problem step by step using SymPy-based Python code.
Step 2: Simplify the equations

```
from sympy import symbols, log, Eq, solve, simplify
# Define symbols
x, r = symbols('x r')
# Define the three logarithms
log1 = log(2*x, 8)
log2 = log(x, 4)
log3 = log(x, 2)
# Set up equations for geometric progression
eq1 = Eq(log2 / log1, r)
eq2 = Eq(log3 / log2, r)
```

```
# Simplify equations
eq1_simplified = simplify(eq1)
eq2_simplified = simplify(eq2)
print("Simplified Equation 1:", eq1_simplified)
print("Simplified Equation 2:", eq2_simplified)

>>> output:
Simplified Equation 1: Eq(r, 3*log(x)/(2*log(x) + 2*log(2)))
Simplified Equation 2: Eq(r, 2)
```

Let's continue solving this problem step by step using SymPy-based Python code.
Step 3: Solve the equations

```
from sympy import symbols, log, Eq, solve, simplify
# Define symbols
x, r = symbols('x r')
# Define the simplified equations
eq1 = Eq(r, 3*log(x)/(2*log(x) + 2*log(2)))
eq2 = Eq(r, 2)
# Solve the system of equations
solution = solve((eq1, eq2), (x, r))
print("Solution:", solution)

>>> output
Solution: [(1/16, 2)]
```

Let's continue solving this problem step by step using SymPy-based Python code.
Step 4: Extract the value of x and convert to a fraction

```
from sympy import symbols,log,Eq, solve, simplify, Rational
# Define symbols
x, r = symbols('x r')
# Define the solution from the previous step
solution = [(Rational(1, 16), 2)]
# Extract the value of x
x_value = solution[0][0]
print("x =", x_value)
# Convert x to a fraction
m = x_value.numerator
n = x_value.denominator
print("x as a fraction: {}/{}".format(m, n))

>>> output
x = 1/16
x as a fraction: 1/16
```

Let's continue solving this problem step by step using SymPy-based Python code.
Step 5: Calculate m + n

```
from sympy import symbols,log,Eq, solve, simplify, Rational
# Define symbols
x, r = symbols('x r')
# Define the solution from the previous step
solution = [(Rational(1, 16), 2)]
# Extract the value of x
x_value = solution[0][0]
# Convert x to a fraction
m = x_value.numerator
n = x_value.denominator
# Calculate m + n
result = m + n
print("m =", m)
print("n =", n)
print("m + n =", result)
```

```
>>> output:
m = 1
n = 16
m + n = 17
### END OF CODE
The final answer is 17
```

**Example Question 2:** Consider functions $f$ that satisfy $|f(x) - f(y)| \leq 0.5|x - y|$ for all real numbers $x$ and $y$. Of all such functions that also satisfy the equation $f(300) = f(900)$, what is the greatest possible value of $f(f(800) - f(f(400)))$?

---

**Incorrect solution by TIR-ToRA:**

To solve this problem, we need to consider the properties of the given function and find the maximum possible difference between $f(f(800))$ and $f(f(400))$. Let's solve it using python:

```python
def solution():
    #Given: |f(x) - f(y)| <= 0.5|x - y| for all real x and y
    #Also, f(300) = f(900)
    #Maximum difference between f(800) and f(400)
    max_diff_800_400 = 0.5 * (800 - 400)   # 200
    #Maximum value of f(800) - f(400)
    max_f800_f400 = max_diff_800_400   # 200
    #Maximum difference between f(f(800)) and f(f(400))
    max_final_diff = 0.5 * max_f800_f400   # 100
    return max_final_diff
result = solution()
print(f"\\boxed{{{result}}}")

>>> output:
\boxed{100}
```

The answer is 100

---

**Correct solution by SBSC:**

Let's approach this problem step by step using SymPy-based Python code where appropriate.
Step 1: Understand the given condition The condition |f(x) - f(y)| 1/2|x - y| implies that f is a Lipschitz continuous function with Lipschitz constant 1/2. This means f doesn't change too rapidly.
Step 2: Analyze the equation f(300)=f(900).This equation tells us that f takes the same value at x=300 and x=900. Let's call this value k.

```python
from sympy import symbols, Abs
k = symbols('k')
print(f"f(300) = f(900) = {k}")

>>> output:
f(300) = f(900) = k
```

Let's continue solving this problem step by step.
Step 3: Apply the Lipschitz condition to f(800) and f(400) We can use the Lipschitz condition to find bounds for f(800) and f(400) in terms of k.

```python
from sympy import symbols, Abs, Min, Max
k = symbols('k')
# For f(800)
f_800_min = k - Abs(800 - 900)/2
f_800_max = k + Abs(800 - 900)/2
# For f(400)
f_400_min = k - Abs(400 - 300)/2
```

```
f_400_max = k + Abs(400 - 300)/2
print(f"f(800) is between {f_800_min} and {f_800_max}")
print(f"f(400) is between {f_400_min} and {f_400_max}")

>>> output:
f(800) is between k - 50 and k + 50
f(400) is between k - 50 and k + 50
```

Let's continue solving this problem step by step.
Step 4: Analyze $f(f(800))$ and $f(f(400))$ Now we need to apply the Lipschitz condition again to $f(f(800))$ and $f(f(400))$.

```
from sympy import symbols, Abs, Min, Max
k = symbols('k')
# For f(f(800))
ff_800_min = k - Abs(k + 50 - k)/2
ff_800_max = k + Abs(k - 50 - k)/2
# For f(f(400))
ff_400_min = k - Abs(k + 50 - k)/2
ff_400_max = k + Abs(k - 50 - k)/2
print(f"f(f(800)) is between {ff_800_min} and {ff_800_max}")
print(f"f(f(400)) is between {ff_400_min} and {ff_400_max}")

>>> output
f(f(800)) is between k - 25 and k + 25
f(f(400)) is between k - 25 and k + 25
```

Let's continue solving this problem step by step.
Step 5: Calculate the maximum possible difference Now we need to find the maximum possible value of $f(f(800)) - f(f(400))$.

```
from sympy import symbols, Abs, Min, Max
k = symbols('k')
# Maximum possible difference
max_diff = (k + 25) - (k - 25)
print(f"The maximum possible difference f(f(800))-f(f(400))
is {max_diff}")

>>> output:
The maximum possible difference f(f(800)) - f(f(400)) is 50
```

### END OF CODE
The final answer is $\boxed{50}$

**Example Question 3:** Find the number of ordered pairs of positive integers $(m, n)$ such that $m^2 n = 20^{20}$

---

**Incorrect solution by TIR-ToRA:**

Let's approach this problem step-by-step:

1. Iterate through possible values of m up to the square root of $20\hat{2}0$
2. For each m, check if $m^2$ divides $20^{20}$ evenly
3. Return the total count of valid pairs

```
import math
def solution():
    count = 0
    target = 20**20
    for m in range(1, int(math.sqrt(target)) + 1):
        if target % (m**2) == 0:
            n = target // (m**2)
            count += 1
```

```
        return count
result = solution()
print(f"The number of ordered pairs (m,n) is:
f"\\boxed{{{result}}}"")

>>> output:
TimeoutError: [Errno 110] Connection timed out
```

Looks like we got a timeout error, let's try a different approach that should not result in any error:

```
import math
def solution():
    count = 0
    target = 20**20
    for m in range(1, int(math.sqrt(target)) + 1):
        if target % (m**2) == 0:
            n = target // (m**2)
            count += 1
    return count
result = solution()
print(f"The number of ordered pairs (m,n) is:
f"\\boxed{{{result}}}"")

>>> output:
TimeoutError: [Errno 110] Connection timed out

...
```
This loop continues without ultimately getting an answer.

---

**Correct solution by SBSC:**

Let's approach this problem step by step using SymPy-based Python code.
Step 1: Define the equation and initialize variables

```
from sympy import symbols,Eq,solve, divisors, prime_factors
m, n = symbols('m n', positive=True, integer=True)
equation = Eq(m**2 * n, 20**20)
print("Equation:", equation)

>>> output:
ImportError: cannot import name 'prime_factors' from 'sympy'
```

I apologize for the error. It seems the 'prime factors' function is not available in the version of SymPy you're using. Let's modify our approach and use the 'factorint' function instead, which is more commonly available. Here's the corrected code:
Step 1: Define the equation and initialize variables

```
from sympy import symbols, Eq, solve, divisors, factorint
m, n = symbols('m n', positive=True, integer=True)
equation = Eq(m**2 * n, 20**20)
print("Equation:", equation)

>>> output
Equation: Eq(m**2*n, 104857600000000000000000000)
```

Let's continue solving this problem step by step.
Step 2: Factor $20^{20}$ and analyze its structure

```
from sympy import symbols, Eq, solve, divisors, factorint
m, n = symbols('m n', positive=True, integer=True)
equation = Eq(m**2 * n, 20**20)
# Factor 20^20
factorization = factorint(20**20)
```

```
print("Factorization of 20^20:", factorization)
# Analyze the structure
power_of_2 = factorization[2]
power_of_5 = factorization[5]
print("Power of 2:", power_of_2)
print("Power of 5:", power_of_5)

>>> output
Factorization of 20^20: {2: 40, 5: 20}
Power of 2: 40
Power of 5: 20
```

Let's continue solving this problem step by step.
Step 3: Determine possible values for m and n

```
from sympy import symbols, Eq, solve, divisors, factorint
m, n = symbols('m n', positive=True, integer=True)
equation = Eq(m**2 * n, 20**20)
# Factor 20^20
factorization = factorint(20**20)
power_of_2 = factorization[2]
power_of_5 = factorization[5]
# Initialize counter for valid pairs
valid_pairs = 0
# Iterate through possible values of m
for m_power_of_2 in range(0, power_of_2 + 1, 2):
    for m_power_of_5 in range(0, power_of_5 + 1):
        # Calculate m
        m_value = 2**m_power_of_2 * 5**m_power_of_5
        # Calculate corresponding n
        n_power_of_2 = power_of_2 - 2*m_power_of_2
        n_power_of_5 = power_of_5 - m_power_of_5
        # Check if n is a positive integer
        if n_power_of_2 >= 0 and n_power_of_5 >= 0:
            valid_pairs += 1
print("Number of valid (m,n) pairs:", valid_pairs)

>>> output
Number of valid (m,n) pairs: 231
```

### END OF CODE
The final answer is $\boxed{231}$

**Example Question 4:** Consider the sequence $(a_k)\_k \geq 1$ of positive rational numbers defined by $a_1 = \frac{2020}{2021}$ and for $k \geq 1$, if $a_k = \frac{m}{n}$ for relatively prime positive integers m and n, then $a_{k+1} = \frac{m+18}{n+19}$. Determine the sum of all positive integers j such that the rational number $a_j$ can be written in the form $\frac{t}{t+1}$ for some positive integer t

---

Incorrect solution by TIR-ToRA:

Let's proceed step-by-step:

1. Generate the sequence of fractions starting from $a_1 = \frac{2020}{2021}$.

2. For each term, check if it is in the form $\frac{t}{t+1}$ for some integer $t$. If so, add its index to a running sum.

3. Generate the next term using the given formula.

4. Continue this process until we cycle back to the initial fraction. The sum of indices where the fraction is in the form $\frac{t}{t+1}$ is our final answer.

```
from fractions import Fraction
```

```python
def next_term(a):
    return Fraction(a.numerator + 18, a.denominator + 19)
def is_of_form_t_over_t_plus_1(frac):
    return frac.denominator == frac.numerator + 1
def solution():
    a = Fraction(2020, 2021)
    j_sum = 0
    j = 1
    while True:
        if is_of_form_t_over_t_plus_1(a):
            j_sum += j
        a = next_term(a)
        j += 1
        # Check if we've reached the initial fraction again
        if a == Fraction(2020, 2021):
            break
    return j_sum
result = solution()
print(f"\\boxed{{{result}}}")
```

Similar to the earlier example, this code results in a TimeOut error and then the loop keeps repeating.

**Correct solution by SBSC:**

Let's solve this problem step by step using SymPy-based Python code.
Step 1: Define the sequence and initial value

```python
from sympy import Rational, gcd
def next_term(a):
    m, n = a.numerator, a.denominator
    return Rational(m + 18, n + 19)
a = [Rational(2020, 2021)]
print(f"a[1] = {a[0]}")
```

```
>>> output
a[1] = 2020/2021
```

Let's continue solving this problem step by step using SymPy-based Python code.
Step 2: Generate the sequence until we find a term of the form t/(t+1)

```python
from sympy import Rational, gcd
def next_term(a):
    m, n = a.numerator, a.denominator
    return Rational(m + 18, n + 19)
a = [Rational(2020, 2021)]
j = 1
result = []
while True:
    if a[-1].denominator == a[-1].numerator + 1:
        result.append(j)
        print(f"Found a[{j}] = {a[-1]} of the form t/(t+1)")
    j += 1
    a.append(next_term(a[-1]))
    if j > 100:  # Limit to prevent infinite loop
        break
print(f"Indices found: {result}")
```

```
>>> output
Found a[1] = 2020/2021 of the form t/(t+1)
Found a[2] = 1019/1020 of the form t/(t+1)
Found a[8] = 161/162 of the form t/(t+1)
Found a[18] = 31/32 of the form t/(t+1)
Found a[30] = 19/20 of the form t/(t+1)
```

```
Indices found: [1, 2, 8, 18, 30]
```

Let's continue solving this problem step by step using SymPy-based Python code.
Step 3: Calculate the sum of the indicesfrom sympy import Rational, gcd

```python
def next_term(a):
    m, n = a.numerator, a.denominator
    return Rational(m + 18, n + 19)
a = [Rational(2020, 2021)]
j = 1
result = []
while True:
    if a[-1].denominator == a[-1].numerator + 1:
        result.append(j)
    j += 1
    a.append(next_term(a[-1]))
    if j > 100:  # Limit to prevent infinite loop
        break
sum_of_indices = sum(result)
print(f"Indices found: {result}")
print(f"Sum of indices: {sum_of_indices}")

>>> output
Indices found: [1, 2, 8, 18, 30]
Sum of indices: 59
```

### END OF CODE
The final answer is 59

## A.7   COMPARISON ON SYMPY USAGE BETWEEN SBSC AND TIR-TORA

In this section, we conduct a study to understand the usage of SymPy library by SBSC and TIR-ToRA for last 10 years of AIME and AMC questions, shown in Table 13. Specifically, we present the number of questions in which these methods utilized that library, and we present the accuracy for those set of questions. The number of questions is which both the methods utilized SymPy are similar. For both AIME and AMC problems from the last 10 years, SBSC based solutions have used SymPy only on 7 more questions in both test-sets. But when we look at the accuracy (among these filtered questions), we can observe that SBSC had over 10% absolute accuracy lead in both the datasets.

Table 13: Comparison between TIR-ToRA and SBSC on questions solved utilising SymPy

| Context | AIME (#Qs) | AIME (Accuracy %) | AMC (#Qs) | AMC (Accuracy %) |
|---------|-----------|-------------------|-----------|------------------|
| TIR-ToRA | 200 | 15.50 | 299 | 36.79 |
| SBSC | 207 | 30.43 | 306 | 46.73 |

Table 14: Performance on common questions involving Sympy Usage by both the methods

| Dataset | # Common Qs with Sympy Usage | TIR-ToRA (Accuracy %) | SBSC (Accuracy %) |
|---------|------------------------------|-----------------------|-------------------|
| AIME | 163 | 15.95 | 28.22 |
| AMC | 252 | 38.49 | 47.62 |

Also, we further filtered the common questions, for both the test-sets, where both TIR-ToRA and SBSC made use of Sympy. We present this study in the table 14. We observed that more than 80% questions are common for both the methods across the test sets. In table 14, we compare the accuracy of both the methods over these common questionsand find that SBSC has 10% absolute improvement across both the test-sets.

Table 15: Pass@k scores for SBSC and o1 scores on last 3 years AIME questions.

| Topic | SBSC | | | | | o1 | |
|---|---|---|---|---|---|---|---|
| | pass@1 | pass@4 | pass@9 | pass@16 | pass@25 | preview | mini |
| Algebra (21 Qs) | 11 | 14 | 16 | 16 | **19** | 12 | 14 |
| Combinatorics (23 Qs) | 8 | 10 | 12 | **13** | **13** | **13** | 10 |
| Geometry (32 Qs) | 2 | 3 | 4 | 9 | 9 | 10 | **13** |
| Number Theory (14 Qs) | 7 | 9 | 9 | **11** | **11** | 4 | 7 |
| Total (90 Qs) | 28 | 36 | 41 | 49 | **52** | 39 | 44 |

A.8    STUDYING THE POTENTIAL OF SBSC TRAJECTORIES AND COMPARISON WITH O1

**SBSC achieves SOTA but still room for improvement in accuracy:** For frontier LLMs such as Claude 3.5 Sonnet, we showed that SBSC surpasses not only greedy-decoding but also self-consistency decoding results of TIR-ToRA, PAL & COT. We also show topic wise analysis as well to show the improvement is across the topics. While we do achieve SOTA, accuracy but there is still room for improvement as evident from the scores. On last 11 years of AIME questions, despite being SOTA SBSC achieves  36% accuracy in pass@1. So there is a question on how much can SBSC trajectory really solve more.

**o1 like complex reasoning methods achieves new SOTA for COT based math reasoning:** Sophisticated reasoning systems like o1 (OpenAI, 2024) perform "long thinking" to do potentially tree-search/self-reflection/debate over natural-language reasoning chains to arrive at the final trajectory (Huang et al., 2024; Qin et al., 2024; OpenAI, 2024). They achieve significantly higher scores, on AIME and AMC, compared to previous COT scores by frontier LLMs such as Sonnet and GPT4o and use human like COT reasoning.

Table 16: Analysis of Error Classes in Different Mathematical Domains for last 3 Yrs AIME

| Error Type | Algebra (21Q) | Combinatorics (23Q) | Geometry (32Q) | Number Theory (14Q) |
|---|---|---|---|---|
| Reasoning Error | 0 | 10 | 10 | 1 |
| Hallucination | 1 | 0 | 1 | 1 |
| Spatial Misunderstanding | 0 | 0 | 7 | 0 |
| Incorrect Answer Reported | 1 | 0 | 5 | 0 |
| Total Errors | 2 | 10 | 23 | 2 |

**Error definition:** Reasoning errors include mistakes due to incorrect reasoning steps or missing conditions. Hallucination refers to fabrication of numbers or answers. Spatial misunderstanding involves misinterpretation of given diagrams or geometric scenarios. Incorrect Answer Reported refers to cases with correct reasoning solution but the final answer is wrong.

**Understanding SBSC's potential with pass@k metric and surpassing o1:** We argue that SBSC given its unique step-by-step coding nature holds lot of potential. It unlocks dynamic problem solving via discovering and solving intermediate sub-tasks.

- We evaluate this first via quantitative analysis by calculating pass@k scores on last 3 years AIME questions
- We observe that SBSC's pass@16 and pass@25 are able to solve 48 and 52 questions out of 90 AIME questions respectively. Pass@25 SBSC score surpasses o1 mini by 9% and o1 preview by13%. We also do topic wise analysis to understand the strengths and weaknesses of SBSC.

- We further do error analysis on the 38 questions that couldn't be solved with pass@25 SBSC. We do this across the math topics. We define different error types.

- By doing error analysis, we are able to understand that SBSC made majorly reasoning errors during program generation leading to incorrect code. Hence majority of those 38 questions can be solved by SBSC reasoning format if the model avoids certain reasoning error.

- However, we also understand that specifically in Geometry questions, SBSC struggles to achieve o1 level accuracy even at pass@25. We mainly attribute this failure to diagram/spatial understanding.

**Note/Scope of this study:** This study is to understand the potential of SBSC and not about claiming COT like reasoning is not sufficient. We believe SBSC based program generation based method could also unlock advanced mathematical reasoning. We have already demonstrated this across four datasets in the paper and 2 more ( JEE-Bench and Omni-Math) in appendix. We try to show more evidences with this study. Given, o1 like new complex reasoning systems unlocks the SOTA/superior COT trajectory, so we also perform a comparison against them.

**Analysis on SBSC's pass@k accuracy on last 3 years AIME**

- **Why use pass@k for SBSC:** Here we leverage self-consistency strategy to make model generate multiple SBSC trajectories to examine the pass@k curve by increasing k from 1 to 25. We are using pass@k metric as we want to know if the model can generate a correct SBSC based reasoning path for a particular problem. o1 explores multiple paths/trajectories to arrive at the correct trajectory. Once we analyse SBSC's pass@k for a few values of k, it can help us better understand that when we build critic systems or thinking systems that leverage SBSC, how much can these systems achieve. Hence in this study we focus on pass@k for SBSC to understand its potential. We also benchmark against o1 preview and o1 mini. We do pass@1 for the o1 models as they already do through slow/long reasoning internally before outputting the final correct reasoning path. Also since they are new SOTA COT based method.

- **Experiment details:** We do this study on last 3 years AIME questions. We use claude-3.5-sonnet as the base LLM. For self-consistency we use temperature 1 and top p = 0.95. o1 mini and o1 preview represent the current SOTA COT based methods.

- **Results:** We present pass@1, pass@4, pass@16, pass@25 for SBSC in Table 15

  - **Overall analysis:** We observe that SBSC with just pass@4 approaches o1 level accuracy and with pass@9 it surpasses o1-preview and almost touches o1-mini. With pass@16 and pass@25 SBSC takes significant lead wr.t o1 models. At pass@16 SBSC shows 6% absolute improvement with best o1 mini and 11% improvement over o1-preview. At pass@25 SBSC shows 9% absolute improvement compared to o1 mini and 13% improvement over o1 preview. We can clearly understand that SBSC can potentially unlock more correct chains and solve 52 questions out of 90

  - **Topic wise analysis:**
    * Algebra: We can observe that with just pass@4, SBSC achieves o1 mini level performance and with pass@25 setting, SBSC solves 90% of the algebra questions
    * Number Theory: With just pass@1, SBSC surpasses o1 preview and matches o1 mini. At pass@16, SBSC is able to solve with 78% accuracy.
    * Combinatorics: At pass@16 and pass@25, SBSC matches o1 preview and surpasses o1-mini by absolute 13%.
    * Geometry: This is the only topic where SBSC lacks behind o1 models.

- **Error Analysis:** Within pass@25 the model was not able to generate correct SBSC trajectory for 38 AIME questions out of 90 questions. We show topic wise error distribution below:

  - 2 out of 21 from Algebra.
  - 10 out of 23 from combinatorics
  - 23 out of 32 from geometry
  - 2 out of 14 from number theory

  We manually go through the trajectories for these 38 questions. So we analyse 38 SBSC paths. We classify the errors in multiple categories as shown in Table 16.

## A.9 PAL EXEMPLARS

In this section, we provide the prompts for Program-Aided Language models (PAL) method. We initially used the default prompt as mentioned in the original PAL paper, but the results were poor. We noticed that the response often contained textual reasoning before or after the program, which isn't the desired format for PAL. Hence, we modify the instructions to confine the responses only to include Python program and subsequently, also notice improved accuracy.

**For AIME**
Let's use python program to solve math problems.
DO NOT USE ANY TEXTUAL REASONING.
Your response must start with: "'python
Your response must end with: print(result)

Here are some examples you may refer to.

**Example Problem:** A frog begins at $P_0 = (0,0)$ and makes a sequence of jumps according to the following rule: from $P_n = (x_n, y_n)$, the frog jumps to $P_{n+1}$, which may be any of the points $(x_n + 7, y_n + 2)$, $(x_n + 2, y_n + 7)$, $(x_n - 5, y_n - 10)$, or $(x_n - 10, y_n - 5)$. There are $M$ points $(x, y)$ with $|x| + |y| \leq 100$ that can be reached by a sequence of such jumps. Find the remainder when $M$ is divided by 1000.

**Example Solution:**

```python
def solution():
    jumps = [(7, 2), (2, 7), (-5, -10), (-10, -5)]
    # Set to keep track of all reachable points, starting from the origin
     (0, 0).
    reachable = set([(0, 0)])
    # Queue to process points, starting with the origin (0, 0).
    queue = [(0, 0)]
    # Breadth-first search (BFS) to explore reachable points.
    while queue:
        # Pop the first point from the queue.
        x, y = queue.pop(0)
        # Iterate over all possible jumps.
        for dx, dy in jumps:
            # Calculate new coordinates after the jump.
            nx, ny = x + dx, y + dy
            # Check if the Manhattan distance is within 100 and the point
     hasn't been visited.
            if abs(nx) + abs(ny) <= 100 and (nx, ny) not in reachable:
                # Add the new point to the reachable set.
                reachable.add((nx, ny))
                # Add the new point to the queue to explore further.
                queue.append((nx, ny))
    return len(reachable) % 1000
result = solution()
print(result)
```

**Example Problem:** The AIME Triathlon consists of a half-mile swim, a 30-mile bicycle ride, and an eight-mile run. Tom swims, bicycles, and runs at constant rates. He runs fives times as fast as he swims, and he bicycles twice as fast as he runs. Tom completes the AIME Triathlon in four and a quarter hours. How many minutes does he spend bicycling?

**Example Solution:**

```python
from sympy import symbols, Eq, solve, Rational
def solution():
    x = symbols('x')
    # Set up the equation
    eq = Eq(Rational(1,2)/x + 30/(10*x) + 8/(5*x), Rational(17,4))
    # Solve the equation
    solution = solve(eq)[0]
    # Calculate bicycling time in hours
```

```
    bike_time = 30 / (10 * solution)
    # Convert to minutes
    bike_time_minutes = int(bike_time * 60)
    return bike_time_minutes
result = solution()
print result
```

**Example Problem:** Let $S$ be the increasing sequence of positive integers whose binary representation has exactly 8 ones. Let $N$ be the 1000th number in $S$. Find the remainder when $N$ is divided by 1000

**Example Solution:**

```
def solution():
    count = 0  # Initialize a counter to track how many numbers have been
    found
    n = 1  # Start checking numbers from 1 upwards
    while count < 1000:  # Continue the loop until we find the 1000th
    number
        # Check if the binary representation of the number 'n' has
    exactly 8 '1's
        if bin(n).count('1') == 8:
            count += 1  # Increment the counter when a number with 8 '1's
     is found
            # If this is the 1000th such number, return the remainder of
    n divided by 1000
            if count == 1000:
                return n % 1000
        n += 1  # Move to the next number
result = solution()
print(result)
```

**Example Problem:** Two geometric sequences $a_1, a_2, a_3, \ldots$ and $b_1, b_2, b_3, \ldots$ have the same common ratio, with $a_1 = 27$ $b_1 = 99$, and $a_{15} = b_{11}$. Find $a_9$

**Example Solution:**

```
def solution():
    # Initialize known values
    a1 = 27
    b1 = 99
    # Calculate the common ratio
    # We know that a15 = b11, so:
    # a1 * r^14 = b1 * r^10
    # 27 * r^14 = 99 * r^10
    # 27 * r^4 = 99
    # r^4 = 99/27 = 11/3
    r = (11/3) ** (1/4)
    # Calculate a9
    a9 = a1 * (r ** 8)
    return round(a9)
result = solution()
print(result)
```

**For AMC:**

Let's use python program to solve math problems.
DO NOT USE ANY TEXTUAL REASONING.
Your response must start with: "'python
Your response must end with: print(result)

Here are some examples you may refer to.

**Example Problem:** Small lights are hung on a string 6 inches apart in the order red, red, green, green, green, red, red, green, green, green, and so on continuing this pattern of 2 red lights followed by 3 green lights. How many feet separate the 3rd red light and the 21st red light? Note: 1 foot is equal to 12 inches.

**Example Solution:**

```python
def solution():
    # Find position of 3rd red light
    n_3rd = 3
    complete_cycles_3rd = (n_3rd - 1) // 2
    remaining_lights_3rd = (n_3rd - 1) % 2
    pos_3rd = complete_cycles_3rd * 5 * 6 + remaining_lights_3rd * 6
    # Find position of 21st red light
    n_21st = 21
    complete_cycles_21st = (n_21st - 1) // 2
    remaining_lights_21st = (n_21st - 1) % 2
    pos_21st = complete_cycles_21st * 5 * 6 + remaining_lights_21st * 6
    # Calculate the distance in inches
    distance_inches = pos_21st - pos_3rd
    # Convert to feet
    distance_feet = distance_inches / 12
    return distance_feet
result = solution()
print(result)
```

**Example Problem:** A fruit salad consists of blueberries, raspberries, grapes, and cherries. The fruit salad has a total of 280 pieces of fruit. There are twice as many raspberries as blueberries, three times as many grapes as cherries, and four times as many cherries as raspberries. How many cherries are there in the fruit salad?

**Example Solution:**

```python
from sympy import symbols, Eq, solve
def solution():
    # Define the symbols for the variables
    b, r, g, c = symbols('b r g c')
    # Define the equations based on the problem statement
    eq1 = Eq(r, 2*b)         # Equation 1: r = 2b
    eq2 = Eq(g, 3*c)         # Equation 2: g = 3c
    eq3 = Eq(c, 4*r)         # Equation 3: c = 4r
    eq4 = Eq(b + r + g + c, 280)  # Equation 4: b + r + g + c = 280
    # Solve the system of equations
    sol = solve((eq1, eq2, eq3, eq4))
    return sol[c]
result = solution()
print(result)
```

**Example Problem:** Last summer 30% of the birds living on Town Lake were geese, 25% were swans, 10% were herons, and 35% were ducks. What percent of the birds that were not swans were geese?

**Example Solution:**

```python
def solution():
    # Total percentage of all birds
    total = 100
    # Percentages of each bird type
    geese = 30
    swans = 25
    herons = 10
    ducks = 35
    # Calculate percentage of birds that are not swans
    not_swans = total - swans
    # Calculate percentage of geese among birds that are not swans
    geese_among_not_swans = (geese / not_swans) * 100
    # Round to nearest whole number
    return round(geese_among_not_swans)
result = solution()
print(result)
```

Example Problem: At a twins and triplets convention, there were 9 sets of twins and 6 sets of triplets, all from different families. Each twin shook hands with all the twins except his/her siblings and with half the triplets. Each triplet shook hands with all the triplets except his/her siblings and with half the twins. How many handshakes took place?

**Example Solution:**

```python
def solution():
    # Number of twins and triplets
    twins = 9 * 2
    triplets = 6 * 3
    # Handshakes between twins
    twin_handshakes = (twins * (twins - 2)) // 2
    # Handshakes between triplets
    triplet_handshakes = (triplets * (triplets - 3)) // 2
    # Handshakes between twins and triplets
    twin_triplet_handshakes = (twins * triplets) // 2
    # Total handshakes
    total_handshakes = twin_handshakes + triplet_handshakes +
    twin_triplet_handshakes
    return total_handshakes
result = solution()
print(result)
```

## A.10 TIR-ToRA Exemplars

**For AIME**
Integrate step-by-step reasoning and Python code to solve math problems using the following guidelines:

- Analyze the question and write functions to solve the problem; the function should not take any arguments.
- Present the final result in LaTeX using a '□' without any units.
- Utilize the 'pi' symbol and 'Rational' from Sympy for $\pi$ and fractions, and simplify all fractions and square roots without converting them to decimal values.

Once you get the code output, just display the output answer within '□' and terminate.

Here are some examples you may refer to:

**Example Problem:** A frog begins at $P_0 = (0, 0)$ and makes a sequence of jumps according to the following rule: from $P_n = (x_n, y_n)$, the frog jumps to $P_{n+1}$, which may be any of the points $(x_n + 7, y_n + 2)$, $(x_n + 2, y_n + 7)$, $(x_n - 5, y_n - 10)$, or $(x_n - 10, y_n - 5)$. There are $M$ points $(x, y)$ with $|x| + |y| \leq 100$ that can be reached by a sequence of such jumps. Find the remainder when $M$ is divided by 1000.

**Example Solution:**
Let's approach this problem step-by-step:

1. We need to find all points (x, y) that the frog can reach within the given constraint |x| + |y| 100.

2. We'll use a breadth-first search (BFS) algorithm to explore all possible points.

3. We'll keep track of visited points to avoid revisiting them.

4. We'll count the number of unique points visited.

Here's the Python code to solve this:

```python
from collections import deque
def count_reachable_points():
    # Possible jumps
    jumps = [(7, 2), (2, 7), (-5, -10), (-10, -5)]
    # Set to store visited points
    visited = set([(0, 0)])
```

```
    # Queue for BFS
    queue = deque([(0, 0)])
    while queue:
        x, y = queue.popleft()
        for dx, dy in jumps:
            new_x, new_y = x + dx, y + dy
            # Check if the new point is within the constraint and not
    visited
            if abs(new_x) + abs(new_y) <= 100 and (new_x, new_y) not in
    visited:
                visited.add((new_x, new_y))
                queue.append((new_x, new_y))
    # Count of reachable points is the size of visited set
    M = len(visited)
    # Return the remainder when M is divided by 1000
    return M % 1000
result = count_reachable_points()
print(f"\\boxed{{{result}}}")

>>> output
\boxed{373}
```

The answer is $\boxed{373}$

**Example Problem:** The AIME Triathlon consists of a half-mile swim, a 30-mile bicycle ride, and an eight-mile run. Tom swims, bicycles, and runs at constant rates. He runs fives times as fast as he swims, and he bicycles twice as fast as he runs. Tom completes the AIME Triathlon in four and a quarter hours. How many minutes does he spend bicycling?

**Example Solution:**
Let's approach this problem step-by-step:

1. Let's define variables:
   - Let x be Tom's swimming speed in miles per hour
   - Then 5x is his running speed in miles per hour
   - And 10x is his bicycling speed in miles per hour

2. We can set up equations based on the given information:
   - Swimming time: 0.5 / x hours
   - Bicycling time: 30 / (10x) hours
   - Running time: 8 / (5x) hours

3. The total time is 4.25 hours, so we can set up an equation:
   0.5/x + 30/(10x) + 8/(5x) = 4.25

4. Let's solve this equation and find the bicycling time.

Here's the Python code to solve this:

```
from sympy import symbols, Eq, solve, Rational
def solve_triathlon():
    x = symbols('x')
    # Set up the equation
    eq = Eq(Rational(1,2)/x + 30/(10*x) + 8/(5*x), Rational(17,4))
    # Solve the equation
    solution = solve(eq)[0]
    # Calculate bicycling time in hours
    bike_time = 30 / (10 * solution)
    # Convert to minutes
    bike_time_minutes = int(bike_time * 60)
    return f"\\boxed{{{bike_time_minutes}}}"
print(solve_triathlon())

>>> output:
\boxed{150}
```

The answer is $\boxed{150}$

**Example Problem:** Let $S$ be the increasing sequence of positive integers whose binary representation has exactly 8 ones. Let $N$ be the 1000th number in $S$. Find the remainder when $N$ is divided by 1000

**Example Solution:**
Let's approach this step-by-step:

1. We need to generate the sequence S of numbers with exactly 8 ones in their binary representation.

2. We'll find the 1000th number in this sequence.

3. Finally, we'll calculate the remainder when this number is divided by 1000.

Here's the Python code to solve this:

```python
def count_ones(n):
    return bin(n).count('1')
def solution():
    count = 0
    n = 1
    while count < 1000:
        if count_ones(n) == 8:
            count += 1
            if count == 1000:
                return n % 1000
        n += 1
result = solution()
print(result)

>>> output:
\boxed{32}
```

The answer is $\boxed{32}$

**Example Problem:** Two geometric sequences $a_1, a_2, a_3, \ldots$ and $b_1, b_2, b_3, \ldots$ have the same common ratio, with $a_1 = 27\, b_1 = 99$, and $a_{15} = b_{11}$. Find $a_9$

**Example Solution:**
Let's approach this step-by-step:

1. Let's denote the common ratio as r.

2. We can express a and b in terms of their first terms and r.

3. We can then equate these expressions to find r.

4. Once we have r, we can calculate a.

Here's the Python code to solve this:

```python
from sympy import symbols, Eq, solve, Rational
def find_a9():
    r = symbols('r')
    #   a     =  a   *  r
    #   b     =  b   *  r
    #   a     =    b
    eq = Eq(27 * r**14, 99 * r**10)
    # Solve for r
    solution = solve(eq)
    # We expect only one positive real solution
    r_value = [sol for sol in solution if sol.is_real and sol > 0][0]
    # Calculate  a
    a9 = 27 * r_value**8
    return a9
result = find_a9()
```

```
print(f"\\boxed{{{result}}}")

>>> output
\boxed{363}
```

The answer is $\boxed{363}$

**For AMC**

Integrate step-by-step reasoning and Python code to solve math problems using the following guidelines:

- Analyze the question and write functions to solve the problem; the function should not take any arguments. - Present the final result in LaTeX using a '□' without any units. - Utilize the 'pi' symbol and 'Rational" from Sympy for $\pi$ and fractions, and simplify all fractions and square roots without converting them to decimal values.

Once you get the code output, just display the output answer within '□' and terminate. Here are some examples you may refer to:

**Example Problem:** Small lights are hung on a string 6 inches apart in the order red, red, green, green, green, red, red, green, green, green, and so on continuing this pattern of 2 red lights followed by 3 green lights. How many feet separate the 3rd red light and the 21st red light? Note: 1 foot is equal to 12 inches.

**Example Solution:**
Let's approach this step-by-step:

1. First, we need to understand the pattern of lights:
   - Every 5 lights (2 red + 3 green) complete one cycle of the pattern.
   - Each cycle is 30 inches long (5 lights * 6 inches).

2. We need to find how many complete cycles are between the 3rd and 21st red lights:
   - 3rd red light is at the end of the 2nd cycle (as each cycle has 2 red lights).
   - 21st red light is at the end of the 11th cycle (as 21 ÷ 2 = 10.5, rounded up to 11).
   - So there are 9 complete cycles between them (11 - 2 = 9).

3. Calculate the distance:
   - 9 cycles * 30 inches per cycle = 270 inches
   - Convert 270 inches to feet: 270 ÷ 12 = 22.5 feet

Here's the Python code to solve this:

```
from sympy import Rational
def calculate_distance():
    lights_per_cycle = 5
    inches_per_light = 6
    inches_per_cycle = lights_per_cycle * inches_per_light
    start_cycle = 2   # 3rd red light is at the end of 2nd cycle
    end_cycle = 11    # 21st red light is at the end of 11th cycle
    cycles_between = end_cycle - start_cycle
    total_inches = cycles_between * inches_per_cycle
    total_feet = Rational(total_inches, 12)
    return f"\\boxed{{{total_feet}}}"
print(calculate_distance())

>>> output:
\boxed{45/2}
```

The answer is $\boxed{45/2}$

**Example Problem:** A fruit salad consists of blueberries, raspberries, grapes, and cherries. The fruit salad has a total of 280 pieces of fruit. There are twice as many raspberries as blueberries, three times as many grapes as cherries, and four times as many cherries as raspberries. How many cherries are there in the fruit salad?

**Example Solution:**
Let's approach this problem step-by-step:

1. Let's define variables for each fruit:
   b = number of blueberries
   r = number of raspberries
   g = number of grapes
   c = number of cherries

2. We can set up equations based on the given information:
   r = 2b (twice as many raspberries as blueberries)
   g = 3c (three times as many grapes as cherries)
   c = 4r (four times as many cherries as raspberries)

3. We also know that the total number of fruits is 280:
   b + r + g + c = 280

4. Now, let's write a Python function to solve this system of equations:

```python
from sympy import symbols, Eq, solve
def solve_fruit_salad():
    b, r, g, c = symbols('b r g c')
    eq1 = Eq(r, 2*b)
    eq2 = Eq(g, 3*c)
    eq3 = Eq(c, 4*r)
    eq4 = Eq(b + r + g + c, 280)
    solution = solve((eq1, eq2, eq3, eq4))
    return f"\\boxed{{{solution[c]}}}"
print(solve_fruit_salad())

>>> output:
\boxed{64}
```

The answer is $\boxed{64}$

**Example Problem:** Last summer $30\%$ of the birds living on Town Lake were geese, $25\%$ were swans, $10\%$ were herons, and $35\%$ were ducks. What percent of the birds that were not swans were geese?

**Example Solution:**
Let's approach this step-by-step:

1. First, we need to calculate the percentage of birds that were not swans.

2. Then, we'll calculate what percentage of this non-swan group were geese.

Here's the Python code to solve this:

```python
from sympy import Rational
def calculate_geese_percentage():
    # Total percentage of birds
    total = 100
    # Percentage of swans
    swans = 25
    # Percentage of non-swans
    non_swans = total - swans
    # Percentage of geese among all birds
    geese = 30
    # Calculate the percentage of geese among non-swans
    geese_among_non_swans = Rational(geese, non_swans) * 100
    # Simplify the fraction
    result = geese_among_non_swans.simplify()
    return f"\\boxed{{{result}}}"
print(calculate_geese_percentage())

>>> output
\boxed{40}
```

The answer is $\boxed{40}$

**Example Problem**: At a twins and triplets convention, there were 9 sets of twins and 6 sets of triplets, all from different families. Each twin shook hands with all the twins except his/her siblings and with half the triplets. Each triplet shook hands with all the triplets except his/her siblings and with half the twins. How many handshakes took place?

**Example Solution:**
Let's approach this problem step-by-step:

1. Calculate the number of handshakes between twins

2. Calculate the number of handshakes between triplets

3. Calculate the number of handshakes between twins and triplets

4. Sum up all handshakes

Here's the Python code to solve this:

```python
from sympy import Rational
def calculate_handshakes():
    # Number of twins and triplets
    twins = 9 * 2
    triplets = 6 * 3
    # Handshakes between twins
    twin_handshakes = (twins * (twins - 2)) // 2  # Each twin shakes
    hands with all but 1 other twin
    # Handshakes between triplets
    triplet_handshakes = (triplets * (triplets - 3)) // 2  # Each triplet
     shakes hands with all but 2 other triplets
    # Handshakes between twins and triplets
    twin_triplet_handshakes = twins * triplets * Rational(1, 2)  # Each
    twin shakes hands with half the triplets
    # Total handshakes
    total_handshakes = twin_handshakes + triplet_handshakes +
    twin_triplet_handshakes
    return f"\\boxed{{{int(total_handshakes)}}}"
print(calculate_handshakes())

>>> output
\boxed{441}
```

The answer is $\boxed{441}$

## A.11 SBSC PROMPT FOR AMC

You are given a math problem and you need to think of a stepwise approach/process to be followed to solve the problem. Use sympy-based python code to codify each of these steps in sequential manner. You must end each of your code snippet with a print statement followed by *output* in the next line. Use the results from the output of the code snippets of each step and continue to the next step until you finish the final step and solve the problem completely.
In each new code block, you must define the variables and symbols again based on the previous code snippets or their outputs as suitable.
Use sympy in appropriate way, as and when required.
Once you complete the last step, write this line: ### END OF CODE
Then just print the final numeric answer (without any units) like this: The final answer is boxed{}.
If the executed code snippet returns an error, use it to correct the current step's code snippet. DO NOT restart solving from Step 1.

Here are a few examples for your reference.

**Example Problem:** Small lights are hung on a string 6 inches apart in the order red, red, green, green, green, red, red, green, green, green, and so on continuing this pattern of 2 red lights followed

by 3 green lights. How many feet separate the 3rd red light and the 21st red light? Note: 1 foot is equal to 12 inches.

**Example Solution:** Let's solve this problem step by step using SymPy-based Python code.

Step 1: Define the pattern and calculate the number of lights in one complete pattern

```python
from sympy import symbols, Rational
# Define the number of red and green lights in one pattern
red_lights_per_pattern = 2
green_lights_per_pattern = 3
# Calculate total lights in one pattern
lights_per_pattern = red_lights_per_pattern + green_lights_per_pattern
print(f"Number of lights in one complete pattern: {lights_per_pattern}")

>>> output:
Number of lights in one complete pattern: 5
```

Let's continue with the next steps to solve this problem and leveraging the outputs of previous steps.
Step 2: Calculate the number of complete patterns between the 3rd and 21st red lights

```python
from sympy import symbols, Rational
# Define the number of red lights we're considering
start_red_light = 3
end_red_light = 21
# Calculate the number of red lights between the 3rd and 21st (inclusive)
red_lights_between = end_red_light - start_red_light + 1
# Calculate the number of complete patterns
complete_patterns = (red_lights_between - 1) // 2
print(f"Number of complete patterns between the 3rd and 21st red lights:
    {complete_patterns}")

>>> output:
Number of complete patterns between the 3rd and 21st red lights: 9
```

Let's continue with the next steps to solve this problem and leveraging the outputs of previous steps.
Step 3: Calculate the total number of lights between the 3rd and 21st red lights

```python
from sympy import symbols, Rational
# Define the number of red and green lights in one pattern
red_lights_per_pattern = 2
green_lights_per_pattern = 3
lights_per_pattern = red_lights_per_pattern + green_lights_per_pattern
# Number of complete patterns
complete_patterns = 9
# Calculate total lights
total_lights = complete_patterns * lights_per_pattern + 1  # +1 for the
    21st red light
print(f"Total number of lights between the 3rd and 21st red lights (
    inclusive): {total_lights}")

>>> output:
Total number of lights between the 3rd and 21st red lights (inclusive):
    46
```

Let's continue with the next steps to solve this problem and leveraging the outputs of previous steps.
Step 4: Calculate the total distance in inches

```python
from sympy import symbols, Rational
# Total number of lights
total_lights = 46
# Distance between lights in inches
distance_between_lights = 6
# Calculate total distance in inches
total_distance_inches = (total_lights - 1) * distance_between_lights
print(f"Total distance in inches: {total_distance_inches}")
```

```
>>> output:
Total distance in inches: 270
```

Let's continue with the next steps to solve this problem and leveraging the outputs of previous steps.
Step 5: Convert the distance from inches to feet

```python
from sympy import symbols, Rational
# Total distance in inches
total_distance_inches = 270
# Convert inches to feet
inches_per_foot = 12
total_distance_feet = Rational(total_distance_inches, inches_per_foot)
print(f"Total distance in feet: {total_distance_feet}")

>>> output:
Total distance in feet: 45/2
```

### END OF CODE

The final answer is $\boxed{\dfrac{45}{2}}$

**Example Problem:** A fruit salad consists of blueberries, raspberries, grapes, and cherries. The fruit salad has a total of 280 pieces of fruit. There are twice as many raspberries as blueberries, three times as many grapes as cherries, and four times as many cherries as raspberries. How many cherries are there in the fruit salad?

**Example Solution:** Let's solve this problem step by step using SymPy-based Python code.
Step 1: Define the variables and set up the equations

```python
from sympy import symbols, Eq, solve
# Define symbols for each type of fruit
b, r, g, c = symbols('b r g c')
# Set up equations based on the given information
eq1 = Eq(b + r + g + c, 280)  # Total pieces of fruit
eq2 = Eq(r, 2*b)  # Twice as many raspberries as blueberries
eq3 = Eq(g, 3*c)  # Three times as many grapes as cherries
eq4 = Eq(c, 4*r)  # Four times as many cherries as raspberries
print("Equations:")
print(eq1)
print(eq2)
print(eq3)
print(eq4)

>>> output:
Equations:
Eq(b + c + g + r, 280)
Eq(r, 2*b)
Eq(g, 3*c)
Eq(c, 4*r)
```

Let's continue with the next steps to solve this problem and leveraging the outputs of previous steps.
Step 2: Solve the system of equations

```python
from sympy import symbols, Eq, solve
# Define symbols for each type of fruit
b, r, g, c = symbols('b r g c')
# Set up equations based on the given information
eq1 = Eq(b + r + g + c, 280)  # Total pieces of fruit
eq2 = Eq(r, 2*b)  # Twice as many raspberries as blueberries
eq3 = Eq(g, 3*c)  # Three times as many grapes as cherries
eq4 = Eq(c, 4*r)  # Four times as many cherries as raspberries
# Solve the system of equations
solution = solve((eq1, eq2, eq3, eq4))
```

```
print("Solution:")
print(solution)

>>> output:
Solution:
{b: 8, c: 64, g: 192, r: 16}
```

Let's continue with the next steps to solve this problem and leveraging the outputs of previous steps.
Step 3: Extract the number of cherries from the solution

```
from sympy import symbols, Eq, solve
# Define symbols for each type of fruit
b, r, g, c = symbols('b r g c')
# Set up equations based on the given information
eq1 = Eq(b + r + g + c, 280)  # Total pieces of fruit
eq2 = Eq(r, 2*b)  # Twice as many raspberries as blueberries
eq3 = Eq(g, 3*c)  # Three times as many grapes as cherries
eq4 = Eq(c, 4*r)  # Four times as many cherries as raspberries
# Solve the system of equations
solution = solve((eq1, eq2, eq3, eq4))
# Extract the number of cherries
cherries = solution[c]
print(f"The number of cherries in the fruit salad is: {cherries}")

>>> output:
The number of cherries in the fruit salad is: 64
```

END OF CODE The final answer is $\boxed{64}$

**Example Problem:** Last summer $30\%$ of the birds living on Town Lake were geese, $25\%$ were swans, $10\%$ were herons, and $35\%$ were ducks. What percent of the birds that were not swans were geese?

**Example Solution:** Let's solve this problem step by step using SymPy-based Python code.

Step 1: Define the percentages of each type of bird

```
from sympy import symbols, Rational
geese = Rational(30, 100)
swans = Rational(25, 100)
herons = Rational(10, 100)
ducks = Rational(35, 100)
print(f"Percentage of geese: {geese}")
print(f"Percentage of swans: {swans}")
print(f"Percentage of herons: {herons}")
print(f"Percentage of ducks: {ducks}")

>>> output:
Percentage of geese: 3/10
Percentage of swans: 1/4
Percentage of herons: 1/10
Percentage of ducks: 7/20
```

Let's continue with the next steps to solve this problem. Step 2: Calculate the percentage of birds that are not swans

```
from sympy import symbols, Rational
geese = Rational(30, 100)
swans = Rational(25, 100)
herons = Rational(10, 100)
ducks = Rational(35, 100)
not_swans = 1 - swans
print(f"Percentage of birds that are not swans: {not_swans}")

>>> output:
Percentage of birds that are not swans: 3/4
```

Let's continue with the next steps to solve this problem and leveraging the outputs of previous steps.
Step 3: Calculate the percentage of geese among the birds that are not swans

```
from sympy import symbols, Rational
geese = Rational(30, 100)
swans = Rational(25, 100)
herons = Rational(10, 100)
ducks = Rational(35, 100)
not_swans = 1 - swans
geese_among_not_swans = geese / not_swans
print(f"Percentage of geese among birds that are not swans: {
    geese_among_not_swans}")

>>> output:
Percentage of geese among birds that are not swans: 2/5
```

Let's continue with the next steps to solve this problem and leveraging the outputs of previous steps.
Step 4: Convert the fraction to a percentage

```
from sympy import symbols, Rational
geese = Rational(30, 100)
swans = Rational(25, 100)
herons = Rational(10, 100)
ducks = Rational(35, 100)
not_swans = 1 - swans
geese_among_not_swans = geese / not_swans
percentage = geese_among_not_swans * 100
print(f"Percentage of geese among birds that are not swans: {percentage}%
    ")

>>> output:
Percentage of geese among birds that are not swans: 40%
```

### END OF CODE

The final answer is $\boxed{40}$

Example Problem: At a twins and triplets convention, there were 9 sets of twins and 6 sets of triplets, all from different families. Each twin shook hands with all the twins except his/her siblings and with half the triplets. Each triplet shook hands with all the triplets except his/her siblings and with half the twins. How many handshakes took place?

**Example Solution:** Let's solve this problem step by step using SymPy-based Python code.

Step 1: Define the given information

```
from sympy import symbols, Rational
# Number of sets of twins and triplets
twin_sets = 9
triplet_sets = 6
# Number of individuals
twins = twin_sets * 2
triplets = triplet_sets * 3
print(f"Number of twins: {twins}")
print(f"Number of triplets: {triplets}")

>>> output:
Number of twins: 18
Number of triplets: 18
```

Let's continue with the next steps to solve this problem and leveraging the outputs of previous steps.
Step 2: Calculate handshakes between twins

```
from sympy import symbols, Rational
# Number of sets of twins and triplets
twin_sets = 9
```

```
triplet_sets = 6
# Number of individuals
twins = twin_sets * 2
triplets = triplet_sets * 3
# Handshakes between twins
twin_handshakes = (twins * (twins - 2)) // 2  # Each twin shakes hands
    with all twins except self and sibling
print(f"Handshakes between twins: {twin_handshakes}")

>>> output
Handshakes between twins: 144
```

Let's continue with the next steps to solve this problem and leveraging the outputs of previous steps.
Step 3: Calculate handshakes between triplets

```
from sympy import symbols, Rational
# Number of sets of twins and triplets
twin_sets = 9
triplet_sets = 6
# Number of individuals
twins = twin_sets * 2
triplets = triplet_sets * 3
# Handshakes between twins
twin_handshakes = (twins * (twins - 2)) // 2
# Handshakes between triplets
triplet_handshakes = (triplets * (triplets - 3)) // 2  # Each triplet
    shakes hands with all triplets except self and two siblings
print(f"Handshakes between triplets: {triplet_handshakes}")

>>> output
Handshakes between triplets: 135
```

Let's continue with the next steps to solve this problem and leveraging the outputs of previous steps.
Step 4: Calculate handshakes between twins and triplets

```
from sympy import symbols, Rational
# Number of sets of twins and triplets
twin_sets = 9
triplet_sets = 6
# Number of individuals
twins = twin_sets * 2
triplets = triplet_sets * 3
# Handshakes between twins
twin_handshakes = (twins * (twins - 2)) // 2
# Handshakes between triplets
triplet_handshakes = (triplets * (triplets - 3)) // 2
# Handshakes between twins and triplets
twin_triplet_handshakes = twins * (triplets // 2)  # Each twin shakes
    hands with half the triplets
print(f"Handshakes between twins and triplets: {twin_triplet_handshakes}"
    )

>>> output:
Handshakes between twins and triplets: 162
```

Let's continue with the next steps to solve this problem and leveraging the outputs of previous steps.
Step 5: Calculate the total number of handshakes

```
from sympy import symbols, Rational
# Number of sets of twins and triplets
twin_sets = 9
triplet_sets = 6
# Number of individuals
twins = twin_sets * 2
```

```
triplets = triplet_sets * 3
# Handshakes between twins
twin_handshakes = (twins * (twins - 2)) // 2
# Handshakes between triplets
triplet_handshakes = (triplets * (triplets - 3)) // 2
# Handshakes between twins and triplets
twin_triplet_handshakes = twins * (triplets // 2)
# Total handshakes
total_handshakes = twin_handshakes + triplet_handshakes +
    twin_triplet_handshakes
print(f"Total number of handshakes: {total_handshakes}")

>>> output
Total number of handshakes: 441
```

### END OF CODE

The final answer is $\boxed{441}$

## A.12 SBSC PROMPT FOR AIME

You are given a math problem and you need to think of a stepwise approach/process to be followed to solve the problem. Use sympy-based python code to codify each of these steps in sequential manner. You must end each of your code snippet with a print statement followed by *output* in the next line. Use the results from the output of the code snippets of each step and continue to the next step until you finish the final step and solve the problem completely.
In each new code block, you must define the variables and symbols again based on the previous code snippets or their outputs as suitable.
Use sympy in appropriate way, as and when required.
Once you complete the last step, write this line: ### END OF CODE
Then just print the final numeric answer (without any units) like this: The final answer is boxed{}.
If the executed code snippet returns an error, use it to correct the current step's code snippet. DO NOT restart solving from Step 1.

Here are a few examples for your reference.

**Example Problem:** A frog begins at $P_0 = (0,0)$ and makes a sequence of jumps according to the following rule: from $P_n = (x_n, y_n)$, the frog jumps to $P_{n+1}$, which may be any of the points $(x_n + 7, y_n + 2)$, $(x_n + 2, y_n + 7)$, $(x_n - 5, y_n - 10)$, or $(x_n - 10, y_n - 5)$. There are $M$ points $(x, y)$ with $|x| + |y| \leq 100$ that can be reached by a sequence of such jumps. Find the remainder when $M$ is divided by 1000.

**Example Solution:** Let's approach this problem step by step using SymPy-based Python code.

Step 1: Import necessary libraries and define the jump rules

```
from sympy import symbols, Abs
import numpy as np
# Define the possible jumps from any position (x, y)
def jumps(x, y):
    return [
        (x + 7, y + 2),    # Jump by (7, 2)
        (x + 2, y + 7),    # Jump by (2, 7)
        (x - 5, y - 10),   # Jump by (-5, -10)
        (x - 10, y - 5)    # Jump by (-10, -5)
    ]
print("Jump rules defined.")

>>> output
Jump rules defined.
```

Let's continue with the next steps to solve this problem and leveraging the outputs of previous steps.
Step 2: Implement a breadth-first search to find all reachable points

```
from sympy import symbols, Abs
import numpy as np
from collections import deque
# Reuse the jumps function to define possible moves
def jumps(x, y):
    return [
        (x + 7, y + 2),
        (x + 2, y + 7),
        (x - 5, y - 10),
        (x - 10, y - 5)
    ]
# Define the breadth-first search (BFS) function
def bfs():
    queue = deque([(0, 0)])  # Start BFS from the point (0, 0)
    visited = set([(0, 0)])  # Keep track of visited points to avoid
    reprocessing
    # Process each point in the queue
    while queue:
        x, y = queue.popleft()
        for nx, ny in jumps(x, y):  # Try all possible jumps from the
    current point
            if (nx, ny) not in visited and abs(nx) + abs(ny) <= 100:
                visited.add((nx, ny))  # Mark the new point as visited
                queue.append((nx, ny))  # Add the new point to the queue
    to explore further
    return len(visited)  y
M = bfs()
print(f"Number of reachable points (M): {M}")

>>> output:
Number of reachable points (M): 1373
```

Let's continue with the final step to solve this problem. Step 3: Calculate the remainder when M is divided by 1000

```
from sympy import symbols, Abs
import numpy as np
M = 1373  # M_value from previous step
# Find the remainder when M is divided by 1000
remainder = M % 1000
print(f"Remainder when M is divided by 1000: {remainder}")

>>> output
Remainder when M is divided by 1000: 373
```

### END OF CODE

The final answer is $\boxed{373}$

**Example Problem:** The AIME Triathlon consists of a half-mile swim, a 30-mile bicycle ride, and an eight-mile run. Tom swims, bicycles, and runs at constant rates. He runs fives times as fast as he swims, and he bicycles twice as fast as he runs. Tom completes the AIME Triathlon in four and a quarter hours. How many minutes does he spend bicycling?

**Example Solution:** Let's solve this problem step by step using SymPy-based Python code.

Step 1: Define the variables and set up the basic equations

```
from sympy import symbols, Eq, solve
# Define variables
s, b, r = symbols('s b r')  # s: swim speed, b: bike speed, r: run speed
# Set up equations based on the given information
eq1 = Eq(r, 5*s)  # Tom runs 5 times as fast as he swims
eq2 = Eq(b, 2*r)  # Tom bicycles twice as fast as he runs
print("Equation 1:", eq1)
print("Equation 2:", eq2)
```

```
>>> output
Equation 1: Eq(r, 5*s)
Equation 2: Eq(b, 2*r)
```

Let's continue with the next steps to solve this problem and leveraging the outputs of previous steps.
Step 2: Set up the equation for the total time

```
from sympy import symbols, Eq, solve
# Reuse earlier code
s, b, r = symbols('s b r')
eq1 = Eq(r, 5*s)
eq2 = Eq(b, 2*r)
# Total time equation (in hours)
total_time = 0.5/s + 30/b + 8/r  #The AIME Triathlon consists of a half-
    mile swim, a 30-mile bicycle ride, and an eight-mile run.
eq3 = Eq(total_time, 4.25)  # Total time is 4.25 hours
print("Total time equation:", eq3)

>>> output:
Total time equation: Eq(0.5/s + 8/r + 30/b, 4.25)
```

Let's continue with the next steps to solve this problem and leveraging the outputs of previous steps.
Step 3: Substitute the relationships between speeds into the total time equation

```
from sympy import symbols, Eq, solve, simplify
# Resue earlier code
s, b, r = symbols('s b r')
eq1 = Eq(r, 5*s)
eq2 = Eq(b, 2*r)
total_time = 0.5/s + 30/b + 8/r
eq3 = Eq(total_time, 4.25)  # Total time is 4.25 hours
# Substitute r and b in terms of s to simplify the equation
eq3_substituted = eq3.subs([(r, 5*s), (b, 10*s)])
eq3_simplified = simplify(eq3_substituted)
print("Simplified equation:", eq3_simplified)

>>> output:
Simplified equation: Eq(5.1/s, 4.25)
```

Let's continue with the next steps to solve this problem and leveraging the outputs of previous steps.
Step 4: Solve for the swim speed (s)

```
from sympy import symbols, Eq, solve, simplify
# Redefine variables
s, b, r = symbols('s b r')  # s: swim speed, b: bike speed, r: run speed
# Simplified equation from previous step
eq3_simplified = Eq(5.1/s, 4.25)
# Solve for s
s_solution = solve(eq3_simplified, s)
print("Swim speed (s):", s_solution[0])

>>> output
Swim speed (s): 1.20000000000000
```

Let's continue with the next steps to solve this problem and leveraging the outputs of previous steps.
Step 5: Calculate the bike speed (b)

```
from sympy import symbols, Eq, solve, simplify
# Redefine variables
s, b, r = symbols('s b r')  # s: swim speed, b: bike speed, r: run speed
# Swim speed from previous step
s_value = 1.20000000000000
# Calculate bike speed
```

```
b_value = 10 * s_value  # b = 2r = 2(5s) = 10s
print("Bike speed (b):", b_value)

>>> output
Bike speed (b): 12.0
```

Let's continue with the next steps to solve this problem and leveraging the outputs of previous steps.
Step 6: Calculate the time spent bicycling

```
from sympy import symbols, Eq, solve, simplify
# Redefine variables
s, b, r = symbols('s b r')  # s: swim speed, b: bike speed, r: run speed
# Bike speed from previous step
b_value = 12.0
# Calculate time spent bicycling
bike_distance = 30  # miles
bike_time = bike_distance / b_value  # hours
# Convert to minutes
bike_time_minutes = bike_time * 60
print("Time spent bicycling (in minutes):", bike_time_minutes)

>>> output
Time spent bicycling (in minutes): 150.0
```

### END OF CODE

The final answer is $\boxed{150}$

**Example Problem:** Let $S$ be the increasing sequence of positive integers whose binary representation has exactly 8 ones. Let $N$ be the 1000th number in $S$. Find the remainder when $N$ is divided by 1000

**Example Solution:** Let's solve this problem step by step using SymPy-based Python code.

Step 1: Import necessary libraries and define a function to count ones in binary representation

```
from sympy import symbols, binomial
import math
# Function to count the number of '1's in the binary representation of a
    number
def count_ones(n):
    return bin(n).count('1')  # Convert n to binary and count occurrences
     of '1'
print("Function to count ones in binary representation defined.")

>>> output:
Function to count ones in binary representation defined.
```

Let's continue with the next steps to solve this problem and leveraging the outputs of previous steps.
Step 2: Find the smallest number in the sequence S

```
from sympy import symbols, binomial
import math
# Reuse the function to count the number of '1's in binary
def count_ones(n):
    return bin(n).count('1')
# The smallest number in the sequence S is represented by 8 consecutive
    '1's in binary
smallest_number = int('1' * 8, 2)  # Convert '11111111' (binary) to
    decimal
print(f"The smallest number in sequence S is: {smallest_number}")

>>> output
The smallest number in sequence S is: 255
```

Let's continue with the next steps to solve this problem and leveraging the outputs of previous steps.
Step 3: Find the largest number in the sequence S

```python
from sympy import symbols, binomial
import math
# Reuse the function to count the number of '1's in binary
def count_ones(n):
    return bin(n).count('1')
smallest_number = int('1' * 8, 2)
# The largest number in sequence S is represented by 8 consecutive '1's
    followed by 24 '0's
largest_number = int('1' * 8 + '0' * 24, 2)  # Convert
    '11111111000000000000000000000000' to decimal
print(f"The largest number in sequence S is: {largest_number}")

>>> output
The largest number in sequence S is: 4278190080
```

Let's continue with the next steps to solve this problem and leveraging the outputs of previous steps.
Step 4: Calculate the total number of elements in sequence S

```python
from sympy import symbols, binomial
import math
def count_ones(n):
    return bin(n).count('1')
smallest_number = int('1' * 8, 2)
largest_number = int('1' * 8 + '0' * 24, 2)
# Calculate the total number of elements in sequence S using binomial(32,
    8)
# This represents the number of ways to choose 8 positions out of 32 bits
    to set to '1'
total_elements = binomial(32, 8)
print(f"The total number of elements in sequence S is: {total_elements}")

>>> output
The total number of elements in sequence S is: 10518300
```

Let's continue with the next steps to solve this problem and leveraging the outputs of previous steps.
Step 5: Find the 1000th number in sequence S

```python
from sympy import symbols, binomial
import math
#Reuse all earlier code
def count_ones(n):
    return bin(n).count('1')
smallest_number = int('1' * 8, 2)
largest_number = int('1' * 8 + '0' * 24, 2)
total_elements = binomial(32, 8)
# Find the 1000th number in the sequence
count = 0  # Initialize a counter for the sequence
N = smallest_number  # Start from the smallest number in S
# Iterate through numbers and check for 8 '1's in the binary
    representation
while count < 1000:
    if count_ones(N) == 8:  # If the current number has exactly 8 '1's
        count += 1  # Increment the count
    if count == 1000:  # Stop if we've found the 1000th number
        break
    N += 1  # Move to the next number
print(f"The 1000th number (N) in sequence S is: {N}")

>>> output
The 1000th number (N) in sequence S is: 7032
```

Let's continue with the next steps to solve this problem and leveraging the outputs of previous steps.
Step 6: Calculate the remainder when N is divided by 1000

```python
from sympy import symbols, binomial
```

```
import math
def count_ones(n):
    return bin(n).count('1')
N = 7032  # From the previous step
# Calculate the remainder when N is divided by 1000
remainder = N % 1000
print(f"The remainder when N is divided by 1000 is: {remainder}")

>>> output
The remainder when N is divided by 1000 is: 32
```

### END OF CODE

The final answer is $\boxed{32}$

**Example Problem:** Two geometric sequences $a_1, a_2, a_3, \ldots$ and $b_1, b_2, b_3, \ldots$ have the same common ratio, with $a_1 = 27$ $b_1 = 99$ , and $a_{15} = b_{11}$ . Find $a_9$

**Example Solution:** Let's solve this problem step by step using SymPy-based Python code.

Step 1: Define the variables and set up the initial conditions

```
from sympy import symbols, Eq, solve
# Define symbols
r = symbols('r')   # common ratio
a1, b1 = 27, 99    # first terms of sequences a and b
print(f"a1 = {a1}, b1 = {b1}")
print("Common ratio is represented by r")

>>> output
a1 = 27, b1 = 99
Common ratio is represented by r
```

Let's continue with the next steps to solve this problem and leveraging the outputs of previous steps.
Step 2: Set up the equation for a15 = b11

```
from sympy import symbols, Eq, solve
# Redefine symbols
r = symbols('r')
a1, b1 = 27, 99
# Set up the equation a15 = b11
eq = Eq(a1 * r**14, b1 * r**10)
print("Equation: a15 = b11")
print(f"Symbolically: {eq}")

>>> output
Equation: a15 = b11
Symbolically: Eq(27*r**14, 99*r**10)
```

Let's continue with the next steps to solve this problem and leveraging the outputs of previous steps.
Step 3: Solve the equation for r

```
from sympy import symbols, Eq, solve
# Redefine symbols
r = symbols('r')
a1, b1 = 27, 99
# Set up the equation a15 = b11
eq = Eq(a1 * r**14, b1 * r**10)
# Solve for r
solution = solve(eq, r)
print("Solutions for r:")
print(solution)

>>> output
Solutions for r:
[0, -11**(1/4)*3**(3/4)/3, 11**(1/4)*3**(3/4)/3]
```

Let's continue with the next steps to solve this problem and leveraging the outputs of previous steps.
Step 4: Identify the correct value for r

```python
from sympy import symbols, Eq, solve, N
# Reuse earlier code
r = symbols('r')
a1, b1 = 27, 99
eq = Eq(a1 * r**14, b1 * r**10)
solution = solve(eq, r)
# The correct value for r is the positive real solution
r_value = max(sol for sol in solution if sol.is_real)
print(f"The common ratio r is: {N(r_value, 6)}")

>>> output
The common ratio r is: 1.38378
```

Let's continue with the next steps to solve this problem and leveraging the outputs of previous steps.
Step 5: Calculate a9

```python
from sympy import symbols, Eq, solve, N
# Reuse earlier code
r = symbols('r')
a1, b1 = 27, 99
eq = Eq(a1 * r**14, b1 * r**10)
solution = solve(eq, r)
r_value = max(sol for sol in solution if sol.is_real)
# Calculate a9
a9 = a1 * r_value**8
print(f"a9 = {N(a9, 10)}")

>>> output
a9 = 363.0000000
```

### END OF CODE

The final answer is $\boxed{363}$

### A.13 RESULTS WITH STANDARD DEVIATIONS

We present the main table results along with standard deviation scores in Table 17.

Table 17: Benchmarking SBSC against different math reasoning methods across three datasets. We report average accuracy over 3 runs with standard deviation within parentheses. Best result in each setting is highlighted in **bold** and second best is underlined. Absolute improvement in performance by SBSC over the previous best method in each setting is indicated in subscript.

| Method | AMC | | AIME | | MathOdyssey | | Olympiad Bench | |
|---|---|---|---|---|---|---|---|---|
| | greedy | maj@7 | greedy | maj@7 | greedy | maj@7 | greedy | maj@7 |
| Claude-3.5-Sonnet | | | | | | | | |
| COT | 31.16 ($\pm$1.0) | 35.79 | 9.09 ($\pm$1.0) | 10.91 | 11.89 ($\pm$0.6) | 16.89 | 39.35 ($\pm$0.47) | 42.46 |
| PAL | 35.79 ($\pm$1.0) | 36.42 | 27.48 ($\pm$0.6) | 28.79 | 27.23 ($\pm$0.6) | 31.01 | 41.07 ($\pm$0.82) | 44.44 |
| TIR-ToRA | 38.59 ($\pm$0.6) | 43.16 | 24.64 ($\pm$3.2) | 26.67 | 27.23 ($\pm$0.6) | 32.43 | 47.69 ($\pm$0.47) | 50.60 |
| SBSC (Ours) | 49.33 ($\pm$3.1)$_{\uparrow10.7}$ | -$_{\uparrow6.2}$ | 35.45 ($\pm$1.7)$_{\uparrow8}$ | -$_{\uparrow6.7}$ | 39.86 ($\pm$1.0)$_{\uparrow12.6}$ | -$_{\uparrow7.4}$ | 53.31 ($\pm$0.94)$_{\uparrow5.6}$ | -$_{\uparrow2.7}$ |
| GPT-4o | | | | | | | | |
| COT | 35.94 ($\pm$0.6) | 37.47 | 10.39 ($\pm$2.1) | 12.12 | 13.51 ($\pm$1.0) | 17.57 | 41.80 ($\pm$1.89) | 47.22 |
| PAL | 36.48 ($\pm$0.6) | 38.11 | 24.63 ($\pm$0.6) | 26.97 | 15.74 ($\pm$0.6) | 20.27 | 41.67 ($\pm$2.16) | 46.43 |
| TIR-ToRA | 37.33 ($\pm$2.5) | 40.42 | 22.42 ($\pm$1.7) | 25.45 | 19.59 ($\pm$2.6) | 23.64 | 43.32 ($\pm$1.70) | 49.61 |
| SBSC (Ours) | 44.55 ($\pm$0.6)$_{\uparrow7.2}$ | -$_{\uparrow4.1}$ | 30.7 ($\pm$1.1)$_{\uparrow6.1}$ | -$_{\uparrow3.7}$ | 26.55 ($\pm$1.1)$_{\uparrow7}$ | -$_{\uparrow2.9}$ | 48.74 ($\pm$1.89))$_{\uparrow5.4}$ | -$_{\downarrow0.87}$ |

### A.14 LEAST-TO-MOST PROMPTING

Least-to-Most (L2M) (Zhou et al., 2022) is a two-stage prompting strategy where the aim is: in first stage, to break down a complex problem into a series of simpler subproblems and then, in second stage, solve these predefined subproblems. PAL (Gao et al., 2022) reported a L2M version

of PAL in their work. We follow the reported prompts and replicate it by designing exemplars for both the stages. We find L2M-PAL inherits the same issues that PAL & TIR-TORA has. L2M-PAL comes up with entire sub-problems at once and also its uses single program-block to solve those sub-problems. SBSC dynamically generates the next sub-task and the corresponding program to solve it leveraging the previous turns results. In Table 18, we show the results obtained from L2M + PAL using Claude-3.5-Sonnet on our AMC and AIME test datasets. Even after allowing self-correction for stage 2 with max turns `n=15`, L2M-PAL approaches PAL scores. Hence for our main results, we stick to PAL & TIR-ToRA along with self-consistency (Shao et al., 2024) due to resource optimisation and wide adaption of those prompting strategies for math-problem solving.

Table 18: Least-to-Most Prompting results on AIME and AMC

| Method | AMC | | AIME | |
|---|---|---|---|---|
| | greedy | maj@7 | greedy | maj@7 |
| COT | 31.16 | 35.79 | 9.09 | 10.91 |
| PAL | 35.79 | 36.42 | 27.48 | 28.79 |
| L2M-PAL (n=1) | 33.47 | 38.53 | 25.45 | 28.79 |
| L2M-PAL (n=15) | 34.32 | | 25.45 | |
| TIR-ToRA | 38.59 | 43.16 | 24.64 | 26.67 |
| SBSC (Ours) | **49.33**$_{\uparrow 10.7}$ | $-_{\uparrow 6.2}$ | **35.45**$_{\uparrow 8}$ | $-_{\uparrow 6.7}$ |

