# OpenReview forum: "SBSC: Step-by-Step Coding for Improving Mathematical Olympiad Performance"
_ICLR.cc/2025/Conference — ICLR 2025 Poster_

### Official Review · Reviewer_uQkW · 2024-10-20

**Soundness:** 4
**Presentation:** 4
**Contribution:** 4
**Rating:** 8
**Confidence:** 4

**Summary:**

This paper introduces Step-by-Step Coding (SBSC), a multi-step reasoning framework where LLMs are prompted to break down complex questions into sub-questions and solve them iteratively with the help of generated code. The experiments demonstrate that SBSC achieves higher accuracy compared to Chain-of-Thought (CoT) and Program-Aided Language (PAL) on competition and Olympiad-level math problems, regardless of problem topics or code errors. The authors also show that self-consistency further improves SBSC's performance.

**Strengths:**

1. The concept of step-by-step, multi-turn coding is both intuitive and innovative. Unlike previous approaches that focus mainly on validation after the final generation or self-debugging, SBSC’s use of intermediate results from an executor provides more actionable and informative feedback. This method effectively combines task decomposition from CoT with the program-assisted capabilities of PAL, offering a promising approach.

2. The experiments on Olympiad-level math problems clearly show SBSC’s effectiveness, outperforming the baseline by a large margin.

**Weaknesses:**

1. While the idea has potential for broader applications, the authors only tested it on Olympiad-level math problems. Additionally, while SBSC appears promising, is it appropriate to apply this method to such difficult problems? (See more detailed discussion in question 1.)

2. The baseline comparisons could be more comprehensive. Although the comparison with CoT and PAL is sufficient to demonstrate SBSC's effectiveness, it does not fully showcase its strengths or weaknesses relative to other complementary methods, such as multi-turn program validation, self-debugging, the use of additional tools, or expert models. Even if such comparisons might be unfavorable to SBSC, they would provide valuable context for understanding the broader landscape and gauging SBSC's relative contribution.

**Questions:**

1. The mainly question is whether using code to solve oplymipcs-level math problem is a reasonable approach. Both baselines PaL and ToRA are not designed for such difficult problems, which often require more than basic operations or listing equations, such as some extra theorems cannot be solved by code. In Figure 1, there are several points where the LLM's use of Python code seems unnatural. First, the code is strongly based on some black-box high-level function provided by some libraries, like `summation` in `sympy` library to get a lemma "$1^3+\dots+n^3=\frac{n^4}{4}+\frac{n^3}{2}+\frac{n^2}{4}$". It is not a natural usage of codes (at least Python codes) because for more challenging tasks, the LLM could not find a propriate library provides the information. Similarly, the program's brute-force traversal from $n=1$ to $1000$ is not a typical way to solve such problems. A more natural way would involve algebraic manipulation like $$ \frac{n^4}{4}+\frac{n^3}{2}+\frac{n^2}{4}\equiv17(\mod n+5)$$ $$ n^4+2\times n^3 + n^2\equiv68(\mod n+5) $$ $$ (-5)^4+2\times(-5)^3+(-5)^2\equiv 68 (\mod n+5) $$ $$ 400 \equiv 68 (\mod n+5) $$ so that $(n+5)\mid 332 \Rightarrow n +5 =83 \text{ or } 166 \text{ or } 332 \Rightarrow n = 78, 161, 327$ then we final check these candidates and get the final answer $n=78\text{ and }161$. The process shown in Figure 1 shows that using Python code is not a natural and reasonable way to solve these problems. The authors should further compare SCSC's approach to human-like reasoning on a subset of problems or some examples, or to analyze which types of Olympiad problems are most/least suited to SBSC's code-based approach.

2. While step-by-step reasoning is a natural approach, writing and displaying code in steps is not common practice. Programmers often organize their code by defining classes and functions first, then calling them in a main function. How can we ensure that LLMs are capable of generating effective step-by-step code? I believe that the authors should add some experiments to support current LLMs have this ability. Additionally, some steps in the reasoning process may not correspond to clear intermediate results in code. For instance, what happens if a step only defines a function with no immediate output to display? I'm curious whether this decomposition approach remains effective across different problem types.

3. Does the method's effectiveness depend on delicate prompt engineering or careful selection of few-shot examples? Although the experiments in Section 5.1 suggest the method is not overly sensitive to the few-shot candidate selection, I would like more clarity on the details and principles behind crafting these 10 examples. When generalizing the method to other tasks, how should the steps be divided to ensure each step yields clear, intermediate results that aid subsequent reasoning?

2. Minor questions:
    1. The authors say "similar to NuminaMath, we remove all the answer choices from each AMC-12 question and modify the question to ensure an integer answer". But I cannot find evidence that the NuminaMath modifies questions to ensure an integer answer. Rather, it is a preprocess in original AIME dataset to keep answer as integer and Numina focuses on this problem only for decontamination. The authors should clarify this point.
    2. In Figure 4, how is the x-axis "error steps per problem" determined? A more detailed explanation of this experiment would be helpful.

I would raise my rating to 8 or higher if the authors can address my two main concerns (questions 1 and 2).

---

> ### Author Response · Authors · 2024-11-21
> **Author Response to Official Review by Reviewer uQkW (1/N)**
>
> We thank the reviewer for the comprehensive review. We appreciate the opportunity to clarify the points raised & to provide additional insights into our research.
> > [Weaknesses] 1.1	While the idea has potential for broader applications, the authors only tested it on Olympiad-level math problems. Additionally, while SBSC appears promising, is it appropriate to apply this method to such difficult problems?
>
> > [Questions 1] 1.2 The main question is whether using code to solve olympiad-level math problem is a reasonable approach. Both baselines PAL and ToRA are not designed for such difficult problems, which often require more than basic operations or listing equations, such as some extra theorems cannot be solved by code.
>
> We would like to address reviewer’s questions using the following points that establishes SBSC’s premise in-context to the scope of this paper:
> **(Answers 1.1 and 1.2) Shift to harder benchmarks in AI4MATH Reasoning Field:** Initially, mathematical reasoning performance of LLMs & reasoning methods [COT, Self Consistency & PAL] benchmarking was limited to GSM8K (grade school level problems). While GSM8K was starting to get saturated, MATH (competition level problems) was widely adopted. TIR-ToRA paper studied its performance (along with COT, PAL) on MATH dataset. Notably, some of the level 4 & 5 problems of MATH dataset are older problems from AIME & AMC. Over time, proprietary and leading LLMs have achieved remarkable performance on 90% +GSM8K & 70%+ on MATH [1,2]. However, as we have mentioned in the Motivation section, the emergence of recent, more difficult benchmarks like OlympiadBench, MathOdyssey, AIME, AMC etc show that both generalist/math-specialist SOTA models & methods continue to struggle with advanced math reasoning. Hence, we have focused on developing a method suitable to tackle such competition-level problems. *Doing comparative study with the previous, relevant & widely adopted reasoning methods such as COT, PAL, TIR-ToRA is important to understand the improvement of SBSC especially on the benchmarks where results are not saturated, and we can measure progress.*
>
> **(Answers 1.2) TIR-ToRA being the SOTA reasoning strategy (ahead of COT) on newer/tougher/olympiad-level benchmarks**:  TIR-ToRA (code/program generation based) has lately been the SOTA math reasoning strategy showing improvements over PAL & COT which is also evident from Table 1. In terms of accuracy, program generation-based approaches have proven to be better than COT across the competition level benchmarks. Even the current SOTA math LLMs such as Numina-MATH, DeepSeekMath have primarily tried to learn TIR-ToRA reasoning during instruct fine-tuning. It has helped them achieve SOTA scores on the tougher/similar benchmarks such as the one we have used. *The TIR instruction fine-tuned variants have consistently scored more than the COT instruction fine-tuned variants on these tougher benchmarks.* In fact, Numina-MATH was the winner of the first AIMO competition [3] as well and leveraged TIR-ToRA strategy. However, the scores on these benchmarks even with TIR-ToRA were still low with significant room for improvement. *Given how impactful TIR-ToRA has been, we wanted to push the boundaries of program-generation based reasoning further to improve scores on tougher benchmarks.*
>
> **(Answer to 1.2) SBSC is the next generation of program-generation based reasoning, improving over TIR-ToRA and PAL, measured over competition level problems:** We wanted to create a new advanced math reasoning strategy that could be more dynamic, granular & yet leveraged program generation. SBSC improves over TIR-ToRA because of this & hence can not only better self-correct but also sustain reasoning over multiple intermediate steps. SBSC generates the sub-task dynamically and uses program generation solve/conclude each of the intermediate step due to the multi-turn nature. As mentioned in “Future Work” section, we believe that the intermediate states/answers generated during SBSC reasoning format could be highly useful, which lacked in previous methods such as TIR-ToRA & PAL, for guided decoding strategies such as in Outcome-Supervised Value Model  [4], AlphaMATH [5].
>
> **(Answer 1.2) Type of math problems currently included in the scope of the paper:** In alignment to the previous three points of SBSC’s premise, we had to benchmark on relevant datasets and these datasets usually included problems that have a unique numerical answer & were not proof based.
>
> [1] Introducing Claude 3.5 Sonnet, https://www.anthropic.com/news/claude-3-5-sonnet.
> [2] GPT-4 Technical Report, arXiv:2303.08774
> [3] AI Mathematical Olympiad - Progress Prize 1, https://www.kaggle.com/competitions/ai-mathematical-olympiad-prize.
> [4] Yu et al., “Outcome-supervised value models for planning in mathematical reasoning”    arXiv:2311.09724.
> [5] Chen et al., “Alphamath almost zero: process supervision without process”    arXiv:2405.03553.

---

> > ### Comment · Reviewer_uQkW · 2024-11-21
> >
> > I appreciate that the authors explain their motivation from the perspective of the historical development in this field. However, the issue lies in the potential essential difference based on problem's difficulty. It may be reasonable to use programs (and some pre-written libraries) for solving GSM8k-level math problems, as these problems do not involve much novelty and exploration. But when the problem difficulty reaches the competition level, its nature may fundamentally change.
> > I believe this paper offers a good attempt at advancing program-assisted mathematical problem-solving. It's OK so I rate it as 6. However, if the authors aim to demonstrate that this approach holds promise for solving the most challenging mathematical problems—especially those where, as the authors note, even the best current models struggle to deliver satisfactory results—then it is essential to convince readers that this approach (i.e., program-assisted reasoning) is a viable path forward. This cannot be justified solely by stating that ToRA achieves SOTA, because the limitations of CoT may simply stem from the fact that "good CoT" (likely requiring annotations by experts highly familiar with mathematics) has not yet been sufficiently learned by large models.

---

> ### Author Response · Authors · 2024-11-21
> **Author Response to Official Review by Reviewer uQkW (2/N)**
>
> > [Questions 1] 1.2	The main question is whether using code to solve olympiad-level math problem is a reasonable approach. Both baselines PAL and ToRA are not designed for such difficult problems, which often require more than basic operations or listing equations, such as some extra theorems cannot be solved by code.
>
>
> **(Continued answer to  1.2) On theorem proving and proof-based questions:** Proof based questions are often found in even more challenging competitions like International Mathematical Olympiad , United States of America Mathematical Olympiad and generally, it is being explored as separate line of research specifically in the form of formalized theorem proving. Lately, there have been multiple developments in leveraging LLMs for automated theorem proving [1, 2]. *Interestingly, automated theorem proving is also utilizing program generation based like paradigm.* A formal theorem-proving model/method must understand the syntax and semantics of formal languages like LEAN and bridge abstract mathematical reasoning with exact formal representations. Given that formal proofs are essentially programs/codes (which are not executed but more of verified), we believe SBSC could potentially even leverage LEAN for intermediate turns to unlock further improvements or can be adopted for formal theorem proving as formal proofs require generating intermediate tactics. Hence, this was one more reason that we wanted to explore theorem proving or proof-based questions as a separate effort.
>
>
> [1] Kumarappan et al., “LeanAgent: Lifelong Learning for Formal Theorem Proving”, 	arXiv:2410.06209.
> [2] Xin et al., “DeepSeek-Prover: Advancing Theorem Proving in LLMs through Large-Scale Synthetic Data”, arXiv:2405.14333

---

> ### Author Response · Authors · 2024-11-21
> **Author Response to Official Review by Reviewer uQkW (3/N)**
>
> > [Questions 1]  In Figure 1, there are several points where the LLM's use of Python code seems unnatural. The process shown in Figure 1 shows that using Python code is not a natural and reasonable way to solve these problems. The authors should further compare SCSC's approach to human-like reasoning on a subset of problems or some examples, or to analyze which types of Olympiad problems are most/least suited to SBSC's code-based approach.
>
>
> We would like to address these questions using the following points:
> 	**Pre-SBSC, program generation-based approaches such as TIR-ToRA have outperformed (in terms of accuracy) COT on newer and tougher/Olympiad like benchmarks:** We had already established the context for this in our previous reply and how this shaped the motivation and scope of SBSC and our paper.
> Consistency in computational library/function usage among PAL, TIR-ToRA and SBSC: As mentioned in the previous method, given how impactful TIR-ToRA has been, we wanted to push the boundaries of program-generation based reasoning further. SBSC is the next generation of program-generation based reasoning, improving over TIR-ToRA and PAL. Given PAL and TIR-ToRA already leveraged these computational libraries and functions like Sympy, SBSC was also expected to leverage them for fair comparison.
>
> **Study on Sympy usage between SBSC and TIR-ToRA:**
> We conduct a study to understand the usage of sympy library by SBSC and TIR-ToRA for last 10 years of AIME and AMC questions, shown in the table below. Specifically, we present the number of questions in which these methods utilized that library, and we present the accuracy for those set of questions. The number of questions is which both the methods utilized Sympy are similar. For both AIME and AMC problems from the last 10 years, SBSC based solutions have used Sympy only on 7 more questions in both test-sets. But when we look at the accuracy (among these filtered questions), we can observe that SBSC had 10%+ absolute accuracy lead in both the dataset.
> | Context   | AIME (#Qs) | AIME (Accuracy %) | AMC (#Qs) | AMC (Accuracy %) |
> |-----------|------------|-------------------|-----------|------------------|
> | TIR-ToRA  | 200        | 15.50%           | 299       | 36.79%          |
> | SBSC      | 207        | 30.43%           | 306       | 46.73%          |
>
>
> Also, we further filtered the common questions, for both the test-sets, where both TIR-ToRA and SBSC made use of Sympy. We present this study in the table just below. We observed that 80+% questions are in common for both the methods across both the test-sets. And when we observe the accuracy of the methods over these common questions, SBSC has 10% absolute improvement across both the test-sets.
> | Dataset | # Common Qs with Sympy Usage  | TIR-ToRA (Accuracy %) | SBSC (Accuracy %) |
> |---------|------------------------------|--------------|----------|
> | AIME    | 163                          | 15.95        | **28.22**    |
> | AMC     | 252                          | 38.49        | **47.62**    |
>
>
> **Topic wise analysis of SBSC (unnatural as program generation based) vs COT (human like):**
> While the goal of SBSC was to introduce a SOTA multi-turn-based program generation method, we acknowledge that the reasoning process followed by tool-integrated reasoning methods including our SBSC as well as existing works in this domain like TIR-ToRA may be different from that of COT, which follows a reasoning structure like how a human math expert would do.
>
> However, in terms of accuracy, we do observe that like other program-generation based approaches, SBSC has also significantly outperformed COT, not only across both AIME and AMC test-sets, but even across every individual topic. Here COT represents the ability of LLMs to generate human like reasoning. As suggested by the reviewer, we have presented the topic wise analysis for various methods across all the datasets.
>
>
> | Dataset         | Topic                  | TIR-ToRA (%) | SBSC (%) | COT (%)  |
> |-----------------|------------------------|--------------|----------|----------|
> | AIME            | Algebra                | 23.7         | **44.7**     | 10.53    |
> |                 | Arithmetic             | **80.0**         | **80.0**     | 40.00    |
> |                 | Counting & Probability | 31.4         | **46.5**     | 10.47    |
> |                 | Geometry               | 1.8          | **7.3**      | 4.55     |
> |                 | Number Theory          | 54.7         | **62.3**     | 13.21    |
> | AMC             | Algebra                | 47.3         | **50.9**     | 33.53    |
> |                 | Arithmetic             | 61.5         | **74.4**     | 64.10    |
> |                 | Counting & Probability | 34.8         | **55.4**     | 23.91    |
> |                 | Geometry               | 17.8         | **31.0**     | 19.38    |
> |                 | Number Theory          | 52.1         | **62.5**     | 41.67    |

---

> > ### Comment · Reviewer_uQkW · 2024-11-21
> > **Reply for 3/N**
> >
> > As I mentioned above, when considering whether program-assisted approaches are reasonable for competition-level math problems, I believe a comparison with ToRA cannot fully address the question—since both rely on program assistance. However, I find the second tables presented to be somewhat convincing, though I worry that CoT may have been underestimated here, especially given the recent emergence of o1, which could significantly improve these numbers. Notably, OpenAI o1 adopts an approach more akin to CoT rather than program assistance. Of course, since o1 was only recently introduced and is still limited in availability, I don’t think it is necessary for the authors to consider it at this stage. However, I still hope to see related results in future versions.

---

> > > ### Author Response · Authors · 2024-11-30
> > > **A study to further understand the potential of SBSC and comparison with o1 (1/N)**
> > >
> > > We would like to thank the reviewer for the prompt response. We also appreciate the points mentioned by the reviewer, specifically, if SBSC could enable solving more challenging questions and how does it compare with new SOTA COT reasoning model like o1. We have conducted a detailed study in an attempt to address this.
> > >
> > > Before we discuss the study we would like to highlight that *for frontier LLMs such as Claude 3.5 Sonnet, we have already showed that SBSC surpasses not only greedy-decoding but also self-consistency decoding results of TIR-ToRA, PAL & COT across 6 challenging benchmarks such as AIME, AMC, MathOdyssesy, OlympiadBench and we also showed it on JEE-Bench & Omni-Math during rebuttal [Author Response to Official Review by Reviewer EW8i (4/N)]. These benchmarks are the most challenging benchmarks for math reasoning and significantly tougher than GSM8K. We also show topic wise analysis as well to show the improvement is across the topics.*
> > >
> > > **SBSC achieves SOTA on olympiad benchmarks but can SBSC score higher ?:** On last 11 years of AIME questions, despite being SOTA SBSC achieves ~36% accuracy in pass@1. **So there is a valid question as the reviewer has aptly raised on how much can SBSC trajectory really solve more.**
> > >
> > > **o1 like complex reasoning methods achieves new SOTA for COT based math reasoning:** Sophisticated reasoning systems like o1 perform "long thinking" to do potentially tree-search/self-reflection/debate over natural-language reasoning chains to arrive at the final trajectory [1,2, 3]. They achieve significantly higher scores, on AIME and AMC, compared to previous COT scores by frontier LLMs such as Claude 3.5 Sonnet and GPT4o and use human like COT reasoning.
> > >
> > > **Understanding SBSC's potential with pass@k metric and comparing o1:**  We argue that SBSC given its unique step-by-step coding nature holds lot of potential. It unlocks dynamic problem solving via discovering and solving intermediate sub-tasks.
> > > - We evaluate this first via quantitatively analysis by calculating pass@k scores on last 3 years AIME questions. We present the table below.
> > > - We observe that SBSC's pass@16 and pass@25 is able to solve 49 and 52 questions out of 90 AIME questions respectively. Pass@25 SBSC score surpasses  o1 mini by 8 questions and o1 preview by 13 questions. We also do topic wise analysis to understand the strengths and weaknesses of SBSC.
> > > - We further do error analysis on the 38 questions that couldn't be solved with pass@25 SBSC. We do this across the math topics. We define different error types.
> > >
> > > **Note/Scope of this study:** This study is to understand the potential of SBSC and not about claiming COT like reasoning is not sufficient. We believe SBSC based program generation based methods could also unlock advanced mathematical reasoning. We have already demonstrated this across four datasets in the paper and 2 more ( JEE-Bench and Omni-Math) during rebuttal. We try to show more evidences with this study. Given, o1 like new complex reasoning systems unlocks the SOTA/superior COT trajectory, so we also perform a comparison against them.
> > >
> > > [1] Qin et al., “O1 Replication Journey: A Strategic Progress Report -- Part 1”,   	arXiv:2410.18982
> > > [2] Huang et al., “O1 Replication Journey -- Part 2: Surpassing O1-preview through Simple Distillation, Big Progress or Bitter Lesson?”, arXiv:2411.16489
> > > [3] https://openai.com/o1/

---

> ### Author Response · Authors · 2024-11-21
> **Author Response to Official Review by Reviewer uQkW (4/N)**
>
> > [Questions #2 part 1] While step-by-step reasoning is a natural approach, writing and displaying code in steps is not common practice. Programmers often organize their code by defining classes and functions first, then calling them in a main function. How can we ensure that LLMs are capable of generating effective step-by-step code? I believe that the authors should add some experiments to support current LLMs have this ability.
>
> ### Answer ###
> Thank you for your question! We would like to attempt to answer it via the following pointers
>
> **Purpose of SBSC:** The reviewer has raised a valid point about traditional software engineering programming practices during program generation of SBSC. However, we believe it is important to distinguish between software development and the cognitive process of problem-solving using code as a computational tool. The SBSC framework is mainly focused on enhancing LLMs' math problem-solving capabilities through multi-turn tool-integrated reasoning using python programs.
>
>
> **Skills Requirements for SBSC:** The SBSC framework primarily requires two fundamental capabilities in the base LLM to generate responses in the desired codified sequential stepwise format:
>
> - Mathematical/logical reasoning: Understanding math problems, formulating solution strategies in a stepwise manner and identifying computational dependencies.
> - Code generation: Ability to generate python code snippets based on given context including earlier code steps and sub-task definition.
> The capabilities required to enable this necessarily don’t require utilizing all the traditional software engineering skills/practices and align more closely with generating reasoning dense programs for problem-solving.
>
> **SBSC Correlation with LLM Code Generation ability:**
> - As discussed above, the code generation ability of the language model is a key factor that may directly influence the generation of suitable SBSC-format responses, and therefore the overall performance of the framework.
> - Two of the most popular benchmarks for evaluating programming abilities of LLMs are LiveCodeBench [1] and SWEBench. Succeeding on SWEBench [2] requires LLMs to do repository level coding where following software engineering practices are imperative. LiveCodeBench includes competitive programming contest problems and is more focused on evaluating the ability to generate code for problem-solving. Given the premise of SBSC, the latter is a more suitable metric for checking capability of LLM in leveraging SBSC effectively.
> - To gain a deeper insight into this, as shown in Section 5.5, we make a comparison involving LLMs with varying level of code generation abilities. Figure 6 clearly shows that the SBSC scores have a strong, positive correlation with the code generation abilities of the base LLM models for all cases that were evaluated on a subset of AIME and AMC data. The code-generation scores were taken from LiveCodeBench benchmark.
>
>
> [1] Jain et al., “LiveCodeBench: Holistic and Contamination Free Evaluation of Large Language Models for Code”, arXiv:2403.07974.
> [2] Gimenez et al., “SWE-bench: Can Language Models Resolve Real-World GitHub Issues?”, 	arXiv:2310.06770

---

> > ### Author Response · Authors · 2024-11-21
> > **Author Response to Official Review by Reviewer uQkW (5/N)**
> >
> > > [Questions #2 part 2] Additionally, some steps in the reasoning process may not correspond to clear intermediate results in code. For instance, what happens if a step only defines a function with no immediate output to display? I'm curious whether this decomposition approach remains effective across different problem types.
> >
> > ### Answer ###
> > We appreciate reviewer’s in-depth inquiry about SBSC which also gives an opportunity to highlight an important property of SBSC.
> >
> >  The SBSC framework accommodates various types of intermediate reasoning steps/sub-tasks, not just those producing immediate numerical or computational outputs. Here's a detailed breakdown on the kind of steps:
> > - Computational Steps:
> >      - Direct calculations producing numerical results
> >      - Data transformations with clear outputs
> > - Preparatory Steps:
> >      - Problem decomposition decisions
> >      - Variable and constant declarations
> >      - Function definitions or Data structure initialization for later use
> >      - Algorithm setup and configuration
> > - Verification/Error Correction Steps:
> >      - Verification of intermediate or results
> >      - Rectifying an erroneous intermediate step
> >
> > The strict output requirement of SBSC framework serves multiple purposes such as: verification of step completion, as input for subsequent reasoning steps, error information for recovery and documentation of solution progress. Its robustness is particularly evident in complex problem-solving scenarios where traditional monolithic approaches might fail to capture intermediate state changes or error conditions effectively. This systematic approach to output management ensures that the LLM maintains a clear understanding of the solution's progress, regardless of the step's nature, while providing necessary feedback for both the model and the user about the solution's development. As mentioned in “Future Work”, we believe that due to this very nature of structure output protocol for intermediate states/answers generated during SBSC could be highly useful, unlike previous methods such as TIR-ToRA and PAL, for guided decoding strategies such as in Outcome-Supervised Value, AlphaMATH.

---

> > > ### Author Response · Authors · 2024-11-21
> > > **Author Response to Official Review by Reviewer uQkW (6/N)**
> > >
> > > > [Questions #3]  Does the method's effectiveness depend on delicate prompt engineering or careful selection of few-shot examples? Although the experiments in Section 5.1 suggest the method is not overly sensitive to the few-shot candidate selection, I would like more clarity on the details and principles behind crafting these 10 examples. When generalizing the method to other tasks, how should the steps be divided to ensure each step yields clear, intermediate results that aid subsequent reasoning?
> > >
> > > ### Answer ###
> > >
> > > We would like to answer reviewer’s question via three topics: Exemplar Curation, Exemplar tuning, Exemplar choice sensitivity analysis
> > >
> > > **Exemplar Curation & Crafting:**
> > >
> > > To enable the multi-turn reasoning for SBSC, we want the format to have the key characteristics:
> > > - The problem is solved using a sequence of meaningful codified steps, each of them yielding clear, intermediate results that aid subsequent reasoning
> > > - At each step, we should be able to leverage outputs of previous steps to define and solve/conclude the the next sub-task.
> > >
> > > To enable this and to ensure the LLM sticks to this format consistently, we use combination of system prompt and few-shot prompting (exemplars). To prepare the exemplars
> > > - We first used 0-shot prompting (system prompt) to generate responses for questions in the SBSC format. As mentioned in section 3.3, for both AIME and AMC, we select 90 questions each, drawn from problems of years other than those included in the evaluation datasets.
> > > -	We observe that out of 90, we only had 15 questions that were solved across the methods (COT, PAL, TIR-ToRA, SBSC). This churn was particularly due to the low accuracy of COT method.
> > > -	A pool of 10 common problems and corresponding exemplars were finally selected. In this step, we ensured the diversity in the problem set and ensured that SBSC format was accurately followed in them.
> > > -	However, the selection of 4 exemplars from the 10 is completely random.
> > > -	Additionally, as mentioned in section 2.1, we also ensure certain practices (mainly 2) for program generations in the exemplars which are inspired from PAL paper. First is to have meaningful and coherent variable names and second is to having natural language intermediate comments within the program. We observed that the base LLM followed the first point (meaningful variable names) by default. For the second point which is the natural language comments within the code, we used GPT-4o to add comments within the generated programs for the final 10 curated exemplars. We even measured the impact of the exemplar tuning in section 5.2 and Table 2.
> > >
> > > **Additional Exemplar Tuning:** Further, in section 5.2, we study the impacts of 2 components of the prompt separately:
> > > a.   One of the instructions in the system prompt that ensures the step-level error correction practice.
> > > b.   Additional natural language comments within each Python code snippet, which acts as a code commentary.
> > > Both components are found to be helpful based on our evaluation on a subset of AMC and AIME problems from last 3 years.
> > >
> > >
> > > **Exemplar Sensitivity Analysis:** As shown in section 5.1, we randomly sample 4 exemplars out of an already created pool of 10 exemplars three times to create 3 variations of 4-shot prompts: v1, v2, and v3. In Figure 3, we can see that the performance remains stable irrespective of the three variations of the exemplar set used.

---

> > > > ### Author Response · Authors · 2024-11-21
> > > > **Author Response to Official Review by Reviewer uQkW (7/N)**
> > > >
> > > > > [ Questions #4 part 1] The authors say "similar to NuminaMath, we remove all the answer choices from each AMC-12 question and modify the question to ensure an integer answer". But I cannot find evidence that the NuminaMath modifies questions to ensure an integer answer. Rather, it is a preprocess in original AIME dataset to keep answer as integer and Numina focuses on this problem only for decontamination. The authors should clarify this point.
> > > > ### Answer ###
> > > >
> > > > We appreciate your keen observation on the test-set preparation.
> > > >
> > > > We would like to clarify by "modifying the question to ensure an integer answer" we mean the same pre-processing step as you mentioned for NuminaMath. What we meant was that wherever necessary, we have modified the representation of the final answer in such a manner that it becomes an integer. Kindly refer to the example shown in Section 3.1.1 to have a clearer understanding of this preprocessing step. For this, we took inspiration from Numina-Math. So, accordingly, a better phrasing would be “similar to NuminaMath, we remove all the answer choices from each AMC-12 question and modify the representation of the final answer whenever necessary to ensure an integer answer”. We will update the paper as well for clearer communication.
> > > >
> > > >
> > > >
> > > >
> > > >
> > > > > [ Questions #4 part 2] In Figure 4, how is the x-axis "error steps per problem" determined? A more detailed explanation of this experiment would be helpful.
> > > > ### Answer ###
> > > >
> > > > Any step in the overall solution that yields an error message as code interpreter output has been classified as an error step (ValueError, KeyError, TimeOut Error, etc.) We don’t individually verify the mathematical output of each step and hence, such errors are not counted if the code runs properly.

---

> ### Author Response · Authors · 2024-11-30
> **A study to further understand the potential of SBSC and comparison with o1 (2/N)**
>
> ## Analysis on SBSC’s pass@k accuracy on last 3 years AIME and comparison with o1 ##
>
> **Why use pass@k for SBSC:** Here we leverage self-consistency settings to make model generate multiple SBSC trajectories to examine the pass@k curve by increasing k from 1 to 25. We are using pass@k metric as we want to know if the model can generate a correct SBSC based reasoning path for a particular problem. If there is even 1 correct chain for a question that means SBSC format supports solving that question. Also, o1 has already established that by exploring multiple paths/trajectories, we can arrive at the correct trajectory. Once we know SBSC’s pass@k, it can help us better understand that when we build critic systems or thinking systems that leverage SBSC, how much can these systems achieve. Hence in this study we focus on pass@k for SBSC to understand its potential. We also benchmark against o1 preview and o1 mini. We do pass@1 for the o1 models as they already go through multiple trajectories via slow/long internal reasoning before outputting the final correct reasoning path. Also, since o1 is the new SOTA COT based method.
>
> **Experiment details:** We do this study on last 3 years AIME questions. We use claude-3.5-sonnet as the base LLM. For generating multiple chains we use temperature 1 and top p = 0.95 for COT & SBSC. We specifically compare with the previous SOTA simple/system-1 COT with same claude model in pass@25 mode. o1 mini and o1 preview represent the current SOTA COT based methods.
>
> | Topic                  | Number of Q's | Sonnet-COT pass@1 | Sonnet-COT pass@25 | Sonnet-SBSC pass@1 | Sonnet-SBSC pass@4 | Sonnet-SBSC pass@9 | Sonnet-SBSC pass@16 | Sonnet-SBSC pass@25 | O1 preview | O1 mini |
> |------------------------|---------------|-------------------|--------------------|--------------------|--------------------|--------------------|---------------------|---------------------|------------|---------|
> | Algebra       | 21            | 4                 | 10                 | 11                 | 14                 | 16                 | 16                  | **19**                  | 12         | 14      |
> | Combinatorics  | 23            | 1                 | 6                  | 8                  | 10                 | 12                 | **13**                  | **13**                  | **13**         | 10      |
> | Geometry       | 32            | 1                 | 6                  | 2                  | 3                  | 4                  | 9                   | 9                   | 10         | **13**      |
> | Number Theory  | 14            | 2                 | 2                  | 7                  | 9                  | 9                  | **11**                  | **11**                  | 4          | 7       |
> | Total Qs solved (90 Qs) | 90      | 8            | 24             | 28             | 36             | 41             | 49             | **52**              | 39     | 44  |
>
> **Results:** We present pass@1, pass@4, pass@9, pass@16, pass@25 for SBSC in the table above
> - Overall analysis: We observe that SBSC with just pass@4 approaches o1 level accuracy and with pass@9 it surpasses o1-preview and almost touched o1-mini. With pass@16 and pass@25 SBSC takes significant lead wr.t o1 models. At pass@16 SBSC solves  5 more questions than o1 mini and 10 more questions than o1-preview. At pass@25 SBSC solves 8 more questions than o1 mini and 13 more questions over o1 preview. We can clearly understand that SBSC can potentially unlock more correct chains and solve 52 questions out of 90.
> - Topic wise analysis:
>     - Algebra: We can observe that with just pass@4, SBSC achieves o1 mini and surpasses o1 preview level performance and with pass@25 setting, SBSC solves 90% of the algebra questions
>     - Number Theory: With just pass@1, SBSC surpasses o1 preview and matches o1 mini. At pass@16 & pass@25, SBSC is able to solve 4 more number -questions than o1 mini and 7 more questions than o1 preview.
>     - Combinatorics: At pass@16 and pass@25, SBSC matches o1 preview and surpasses o1-mini by solving 13 questions out 23.
>     - Geometry: This is the only topic where SBSC lacks behind o1 models. SBSC is unable to solved 23 out of 32 questions. However, o1 mini too is unable to solve 19 questions and o1 preview fails to solve 22 questions.
>
> **Error Analysis (PART 1)**: Within pass@25 the model was not able to generate correct SBSC trajectory for 38 AIME questions out of 90 questions. We show topic wise error distribution
> - 2 out of 21 from Algebra.
> - 10 out of 23 from combinatorics
> - 23 out of 32 from geometry
> - 3 out of 14 from number theory

---

> ### Author Response · Authors · 2024-11-30
> **A study to further understand the potential of SBSC and comparison with o1 (3/N)**
>
> We manually go through the trajectories(one per question) for these 38 questions. So we analyse 38 SBSC paths. We classify the errors in multiple categories as shown in the table below.
>
>
> | Error Type                  | Error Definition                                                                    | Algebra | Combinatorics | Geometry | Number Theory |
> |-----------------------------|------------------------------------------------------------------------------------|---------|---------------|----------|---------------|
> | Conceptual Reasoning Error             | Mistakes due to incorrect reasoning steps or missing conditions                    | -       | 10            | 10       | 2             |
> | Hallucination               | Fabrication of numbers or answers                                                  | 2       | -             | 6        | 1             |
> | Spatial Reasoning    | Misinterpretation or inability to manipulate given diagrams or geometric scenarios                         | -       | -             | 7        | -             |
> | **Total Errors**            |                                                                                    | **2**   | **10**        | **23**   | **3**         |
>
> We perform qualitative error analysis on geometry and combinatorics (as algebra and number theory scores are high)
> ### Qualitative error analysis For Geometry ###
> While we acknowledge that Chain-of-Thought reasoning could potentially improve with better mathematical expertise in training data, our error analysis suggests that certain mathematical challenges face fundamental limitations that go beyond reasoning methodology.
>
> Across all datasets we have evaluated on, Geometry has emerged as a particularly challenging topic, having the least success rate compared to other topics of the same competition dataset. Our analysis reveals that geometry problems present unique challenges that transcend both program-assisted and pure reasoning approaches. This is evidenced by consistently weak performance patterns across different methodologies - whether program-assisted (SBSC, ToRA) or CoT-based (including even OpenAI's o1). The universal struggle with geometric problems suggests that the limitations aren't tied to the choice between program-assisted or CoT approaches, but rather to fundamental challenges in spatial reasoning and visualization.
>
> Geometry is particularly challenging because:
> 1.	Geometry requires deep knowledge of specific theorems and properties involving circumcenters, orthocenters, etc., as well as sophisticated techniques like the notion of Power of Point, Projective Geometry, Inversion, etc.
> 2.	Problems with accompanying diagrams in Asymptote language seem to pose additional challenges for both approaches. Some of them further require quite sophisticated constructions to be done cleverly to solve the problem efficiently.
> We also provide some examples of such questions where most of the solutions (available on AoPS forum) make use of auxiliary constructions): https://artofproblemsolving.com/wiki/index.php/2022_AIME_I_Problems/Problem_8, https://artofproblemsolving.com/wiki/index.php/2023_AIME_I_Problems/Problem_12,
> https://artofproblemsolving.com/wiki/index.php/2024_AIME_II_Problems/Problem_10,
> https://artofproblemsolving.com/wiki/index.php/2024_AIME_I_Problems/Problem_10
>
>
>
> SBSC finds it relatively easy to deal with problems which can be solved without the need of diagrammatic manipulations, some of which can be based on mensuration or can be solved using trigonometric relations. In fact, we have even found cases where o1_mini failed but SBSC succeeded. Example:
> {https://anonymous.4open.science/r/SBSC-69D3/example_responses/AIME_23_Q8_comparison.md}
>
>
> Pertaining to Olympiad geometry solving using AI, Alpha Geometry [1] has been the most successful. They acknowledge similar challenges as us: *“Olympiad geometry problems are based on diagrams that need new geometric constructs to be added before they can be solved, such as points, lines or circles. AlphaGeometry’s language model predicts which new constructs would be most useful to add, from an infinite number of possibilities. These clues help fill in the gaps and allow the symbolic engine to make further deductions about the diagram and close in on the solution.”* This was achieved by pretraining a language model on synthetic data and fine-tuned to focus on auxiliary construction alongside a database of geometry axioms and theorems that enable the solution generation process. They try to overcome similar issues that we observed for SBSC faced as mentioned above.
>
> [1] Trinh, T.H., Wu, Y., Le, Q.V. et al. Solving olympiad geometry without human demonstrations. Nature 625, 476–482 (2024). https://doi.org/10.1038/s41586-023-06747-5

---

> ### Author Response · Authors · 2024-11-30
> **A study to further understand the potential of SBSC and comparison with o1 (4/N)**
>
> ### Error Analysis On Combinatorics ###
>
> Though SBSC has failed to correctly solve some of the Combinatorics problems, our analysis suggests no evidence to believe that program-generation based methods cannot solve these problems. There is no distinct difference between the types of combinatorial problems that it succeeds and fails to solve. The cause of failure here are conceptual errors such as:
> - Unable to understand independence of events
> - Counting duplicate cases or failing to consider all relevant scenarios
> - Forgetting certain constraints midway
>
> We notice that even COT based methods make similar errors and hasn’t managed to surpass SBSC pass@25. Also, our method itself has successfully solved similar problems requiring the understanding of these concepts. Here also, we provide an instance where SBSC managed to correctly solve a problem that o1_mini couldn’t {https://anonymous.4open.science/r/SBSC-69D3/example_responses/AIME_24_Q16_comparison.md}.
>
> ### Additional positive qualitative observation about SBSC ###
>
> - SBSC has shown promise in solving challenging problems at Olympiad level. SBSC dynamically discovers sub-tasks and solves the sub-task using program generation. This program generation often mimic human-like reasoning at its core, only the logic is codified instead of being written in a natural language form. We provide example SBSC response for a question covering that portray the same. {https://anonymous.4open.science/r/SBSC-69D3/example_responses/AIME_22_Q9_SBSC.md}
>
> - Program generation enables precise and accurate mathematical computations. The access to useful mathematical libraries like SymPy  allows it to simplify expressions, expand polynomials, and solve algebraic equations exactly rather than approximately. As such, these additional features are often useful in solving complex problems that cannot be solved correctly even by the strongest language models using natural language reasoning. We present an examples where SBSC could correctly solve problems that o1_mini or preview could not, by cleverly making use of these libraries alongside natural language reasoning. {https://anonymous.4open.science/r/SBSC-69D3/example_responses/AIME_24_Q13_comparison.md}
>
> ## Conclusion of the study: ##
>
> To summarise, in this study we presented the performance of pass@k (k=1,4,9,16,25) scores for SBSC with claude 3.5 sonnet on last 3 years AIME questions. We calculate pass@k scores to understand the potential of SBSC reasoning. To understand to what extent SBSC reasoning format can be used to solve challenging math problems. We show that pass@16 and pass@25 SBSC scores surpasses o1-mini and o1 preview. o1 mini and preview represent SOTA COT models. At pass@25 SBSC solves 8 more questions (out of 90) than o1 mini and 13 more questions over o1 preview. Even topic-wise except for geometry, SBSC surpasses o1 models in all other topics. We also deep dive on geometry problems that presents certain unique challenges not just for program generation based SBSC but SOTA COT models like o1. We also highlight that in combinatorics, the model made conceptual errors which does not prove that SBSC cant solve them as the model had solved other conceptual questions using SBSC. We also share couple of positive pointers on SBSC that it also can do natural reasoning via code and can smartly leverage programs/libraries to solve questions that even o1 cant solve.

---

> > ### Comment · Reviewer_uQkW · 2024-12-02
> > **Thank you for your further explanation!**
> >
> > Thank you for your detailed explanation regarding the comparison between program-aided and CoT methods. Some of my concerns have been addressed. I agree that if step-by-step programming can be integrated into a tree-like, long-thinking model, it could enhance performance. Additionally, while I believe certain challenging math problems (such as those in geometry or combinatorics) may not naturally lend themselves to programming, I also think that if CoT can eventually solve them, a program-aided model could potentially leverage the appropriate library for the task.
> >
> > Another issue is interpretability (or readability); for example, traditional computer-generated proofs are often not easily understood by humans. However, this could be a future step to address.
> >
> > There are still many challenges to making LLMs more capable of solving these very difficult math problems. For instance, they should be more familiar with the latest libraries containing advanced mathematical tools and should strive to solve problems in a more concise, human-like manner rather than relying on brute-force approaches. If the authors believe that program-aided LLMs are essential, there is certainly much work to be done to make them practical. Regardless, I believe this paper is strong enough to warrant acceptance at this stage, and I am raising my score.
> >
> > Finally, as a friendly reminder, some parts of your reply were a bit long-winded and deviated from my original questions. It would be helpful if the responses could be more focused and directly relevant to the points raised.

---

> ### Author Response · Authors · 2024-12-04
> **Thank you for the valuable discussion**
>
> Dear Reviewer,
>
> Thank you for acknowledging our efforts in answering your queries and we truly appreciate the increase in overall rating. We would also like to thank you for asking such insightful questions that allowed us to showcase a deeper understanding of our method as well as highlight its further potential in the domain of Olympiad-level math reasoning and the broader implications of our work.

---

### Official Review · Reviewer_BqFn · 2024-11-03

**Soundness:** 3
**Presentation:** 3
**Contribution:** 2
**Rating:** 6
**Confidence:** 4

**Summary:**

The paper presents Step-by-Step Coding (SBSC), a framework that guides Large Language Models (LLMs) to solve advanced math problems by producing a sequence of programs. SBSC operates iteratively, using the outcome of code from previous steps to inform the generation of subsequent sub-tasks and solutions. This approach is shown to be more detailed, adaptable, and accurate than existing methods. The efficacy of SBSC is supported by extensive testing on competition-level math problems. For the Claude-3.5-Sonnet model, improvements using SBSC with greedy decoding are significant, with performance gains of 10.7% on AMC12, 8% on AIME, and 12.6% on MathOdyssey over current state-of-the-art strategies. Additionally, when comparing SBSC's greedy decoding to self-consistency decoding of current methods, there are notable increases of 6.2% on AMC, 6.7% on AIME, and 7.4% on MathOdyssey.

**Strengths:**

- The empirical results demonstrate that SBSC achieves superior performance over state-of-the-art methods, suggesting that its iterative nature allows for more accurate problem-solving in the context of advanced math problems.
- The paper's experiments cover a variety of math competitions, indicating that SBSC is not tailored to a specific type of problem but is instead broadly applicable to Olympiad-level mathematics.
- The paper is well-structured and clearly written.
- The ablation and analysis section offers a deep analysis of the results, providing insights into why SBSC performs better than existing methods and under what circumstances it might be most beneficial.

**Weaknesses:**

I appreciate the efforts made in this paper to apply step-by-step partial code generation and execution within the challenging domain of Olympiad-level mathematics and the promising results. My query pertains to the aspect of novelty concerning this approach. The paper states in Line 62 that "Fundamentally, both PAL & TIR-ToRA generate a single program block to solve the entire problem", highlighting step-by-step code generation and execution as a key contribution of SBSC. However, given that the concept of language models incrementally generating code to interface with external tools has been explored in prior research, such as ReAct (https://arxiv.org/abs/2210.03629) and ToolFormer (https://arxiv.org/abs/2302.04761), and the idea is highly related to OpenAI's code interpreter and assistant APIs, I am curious about the distinctiveness of the approach presented in this paper. May I kindly suggest that the authors could provide a more explicit comparison with these precedents? An elaboration on how the proposed SBSC framework diverges from or advances these existing methods, particularly in the unique setting of mathematical problem-solving, would be enlightening. This addition would help the readers and reviewers to better understand the specific innovation and value of the proposed work in the broader context of step-by-step code generation research.

**Questions:**

Apart from the issues expressed in the weaknesses section, I have no other concerns.

---

> ### Author Response · Authors · 2024-12-03
> **Author Response to Official Review by Reviewer BqFn**
>
> We would like to sincerely thank the reviewer for the detailed review of our paper. We truly appreciate that the reviewer acknowledged the superior performance of SBSC over the state-of-the-art methods. We are glad that the reviewer found our paper well-structured and also that our ablations and analysis helped in providing deeper insight about SBSC performance. Further, we appreciate the opportunity to clarify the points raised & to provide additional insights into our research.
>
> **Toolformer** is a framework that enables LLMs to learn to use external tools through self-supervised learning. It automatically learns to use external APIs/tools (like calculators, search engines, translation services, etc.) by embedding API calls into text.
>
> - In SBSC however, like previous methods PAL specifically designed for math reasoning or question-answering based scenarios, the tool is already selected which is the python interpreter. Similar to TIR-ToRA, this tool is called when the stop word is triggered.
>
> - Moreover, SBSC is specifically designed for mathematical reasoning. It treats code generation not just as a tool interface but as a mathematical reasoning strategy, allowing for rigorous concept exploration at each step.
>
> - PAL has already highlighted that "offloading" the final solution to an external Python interpreter helps in increasing accuracy by mitigating computational errors. TIR-ToRA which introduced tool-integrated reasoning for math, uses natural language-based reasoning followed by a single block of code to arrive at the final answer. Both PAL and TIR-ToRA focused on generating the final answer in the first tool call itself. However in SBSC, the idea is to focus on intermediate outputs and leverage them to dynamically identify and solve subsequent subtasks using Python codes until the answer is found.
>
>
> **ReAct** is a framework for solving various problems, that uses LLMs to generate both reasoning traces and task-specific actions in an interleaved manner by alternately thinking through tasks and interacting with external tools or environments.
>
> * We would like to highlight that ToRA authors and reviewers have acknowledged in the discussion [https://openreview.net/forum?id=Ep0TtjVoap] and the paper that alternate rationale and interaction-based reasoning is a probable paradigm to solve complex problems using LLMs.
>
> * TIR-ToRA first generates the rationale for solving a problem and then the corresponding Python program for it. In case the code fails to return the answer, the method restarts by generating a new rationale and the corresponding code. To summarize, it attempts to solve the entire problem at one go in every attempt.
>
> * Our proposed SBSC framework involves intermediate steps, which are mathematical sub-tasks that build upon each other sequentially, where each step's code execution result provides mathematical insights crucial for discovering the next sub-task.
>
> * SBSC provides fine-grained control over each step of the solution exploration. This helps in:
>   * Rectification of a specific intermediate step without affecting the entire solution chain
>   * Revisiting the solution in case the initial approach fails

---

### Official Review · Reviewer_EW8i · 2024-11-04

**Soundness:** 4
**Presentation:** 2
**Contribution:** 3
**Rating:** 6
**Confidence:** 3

**Summary:**

The paper proposes Step-by-Step Coding (SBSC) for solving complex mathematical problems for large language models. The authors conduct extensive experiments comparing SBSC against other state-of-the-art methods like COT, PAL, and TIR-ToRA on datasets like AIME, AMC, and MathOdyssey.

**Strengths:**

* The paper is easy to understand, with a well-defined problem statement that introduces multi-turn TIR as a solution for complex multi-step issues.
* The main experimental results are sufficiently thorough, providing reasonable evidence for the improvements attributed to SBSC.

**Weaknesses:**

* SBSC introduces an additional step of decomposing a large problem into smaller sub-problems during the decoding phase. How to ensure that this decomposition is both correct and rational? Could this lead to the introduction of additional errors?

* Existing methods, such as TIR-ToRA, output a planning-like COT before code generation. Does this serve the same function as the multi-turn approach proposed in this paper? Is it merely a modification that incorporates multiple compiler feedback loops?

* The introduction of multi-turn in SBSC may result in decreased efficiency. The manuscript should include comparisons under equivalent token costs.

* I strongly recommend that the authors review their writing for numerous typographical errors, particularly the inconsistent use of citet and citep.
    * line 157: "Our inference procedure is inspired by ToRA Gou et al. (2023)"
    * line 196: "AIME, AMC, MathOdyssey Fang et al. (2024) and OlympiadBench He et al. (2024)"
    * line 199: " MathOdyssey Fang et al. (2024), a popular benchmark..."
    * ....

**Questions:**

* Has SBSC demonstrated improvements across all domains of mathematical reasoning? Are there instances where multi-step code decomposition is not feasible or effective?

* Given the authors' claim regarding enhanced performance in Olympiad competitions, it would be beneficial to include benchmark results from:
    * JEE-Bench
    * OlympicArena
    * Omni-MATH
    * Additionally, consideration should be given to experiments involving GSM8K and MATH, as many problems in MATH derive from AMC and AIME questions.

---

> ### Author Response · Authors · 2024-11-24
> **Author Response to Official Review by Reviewer EW8i (1/N)**
>
> We thank the reviewer for the comprehensive review. We appreciate the opportunity to clarify the points raised & to provide additional insights into our research.
>
> > [Weakness 1] SBSC introduces an additional step of decomposing a large problem into smaller sub-problems during the decoding phase. How to ensure that this decomposition is both correct and rational? Could this lead to the introduction of additional errors?
>
> ### Answer ###
>
> SBSC starts with an initial sub-problem, which it considers to be a reasonable starting point to solve the problem, and then solves that sub-problem. At each step, leveraging the outputs/explorations of previous sub-tasks, the model generates the next rational sub-tasks and solves them. Notably, this means that there are no pre-decided fixed steps that must be codified to reach the final answer. Moreover, since SBSC solves the problem step by step so a wrong step is easily identified and self-corrected before the model moves to the subsequent steps. We can observe this in Fig 1, where due to dynamic and granular nature of SBSC, it is able to focus on the precise step and revise the sub-task and the approach to find the way forward. In section 5.3 and Fig 4 we also quantitatively highlight the significant improvement in self-correction ability compared to TIR-ToRA.
>
> > [Weakness 4] I strongly recommend that the authors review their writing for numerous typographical errors, particularly the inconsistent use of citet and citep.
>
> ### Answer ###
>
> We would like to thank the reviewer for pointing out the formatting issue. We have fixed this in our revised version.

---

> > ### Author Response · Authors · 2024-11-24
> > **Author Response to Official Review by Reviewer EW8i (2/N)**
> >
> > > [Weakness 2] 	Existing methods, such as TIR-ToRA, output a planning-like COT before code generation. Does this serve the same function as the multi-turn approach proposed in this paper? Is it merely a modification that incorporates multiple compiler feedback loops?
> >
> > ### Answer ###
> > **Motivation**: As mentioned in the paper, we think “Olympiad level math problem-solving can be viewed as solving/exploring an intermediate subtask/key-concept in depth; and discovering + solving the next critical sub-task dynamically basis the accumulated knowledge of previous sub-tasks/key-concepts explorations”.
> >
> > **Comparison**: We believe that the previous program generation-based methods have followed more of a static approach and hence were better suited for simpler problems. PAL directly codes the entire solution in a single block. TIR-ToRA does COT rationale generation in one go beforehand and then it codes the entire solution to get the final answer using single code block. Additionally, in case of an error, TIR-ToRA tries to re-attempt in similar fashion. But all these methods lack producing/exploring any intermediate steps or arriving at the solution in step-by-step progression.
> >
> > For complex math problems it is be difficult to have a clear idea of the entire solution at the start itself. They require an element of flexibility and exploration while still progressing towards the solution. Hence, SBSC achieves significant improvement over TIR-ToRA across 4 different benchmarks as can be seen in Table 1. SBSC also shows significant improvement in self-correction compared to TIR-ToRA as visible in Figure 4.
> >
> > Moreover, TIR-ToRA completely lacks any intermediate steps and is not dynamic and flexible. Given the multi-turn nature, SBSC produces intermediate outputs at each step. The SBSC framework accommodates various types of intermediate reasoning steps/sub-tasks, not just those producing immediate numerical or computational outputs. Here's a detailed breakdown on the kind of steps:
> > - Computational Steps:
> >      - Direct calculations producing numerical results
> >      - Data transformations with clear outputs
> > - Preparatory Steps:
> >      - Problem decomposition decisions
> >      - Variable and constant declarations
> >      - Function definitions or Data structure initialization for later use
> >      - Algorithm setup and configuration
> > - Verification/Error Correction Steps:
> >      - Verification of intermediate or results
> >      - Rectifying an erroneous intermediate step
> >  	Rectifying an erroneous intermediate step
> >
> >  The “intermediate output printing protocol at each step” of SBSC framework serves multiple purposes such as: verification of step completion, as input for subsequent reasoning steps, error information for recovery and documentation of solution progress. Its robustness is particularly evident in complex problem-solving scenarios where traditional monolithic approaches might fail to capture intermediate state changes or error conditions effectively. This systematic approach to output management ensures that the LLM maintains a clear understanding of the solution's progress, regardless of the step's nature, while providing necessary feedback for both the model and the user about the solution's development.
> >
> > **What SBSC could potentially unlock?** Process supervision [1] has shown to be much effective than Outcome supervision as it rewards intermediate reasoning steps. However, it requires lot of manual effort to annotate the intermediate level step. Newer methods such as Outcome-Supervised Value [2], Alphamath [3], Q* framework [4] have tried to achieve similar results without the manual effort of annotating the intermediate steps by using a value model instead.  These methods could benefit further if the key intermediate steps are already well defined and can further assist the value models. As mentioned in “Future Work”, we believe that due to this very nature of structure output protocol for intermediate states/answers generated, SBSC could be highly useful for the above-mentioned methods, unlike previous methods such as TIR-ToRA and PAL. Similar to process supervision, It would be also well suited for stepwise preference optimization for reasoning such as in Step-DPO [5]. SBSC trajectories could be used also for imitation learning via SFT. We plan to cover the down-stream benefits of SBSC in the next iteration of our paper.
> >
> > [1] Lightman et al., “Let's Verify Step by Step”	arXiv: 2305.20050
> > [2] Yu et al., “Outcome-supervised value models for planning in mathematical reasoning”    arXiv:2311.09724
> > [3] Chen et al., “Alphamath almost zero: process supervision without process”    arXiv:2405.03553
> > [4] Wang et al., “Q*: Improving multi-step reasoning for llms with deliberative planning”     arXiv:2406.14283
> > [5] Lai et al., “Step-DPO: Step-wise Preference Optimization for Long-chain Reasoning of LLMs ”    arXiv: 2305.20050

---

> > > ### Author Response · Authors · 2024-11-24
> > > **Author Response to Official Review by Reviewer EW8i (3/N)**
> > >
> > > >  [Weakness 3] The introduction of multi-turn in SBSC may result in decreased efficiency. The manuscript should include comparisons under equivalent token costs
> > >
> > >
> > > We appreciate reviewer’s keen observation. We would like to answer this via the following points
> > >
> > > First, we would like to highlight that we have already accounted for this and acknowledged this in the paper. In section 3.2 Baselines & Configurations under “Experiment” section, we have referred to Table 3 in Appendix A.1 and mentioned that, on an average, SBSC takes 6-7 turns (api calls) per problem. That section, 3.2, also mentions that, this was one of the two main reasons that we also benchmark SBSC’s greedy decoding results against self-consistency decoding results (majority@7) of COT, PAL & TIR-TORA. In Table 1, for Claude 3.5 Sonnet when comparing SBSC's greedy decoding against self-consistency decoding (majority@7) of the previous methods we observe significant absolute increase of 6.2% on AMC, 6.7% on AIME, and 7.4% on MathOdyssey.
> > >
> > > Second, we also have done a study with Claude 3.5 Sonnet on average cost per question to highlight that our original results in Table 1 in “Results section” already accounts for equivalent comparisons. We have taken average over 50 AIME + AMC Questions. We leveraged the prompt-caching of repeated input tokens while doing this project. Specifically, we cache the system prompt and exemplars for all the methods. The results are presented in the table below. When comparing average cost per question in caching mode, we observe that SBSC is 3X of TIR-ToRA, 4X of COT and 7X of PAL. When comparing average cost per question even in none-caching mode, we observe that SBSC is 6.2X of TIR-ToRA, 7X of COT and 14.3x of PAL. Hence, Table 1 already covers fair comparison part as well even if we take average cost per question under consideration. In Table 1, we already compared 1 chain of SBSC with 7 chains of other methods. Just as experimental detail, we leveraged prompt caching for our reported experiments with Claude 3.5 Sonnet.
> > >
> > >
> > >
> > > Average Cost per Question(In USD)
> > >
> > > | Claude with prompt caching                    | COT      | PAL      | TIR-ToRA  | SBSC (sum of all turns per Q) |
> > > |----------------------------|----------|----------|-----------|-------------------------------|
> > > | avg_input_tokens           | 110.75   | 107.75   | 479.25    | 4788.9                        |
> > > | avg_output_tokens          | 543.95   | 353.2    | 685.15    | 1070                          |
> > > | avg_cache_tokens           | 2754     | 1255     | 2692.2    | 23610.6                       |
> > > | avg_cost_per_question      | 0.0093   | 0.0060   | 0.0125    | 0.0375                        |
> > > | Cost ratio (SBSC/Methods)  | 4        | 6.25     | 3         | 1                             |
> > >
> > > | Claude without prompt caching   | COT      | PAL      | TIR-ToRA  | SBSC (sum of all turns per Q) |
> > > |----------------------------|----------|----------|-----------|-------------------------------|
> > > | avg_input_tokens           | 2858.75  | 1356.75  | 3565.45   | 32846.7                       |
> > > | avg_output_tokens          | 536.05   | 267.35   | 543.7     | 1161.5                        |
> > > | avg_cost_per_question      | 0.0166   | 0.0081   | 0.0189    | 0.1160                        |
> > > | Cost ratio (SBSC/Methods)  | 7        | 14.3     | 6.2       | 1                             |

---

> ### Author Response · Authors · 2024-11-24
> **Author Response to Official Review by Reviewer EW8i (4/N)**
>
> > [Questions 1]	Has SBSC demonstrated improvements across all domains of mathematical reasoning? Are there instances where multi-step code decomposition is not feasible or effective?
> ### Answer ###
> We appreciate reviewer’s enquiry about SBSC. To answer this question, we would like to highlight our section 5.4 and Fig 5, where we have presented topic wise improvement for SBSC over TIR-ToRA. As we can see SBSC shows improvement across all the topics and across all the datasets. While this comparison was between two program-generation based methods, to this end we also present a table where we perform topic wise analysis of SBSC and COT as well in the table below. As we can see SBSC not only improves over TIR-ToRA but also again natural language based reasoning/COT. We believe multi-turn reasoning is important for complex math problems, and enabling that with program generation like SBSC which allows step by step exploration of intermediate sub-tasks via program generation is why we observe improvement in accuracy.
>
> | Dataset         | Topic                  | TIR-ToRA (%) | SBSC (%) | COT (%)  |
> |-----------------|------------------------|--------------|----------|----------|
> | AIME            | Algebra                | 23.7         | **44.7**     | 10.53    |
> |                 | Arithmetic             | **80.0**         | **80.0**     | 40.00    |
> |                 | Counting & Probability | 31.4         | **46.5**     | 10.47    |
> |                 | Geometry               | 1.8          | **7.3**      | 4.55     |
> |                 | Number Theory          | 54.7         | **62.3**     | 13.21    |
> | AMC             | Algebra                | 47.3         | **50.9**     | 33.53    |
> |                 | Arithmetic             | 61.5         | **74.4**     | 64.10    |
> |                 | Counting & Probability | 34.8         | **55.4**     | 23.91    |
> |                 | Geometry               | 17.8         | **31.0**     | 19.38    |
> |                 | Number Theory          | 52.1         | **62.5**     | 41.67    |
>
> > [Questions 2] Given the authors' claim regarding enhanced performance in Olympiad competitions, it would be beneficial to include benchmark results from:
> o	JEE-Bench
> o	OlympicArena
> o	Omni-MATH
> o	Additionally, consideration should be given to experiments involving GSM8K and MATH, as many problems in MATH derive from AMC and AIME questions.
> ### Answer ###
> We appreciate reviewer’s suggestion to include more benchmarks. Before that we would like to mention that, in Table 1, we have currently added 4 datasets that cater to competition and Olympiad level questions, namely: AIME, AMC, MathOdyssey and Olympiad Bench.
>
> Given the resource constraint we are unfortunately unable to cater to all the recommended benchmarks. We think SBSC is best tested on the tougher benchmarks. So we choose to present more results on two benchmarks: JEE-Bench and Omni-Math. For JEE-Bench, we evaluate on 98 numerical answer type questions. For Omni-MATH, we filter out 576 questions with all 26 Qs from calculus topics and 110 Qs randomly sampled from each one of the remaining topics.
>
> For JEE-Bench, we observe significant improvements compared to other methods. For Claude 3.5 Sonnet, SBSC’s greedy decoding takes an absolute lead (against greedy decoding) of 16% over TIR-ToRA, 20% over PAL and 30% over COT. Similarly benchmarking against self-consistency decoding (majority@7) of other methods, SBSC takes an absolute lead of 11% over TIR-ToRA, 16% over PAL and 25% over COT.
>
> |JEE-Bench |COT (greedy) |COT (maj @7) |PAL (greedy)|PAL (maj @7)|TIR-ToRA (greedy) |TIR-ToRA (maj @7) |SBSC (greedy) |
> |---------------------|--------------|--------------|--------------|--------------|-------------------|-------------------|---------------|
> | Claude 3.5 Sonnet   | 32.65        | 37.76        | 42.86        | 46.94        | 45.92             | 51.02             | **62.24**         |
> | GPT-4o              | 29.59        | 33.67        | 35.71        | 38.78        | 32.65             | 37.76             | **50.00**         |
>
>
> On omni-math dataset, SBSC(greedy decoding) shows an absolute improvement over TIR-ToRA (greedy decoding) by roughly ~10% using Claude-3.5-Sonnet and 6% using GPT-4o. SBSC greedy decoding results outperforms self-consistency decoding results of TIR-TORA by absolute 6% and 4%, for Claude-3.5-Sonnet and GPT-4o respectively.
>
> | OmniMath |COT (greedy)|COT (maj @7)|PAL (greedy) |PAL (maj @7) |TIR-ToRA (greedy) |TIR-ToRA (maj @7) |  SBSC (greed) |
> |---------------------|--------------|--------------|--------------|--------------|-------------------|-------------------|---------------|
> | Claude 3.5 Sonnet   | 23.09        | 26.22        | 32.81        | 35.94        | 33.68             | 37.15             | **43.06**        |
> | GPT-4o              | 27.78        | 29.86        | 34.38        | 37.15        | 33.85             | 36.28             | **40.45**         |

---

> > ### Author Response · Authors · 2024-11-30
> > **Gentle Reminder**
> >
> > Dear reviewer EW8i,
> >
> > As the discussion period is approaching the closing date, we want to check in on whether our responses have addressed your concerns. We would be happy to discuss further.
> >
> > Authors

---

> > > ### Comment · Reviewer_EW8i · 2024-12-02
> > > **Response to the Authors**
> > >
> > > Thank the author for the extra experiments and efforts to address my concerns. I will correspondingly raise the **soundness to 4**.

---

> > > > ### Author Response · Authors · 2024-12-02
> > > > **Summary of Rebuttals & Revisions**
> > > >
> > > > Dear Reviewer EW8i
> > > >
> > > > We are glad that we were able to address your concerns. We truly appreciate the increase in soundness score. We have tried to summarize our rebuttal discussion below:
> > > >
> > > > - **Clarified the flexible and dynamic nature of sub-task generation** in SBSC that leverages the explorations of previous sub-tasks and enables granular step-level correction if required. (Reply 1/N)
> > > >
> > > > - Provided a **detailed explanation of various types of intermediate reasoning steps/sub-tasks** that may be Computational/Preparatory/Error-Correction or Verification based, all of which provide important feedback to the model and aids in subsequent solution development. (Reply 2/N, *added Appendix A.7 in paper*)
> > > >
> > > > - Supplemented our **comparative results under equivalent token costs in Section 4** by providing details on **average cost per question** using different methods, under both prompt caching and non-caching settings. (Reply 3/N, *added Appendix A.4 in paper*)
> > > >
> > > > - **Added COT scores to the table comparing the topic-wise results** to demonstrate that SBSC outperforms not just program-generation based methods like TIR-ToRA but also natural language-based ones across all topics on all datasets. (Reply 4/N, *added Section 5.2 in paper*)
> > > >
> > > > - Provided additional **benchmarking results on two new datasets as requested: OMNI-Math and JEE-BENCH**. (Reply 4/N, *added Appendix A.2 in paper*)
> > > >
> > > > - Lastly, we have rectified the typographical errors and presentation issues in our paper that were pointed out.
> > > >
> > > > If all concerns have been addressed to your satisfaction, we would be truly grateful if you could consider upgrading your score.
> > > >
> > > > Authors

---

### Official Review · Reviewer_f1zY · 2024-11-05

**Soundness:** 3
**Presentation:** 4
**Contribution:** 1
**Rating:** 5
**Confidence:** 4

**Summary:**

The paper introduces a step-by-step prompting technique for solving math problems using large language models (LLMs). In this approach, the model is prompted to generate sub-tasks in code format at each step, which are then executed to produce intermediate outputs. These outputs, along with the code from previous steps, are used to prompt the model again to solve the problem sequentially. The proposed method is tested a few benchmarks, including MC, AIME, and MathOdyssey, demonstrating improvements over existing prompting techniques.

**Strengths:**

A key strength of this paper is the step-by-step approach, which leverages inference time compute to break down complex problems into manageable sub-tasks. For each sub-task, SymPy-based Python code is generated, and the model is instructed to include a print statement at the end of each code snippet so that intermediate execution outputs can be generated.

This structure allows the model to build on the results of previous steps, using the executions output/feedback to condition its generation of the next step and sub-task.

The step by step process with intermediate feedback provides the model with a chance to detect and correct errors or rewrite parts of the code. This improves the overall performance of the model.

**Weaknesses:**

The paper's approach relies heavily on the careful design of the step by step coding framework exemplars. This with the inclusion of detailed comments within the python codes in these exemplars seems to require considerable manual effort. While this strategy is effective for the test sets discussed, it may be overfitted to these specific benchmarks. This raises some concerns about the generalization of the method to broader problem sets or applications.

Another limitation is that the method is highly dependent on the coding capabilities of the underlying language model. While the authors acknowledge the correlation between the model's coding performance and the accuracy of the SBSC approach, the exact nature of this correlation is unclear.
In addition, the method relies heavily on SymPy which might further limit its flexibility and adaptability.

The approach demands a relatively long context length, which could be a constraint in certain settings. It would be helpful if the authors specified a rough minimum context length required for the given shots and the math problem. This would help get better insights into the method's scalability.

Lastly, the discussion of the related works section seems somewhat rushed. In my opinion, the first sentence feels awkward and reads more like filler rather than contributing meaningfully to the content. A more thorough comparison with existing literature on program or tool-aided generation (the section following POT and PAL) could be extended, as it is the most relevant part of the discussion. This would provide deeper insights into the novelty and strengths of the proposed method.

**Questions:**

I believe that randomly sampling 4 examples out of a set of 10, even multiple times, may not provide sufficient variety to effectively test the sensitivity of the prompts. It's unclear how diverse the 10 exemplars are to begin with. Perhaps a more meaningful  test would be to have a larger pool of exemplars?

In figure 5, are the methods compute/token matched? What is the ratio of the tokens generated by the SBSC method compared to others such as TIR-ToRA?

Figure 7 seems to have the wrong caption.

What is the advantage of this step-by-step approach compared to other similar methods, particularly in cases where the model is unable to revisit and revise a problematic step (e.g., Step 1) once it has progressed to later steps?

In table 1, why is there no number for maj@7 for SBSC?

How does the method perform when used with open LLMs such as LLaMA?

What types of errors are reported in Section 5.3? Are they primarily syntactic or semantic in nature?

---

> ### Author Response · Authors · 2024-11-26
> **Author Response to Official Review by Reviewer f1zY (1/N)**
>
> We thank the reviewer for the comprehensive review. We appreciate the opportunity to clarify the points raised & to provide additional insights into our research.
> > [Weakness] The paper's approach relies heavily on the careful design of the step-by-step coding framework exemplars. This with the inclusion of detailed comments within the python codes in these exemplars seems to require considerable manual effort. While this strategy is effective for the test sets discussed, it may be overfitted to these specific benchmarks. This raises some concerns about the generalisation of the method to broader problem sets or applications.
>
> > [Question] I believe that randomly sampling 4 examples out of a set of 10, even multiple times, may not provide sufficient variety to effectively test the sensitivity of the prompts. It's unclear how diverse the 10 exemplars are to begin with. Perhaps a more meaningful test would be to have a larger pool of exemplars?
>
> ### PART 1 ANSWER ###
>
> **Brief on Exemplar Curation & Crafting and why it didn’t require manual effort:**
>
> To enable the multi-turn reasoning for SBSC, we use combination of system prompt and few-shot prompting (exemplars) similar to what POT, PAL, TIR-ToRA used.  Similar to past works, our exemplar generation process didn’t require any manual effort other than writing the system prompt and manually inspecting the final pool for format check.
>
> To prepare the exemplars,
> - We first used 0-shot prompting (system prompt) to generate responses for questions in the SBSC format. As mentioned in section 3.3 “Prompting/Few-Shot Exemplars”, for both AIME and AMC, we select 90 questions each, drawn from problems of years other than those included in the evaluation datasets.
> -	We observe that out of 90, we only had 15 questions that were solved across the methods (COT, PAL, TIR-ToRA, SBSC). This churn was particularly due to the low accuracy of COT method.
> -	A pool of 10 common problems and corresponding exemplars were finally selected. In this step, we ensured the diversity in the problem set (we used GPT-4o to get the question categories) and ensured that SBSC format was accurately followed in them. The distribution was: 3 Algebra, 3 Number Theory, 2 Geometry, 2 Combinatorics.
> -	However, the selection of 4 exemplars from the 10 is completely random.
> -	**Additionally, as mentioned in section 2.1, we also ensure certain practices (mainly 2) for program generations in the exemplars which are inspired from PAL paper.** First is to have meaningful and coherent variable names and second is to having natural language intermediate comments within the program. We observed that the base LLM followed the first point (meaningful variable names) by default. **For the second point, which is the natural language comments within the code, we used GPT-4o to add comments within the generated programs for the final 10 curated exemplars. PAL paper already established (and used) these two things improve the performance.** We even measured the impact of the comment based exemplar tuning in section 5.2 and Table 2.
> -	The only thing manual in this entire exemplar curation process was writing the system prompt and final inspection if SBSC format was correctly followed or not.
>
>
> **Addressing Generalization doubt with existing work already in the paper:** As mentioned in section in section 3.3 under “Prompting/Few-Shot Exemplars”, we used AIME exemplars itself for MathOdyssey and OlympiadBench. As seen in result section 4 and Table 1, SBSC improves over TIR-ToRA and other methods even for different source of test-set as well for same exemplars. In addition to this, as shown in section 5.1, we randomly sample 4 exemplars out of an already created pool of 10 exemplars three times to create 3 variations of 4-shot prompts: v1, v2, and v3. In Figure 3, we can see that the performance remains stable irrespective of the three variations of the exemplar set used.
>
> Answer Continued .....

---

> ### Author Response · Authors · 2024-11-26
> **Author Response to Official Review by Reviewer f1zY (2/N)**
>
> > [Weakness] The paper's approach relies heavily on the careful design of the step-by-step coding framework exemplars. This with the inclusion of detailed comments within the python codes in these exemplars seems to require considerable manual effort. While this strategy is effective for the test sets discussed, it may be overfitted to these specific benchmarks. This raises some concerns about the generalisation of the method to broader problem sets or applications.
>
> > [Question] I believe that randomly sampling 4 examples out of a set of 10, even multiple times, may not provide sufficient variety to effectively test the sensitivity of the prompts. It's unclear how diverse the 10 exemplars are to begin with. Perhaps a more meaningful test would be to have a larger pool of exemplars?
>
> ### PART 2 ANSWER ###
>
> **Additional Exemplar Sensitivity & Generalization Analysis with bigger pool:** We conduct an additional study to expand the pool of exemplars for sensitivity analysis as suggested by the reviewer. We expand the pool to 30 AIME exemplars (prepared in the same manner as made initial 10 exemplars as mentioned above and in section 3.3 “Prompting/Few-Shot Exemplars”) that were solved where we have 8 Algebra, 8 Number Theory, 8 Combinatorics and 6 Geometry questions. We randomly sample 4 exemplars from these 30 exemplars 9 times to create 9 sets of 4-shot prompts: v1, v2, v3, v4, v5, v6, v7, v8, v9. We test them on the same set of last 3 years AIME questions.
>
> For further testament to our generalisation, we use AIME exemplars on MathOdyssey test-set.
>
> We can clearly observe that the performance remains stable as experienced as previous study with 9 exemplar-set randomly sampled from 30 AIME Qs. Also we can see the performance on MathOdyssey stays stable with even 9 AIME exemplar sets.
>
> | Exemplar Variation | Last 3 Yrs AIME Accuracy (%) |
> |-----------|--------------|
> | v1         | 31.11        |
> | v2         | 32.22        |
> | v3         | 32.22        |
> | v4         | 33.33        |
> | v5         | 32.22        |
> | v6         | 30.00        |
> | v7         | 28.88        |
> | v8         | 33.33        |
> | v9         | 28.88        |
>
> | Exemplar Variation | MathOdyssey Accuracy |
> |-----------|---------|
> | v1         | 39.86   |
> | v2         | 38.51   |
> | v3         | 37.17   |
> | v4         | 38.51   |
> | v5         | 41.89   |
> | v6         | 37.84   |
> | v7         | 39.86   |
> | v8         | 39.86   |
> | v9         | 37.84   |
>
> **We additionally use the AIME exemplars to test SBSC on additional & challenging datasets: JEE-Bench & Omni-Math**
>
> For JEE-Bench [1], we evaluate on 98 numerical answer type questions. For Omni-MATH [2], we filter out 576 questions out of that with all 26 Qs from calculus topics & 110 Qs randomly sampled from each one of the remaining topics.
> For JEE-Bench, we observe significant improvements compared to other methods. For Claude 3.5 Sonnet, SBSC’s greedy decoding takes an absolute lead (against greedy decoding) of 16% over TIR-ToRA, 20% over PAL and over 30% COT. Similarly benchmarking against self-consistency decoding (majority@7) of other methods, SBSC takes an absolute lead of 11% over TIR-ToRA, 16% over PAL and over 25% COT.
>
> | Model|COT (greedy) |COT (maj @7) |PAL (greedy)|PAL (maj @7)|TIR-ToRA (greedy) |TIR-ToRA (maj @7) |SBSC (greedy) |Δ% (greedy)|Δ% (maj@7)|
> |---------------------|--------------|--------------|--------------|--------------|-------------------|-------------------|---------------|------------|-----------|
> | Claude 3.5 Sonnet   | 32.65        | 37.76        | 42.86        | 46.94        | 45.92             | 51.02             | **62.24**     | +16.32     | +11.22    |
> | GPT-4o              | 29.59        | 33.67        | 35.71        | 38.78        | 32.65             | 37.76             | **50.00**     | +14.29     | +11.22    |
>
>
> On Omni-Math dataset, SBSC(greedy decoding) shows an absolute improvement over TIR-ToRA (greedy decoding) by roughly ~10% using Claude-3.5-Sonnet and 6% using GPT-4o. SBSC greedy decoding results outperforms self-consistency decoding results of TIR-TORA by absolute 6% and 4%, for Claude-3.5-Sonnet and GPT-4o respectively.
>
>
> | Model | COT(greedy) | COT (maj @7) | PAL(greedy) | PAL (maj @7) | TIR-ToRA(greedy) | TIR-ToRA (maj @7) | SBSC (greedy) | Δ% (greedy) | Δ% (maj@7) |
> |-------|-----|--------------|-----|--------------|----------|-------------------|------|-------------|------------|
> | Claude 3.5 Sonnet | 23.09 | 26.22 | 32.81 | 35.94 | 33.68 | 37.15 | **43.06** | \+9.38 | \+5.91 |
> | GPT-4o | 27.78 | 29.86 | 34.38 | 37.15 | 33.85 | 36.28 | **40.45** | \+6.07 | \+3.40 |
>
> We can observe that AIME exemplars help achieve SOTA scores on MathOdyssey, JEE-Bench and OmniMath.
>
> [1] Arora et al., "Have LLMs Advanced Enough? A Challenging Problem Solving Benchmark For Large Language Models"
> [2] Gao et al., "Omni-MATH: A Universal Olympiad Level Mathematic Benchmark For Large Language Models"

---

> > ### Author Response · Authors · 2024-11-26
> > **Author Response to Official Review by Reviewer f1zY (3/N)**
> >
> > > [Weakness] Another limitation is that the method is highly dependent on the coding capabilities of the underlying language model. While the authors acknowledge the correlation between the model's coding performance and the accuracy of the SBSC approach, the exact nature of this correlation is unclear.
> >
> >
> > ### Answer ###
> >
> > We welcome reviewer’s question on dependence on the coding abilities of the underlying LLMs. First, we explicitly demonstrated this relationship in Figure 6 of our paper, which shows a strong positive correlation between base LLM's LiveCodeBench [1] code-generation scores and their SBSC performance for a subset of both AMC and AIME datasets. Second, we would like to deep dive on why we selected LiveCodeBench as the benchmark. Specifically we want to talk about the following two points:
> >
> > **Skills Requirements for SBSC:** The SBSC framework primarily requires two fundamental capabilities in the base LLM to generate responses in the desired codified sequential stepwise format:
> >
> > -  Mathematical/logical reasoning: Understanding math problems, formulating solution strategies in a stepwise manner and identifying computational dependencies.
> > -  Code generation: Ability to generate python code snippets based on given context including earlier code steps and sub-task definition.
> >
> > The capabilities required to enable this necessarily don’t require utilizing all the traditional software engineering skills/practices and align more closely with generating reasoning dense programs/codes for problem-solving.
> >
> >
> > **SBSC Correlation with LLM Code Generation ability:** As discussed above, the code generation ability of the language model is a key factor that may directly influence the generation of suitable SBSC-format responses, and therefore the overall performance of the framework.
> >
> > Two of the most popular benchmarks for evaluating programming abilities of LLMs are LiveCodeBench [1] and SWEBench. Succeeding on SWEBench [2] requires LLMs to do repository level coding where following software engineering practices are imperative. LiveCodeBench includes competitive programming contest problems and is more focused on evaluating the ability to generate code for problem-solving. Given the premise of SBSC, the latter is a more suitable metric for checking capability of LLM in leveraging SBSC effectively.
> >
> > To gain a deeper insight into this, as shown in Section 5.5, we make a comparison involving LLMs with varying level of code generation abilities. Figure 6 clearly shows that the SBSC scores have a strong, positive correlation with the code generation abilities of the base LLM models for all cases that were evaluated on a subset of AIME and AMC data. The code-generation scores were taken from LiveCodeBench benchmark.
> > [1] Jain et al., “LiveCodeBench: Holistic and Contamination Free Evaluation of Large Language Models for Code”, arXiv:2403.07974.
> > [2] Gimenez et al., “SWE-bench: Can Language Models Resolve Real-World GitHub Issues?”, 	arXiv:2310.06770

---

> ### Author Response · Authors · 2024-11-26
> **Author Response to Official Review by Reviewer f1zY (4/N)**
>
> > [Weakness] In addition, the method relies heavily on SymPy which might further limit its flexibility and adaptability.
>
> ### Answer ###
>
> **Consistency in computational library/function usage among POT, PAL, TIR-ToRA and SBSC:** SBSC is an advanced program-generation based reasoning, improving over TIR-ToRA and PAL/POT. Given POT/PAL and TIR-ToRA already leverage these computational libraries and functions like Sympy, SBSC also leverages them as required for fair comparison. POT mentions:  “We use Python 3.8 with the SymPy library4 to execute the generated program.”
> Also, the prompt used by TIR-ToRA explicitly mentions the use of sympy as follows: “Utilize the pi symbol and `Rational`` from Sympy for π  and fractions, and simplify all fractions and square roots without converting them to decimal values.”
>
> **Study on Sympy usage & efficiency usage between SBSC and TIR-ToRA:**
> We conduct a study to understand the usage of sympy library by SBSC and TIR-ToRA for last 10 years of AIME and AMC questions, shown in the table below. Specifically, we present the number of questions in which these methods utilized that library, and we present the accuracy for those set of questions. The number of questions in which both the methods utilized Sympy are similar. For both AIME and AMC problems from the last 10 years, SBSC based solutions have used Sympy only on 7 more questions in both test-sets. But when we look at the accuracy, we can observe that SBSC had 10%+ absolute accuracy (among the filtered questions) lead in both the dataset.
> | Context   | AIME (#Qs out of 300) | AIME (Accuracy %) | AMC (#Qs out of 435) | AMC (Accuracy %) |
> |-----------|------------|-------------------|-----------|------------------|
> | TIR-ToRA  | 200        | 15.50%           | 299       | 36.79%          |
> | SBSC      | 207        | 30.43%           | 306       | 46.73%          |
>
> Also, we further filtered the common questions, for both the test-sets, where both TIR-ToRA and SBSC made use of Sympy. We present this study in the table just below. We observed that 80+% questions are in common for both the methods across both the test-sets. And when we observe the accuracy of the methods over these common questions, SBSC makes ~10% absolute improvement across both the test-sets.
>
> | Dataset | #Common Qs with Sympy Usage  | TIR-ToRA (Accuracy %) | SBSC (Accuracy %) |
> |---------|------------------------------|--------------|----------|
> | AIME    | 163                          | 15.95        | **28.22**    |
> | AMC     | 252                          | 38.49        | **47.62**    |
>
>
>
> We believe the multi-turn nature for dynamic generation and solving of a sub-task unlocks better performance compared to previous SOTA methods.
>
>
> > [Weakness] The approach demands a relatively long context length, which could be a constraint in certain settings. It would be helpful if the authors specified a rough minimum context length required for the given shots and the math problem. This would help get better insights into the method's scalability.
>
> ### Answer ###
>
> We thank the reviewer us suggesting us to discuss around the context length via analysing token length for exemplars, system prompt and, in case of SBSC, understanding the average input token length in the last turn. In the table below, we present the token count for exemplars and system prompt across all the methods. We would like to highlight that due to prompt caching mode available in LLMs, the entire prompt is charged only once.
>
>
>
> | Dataset | Method         | COT (tokens)  | PAL (tokens) | TIR-ToRA (tokens) | SBSC (tokens) |
> |---------|----------------|------|------|----------|-------|
> | AIME    | system_prompt  | 11   | 42   | 123      | 208   |
> |         | exemplars      | 2347 | 1049 | 1588     | 4012  |
> | AMC     | system_prompt  | 11   | 42   | 123      | 208   |
> |         | exemplars      | 1557 | 1013 | 1554     | 3486  |
>
> In case of SBSC, the last turn requires the largest context length as last turn(n) input consists of = system prompt + exemplars + question + output trajectory from 1 to n-1. We present the study in the table below.
>
>
> | Dataset | System Prompt + Exemplars (tokens) | Trajectory 1 to n-1 (tokens) | Total (tokens) |
> |---------|------------------------------------|------------------------------|----------------|
> | AIME    | 4220                              | 899.2                        | 5119.2        |
> | AMC     | 3694                              | 835.85                       | 4529.85       |
>
> We do a detail analysis on the cost per question in the discussion below.

---

> ### Author Response · Authors · 2024-11-26
> **Author Response to Official Review by Reviewer f1zY (5/N)**
>
> > [Questions] What is the ratio of the tokens generated by the SBSC method compared to others such as TIR-ToRA?
>
> > [Questions] In table 1, why is there no number for maj@7 for SBSC?
>
> ### Answer ###
> We would like to answer these via the following points
>
> First, we don’t report maj@7  for SBSC in Table 1 as we wanted to mainly focus on its greedy decoding results and its impact. We would like to highlight that in section 3.2 “Baselines & Configurations” under “Experiment” section, we have addressed why we benchmark SBSC’s greedy decoding results against self-consistency (SC) decoding results (majority@7) of COT, PAL & TIR-TORA. We do this primarily for two reasons: First, SBSC takes multiple turns before arriving at the final answer (on average 6-7 turns per problem , Table 3 in Appendix A.1) and Secondly, to benchmark against the reliance of the current existing prompting strategies on majority voting for boosting accuracy. In Table 1, for Claude 3.5 Sonnet when comparing SBSC's greedy decoding against self-consistency decoding (majority@7) of the previous methods we observe significant absolute increase of 6.2% on AMC, 6.7% on AIME, and 7.4% on MathOdyssey. That’s why we don’t report maj@7 to mainly focus on greedy decoding results of SBSC. Although, we do calculate majority@7 of SBSC for subset of AIME & AMC and display the results in Figure 7. We observe a healthy improvement across both test-sets.
>
> Second, we also have done a study with Claude 3.5 Sonnet to compare the average cost per question. We have taken average over 50 AIME+AMC questions. We leveraged the prompt-caching of repeated input tokens while doing this project. Specifically, we cache the system prompt and exemplars for all the methods. The results are presented in the table below. When comparing average cost per question in caching mode, we observe that SBSC is 3X of TIR-ToRA, 4X of COT and 7X of PAL. When comparing average cost per question even in non-caching mode, we observe that SBSC is 6.2X of TIR-ToRA, 7X of COT and 14.3x of PAL. Hence, Table 1 already covers fair comparison part as well even if we take average cost per question under consideration. In Table 1, we already compared 1 chain of SBSC with 7 chains of other methods. Just as experimental detail, we would like to mention we leveraged prompt caching for our reported experiments with Claude 3.5 Sonnet.
>
>
>
> Average Cost per Question(In USD)
>
> | With prompt caching                    | COT      | PAL      | TIR-ToRA  | SBSC  |
> |----------------------------|----------|----------|-----------|-------------------------------|
> | avg_input_tokens           | 110.75   | 107.75   | 479.25    | 4788.9                        |
> | avg_output_tokens          | 543.95   | 353.2    | 685.15    | 1070                          |
> | avg_cache_tokens           | 2754     | 1255     | 2692.2    | 23610.6                       |
> | avg_cost_per_question      | 0.0093   | 0.0060   | 0.0125    | 0.0375                        |
> | Cost ratio (SBSC/Methods)  | 4        | 6.25     | 3         | 1                             |
>
> | Without prompt caching   | COT      | PAL      | TIR-ToRA  | SBSC |
> |----------------------------|----------|----------|-----------|-------------------------------|
> | avg_input_tokens           | 2858.75  | 1356.75  | 3565.45   | 32846.7                       |
> | avg_output_tokens          | 536.05   | 267.35   | 543.7     | 1161.5                        |
> | avg_cost_per_question      | 0.0166   | 0.0081   | 0.0189    | 0.1160                        |
> | Cost ratio (SBSC/Methods)  | 7        | 14.3     | 6.2       | 1                             |

---

> > ### Author Response · Authors · 2024-11-27
> > **Author Response to Official Review by Reviewer f1zY (6/N)**
> >
> > > [Questions] In figure 5, are the methods compute/token matched?
> > ### Answer ###
> > We appreciate reviewer’s keen observation on the token equivalence for the study presented in figure 5. We compared greedy decoding of TIR-ToRA against greedy decoding of SBSC in Figure 5.
> >
> > We present a topic wise analysis for comparing greedy decoding of SBSC against self-consistency decoding of TIR-ToRA & COT. As mentioned earlier, SBSC is 3x costlier than TIR-ToRA in prompt caching mode and 6x in non-prompt caching mode. So for this study we use majority@7 for TIR-ToRA and COT which is more than fair comparison.
> >
> > We can clearly observe in the table below that SBSC is better across all the topics.
> >
> > | Dataset          | Topic          | COT   | COT ma@7 | TIR-ToRA | TIR-ToRA maj@7 | SBSC  |
> > |------------------|----------------|-------|----------|----------|----------------|-------|
> > | AIME             | Algebra        | 10.53 | 14.47    | 23.7     | 26.32          | **44.7**  |
> > |                  | Arithmetic     | 40.00 | 40.00    | **80.0**     | **80.00**          | **80.0**  |
> > |                  | Combinatorics  | 10.47 | 13.95    | 31.4     | 39.53          | **46.5**  |
> > |                  | Geometry       | 4.55  | 5.45     | 1.8      | 4.55           | **7.3**   |
> > |                  | Number Theory  | 13.21 | 18.87    | 54.7     | 58.49          | **62.3**  |
> > | AMC              | Algebra        | 33.53 | 44.53    | 47.3     | 49.64          | **50.9**  |
> > |                  | Arithmetic     | 64.10 | 67.35    | 61.5     | 69.39          | **74.4**  |
> > |                  | Combinatorics  | 23.91 | 26.00    | 34.8     | 42.39          | **55.4**  |
> > |                  | Geometry       | 19.38 | 24.06    | 17.8     | 27.82          | **31.0**  |
> > |                  | Number Theory  | 41.67 | 44.64    | 52.1     | 55.36          | **62.5**  |
> > | MATH ODYSSEY     | Algebra        | 12.20 | 14.63    | 28.0     | 34.15          | **41.5**  |
> > |                  | Combinatorics  | 13.51 | 18.92    | 32.4     | 35.14          | **43.2**  |
> > |                  | Geometry       | 8.00  | 12.00    | 16.0     | 20.00          | **28.0**  |
> > |                  | Number Theory  | 0.00  | 25.00    | 25.0     | **50.00**          | **50.0**  |
> > | Olympiad Bench   | Algebra        | 50.52 | 52.67    | 55.7     | 57.33          | **61.9**  |
> > |                  | Combinatorics  | 32.97 | 37.72    | 41.8     | 43.26          | **46.2**  |
> > |                  | Geometry       | 25.86 | 27.60    | 27.6     | 29.14          | **31.9**  |
> > |                  | Number Theory  | 38.24 | 46.27    | 62.7     | 65.58          | **67.6**  |
> >
> > We have also updated the Figure 5 in the revised version of the paper.
> >
> >
> > > [Questions]  How does the method perform when used with open LLMs such as LLaMA?
> > ### Answer ###
> > We thank the reviewer for the suggestion to showcase the performance of SBSC using open source LLMs.
> >
> > To this end, we performed benchmarking with “DeepSeekCoder 2.5” which is an open source model. Note: given the resource constraint and that most of the frontier open source LLMs such as DeepSeekCoder 2.5 require multiple gpus and days to host and run the experiments respectively, we leveraged this open source model via api provided by deepseek which was highly cost-effective. We present the results on 2 test-sets: AIME and AMC.
> >
> > We can clearly observe that SBSC outperforms both greedy decoding and self-consistency decoding results of TIR-ToRA, PAL & COT on both last 11 years AIME & AMC. The delta column represents the percentage improvement over the next best method.
> >
> > | DeepSeekCoder V 2.5 | COT(greedy) | COT(maj @7) | PAL(greedy) | PAL(maj @7) | TIR-ToRA(greedy) | TIR-ToRA(maj @7) | SBSC(greedy) | Δ% (greedy) | Δ% (maj@7) |
> > |-------|-----|------------|-----|------------|-----------|-----------------|------|--------------|-------------|
> > | AIME | 8.48 | 10.30 | 18.79 | 20.00 | 19.70 | 20.61 | **23.64** | +3.94 | +3.03 |
> > | AMC | 24.84 | 26.53 | 29.47 | 30.53 | 31.16 | 32.42 | **34.53** | +3.37 | +2.11 |
> >
> > > [Questions] Figure 7 seems to have the wrong caption. (Edit in paper along with other typos)
> >
> > We are grateful to the reviewer for pointing out this drafting error. We have fixed this caption for Figure 7 in the revised version.
> >
> > > [Questions]  What types of errors are reported in Section 5.3? Are they primarily syntactic or semantic in nature?
> >
> > #### Answer ####
> > Any step in the overall solution that yields an error message as code interpreter output has been classified as an error step (ValueError, KeyError, TimeOut Error, etc.) We don’t individually verify the mathematical output of each step and hence, such errors are not counted if the code runs properly.  It is best to label the errors encountered in our method as belonging to runtime errors.

---

> ### Author Response · Authors · 2024-11-27
> **Author Response to Official Review by Reviewer f1zY (7/N)**
>
> > [Weakness] Lastly, the discussion of the related works section seems somewhat rushed. In my opinion, the first sentence feels awkward and reads more like filler rather than contributing meaningfully to the content. A more thorough comparison with existing literature on program or tool-aided generation (the section following POT and PAL) could be extended, as it is the most relevant part of the discussion. This would provide deeper insights into the novelty and strengths of the proposed method.
>
> ### Answer ###
> We are thankful to the reviewer for the constructive feedback. We want to let the reviewer know that we have added a comparison part in the revised paper and also reframed our first sentence.
>
> > [Questions] What is the advantage of this step-by-step approach compared to other similar methods, particularly in cases where the model is unable to revisit and revise a problematic step (e.g., Step 1) once it has progressed to later steps?
>
> ### PART 1 ANSWER ###
> We appreciate the reviewer's insightful question regarding the advantages of our Step-by-Step Coding (SBSC) approach, particularly in cases where the model cannot revisit and revise a problematic early step once it has progressed to later steps.
> SBSC framework offers multiple key advantages over other methods.
>
> **TL;DR version of advantages:** SBSC is granular in nature where the problem is solved by solving sequence of sub-tasks. SBSC embodies exploration by discovering new sub-tasks basis the previous sub-tasks. These sub-tasks are solved via program generation to generate intermediate outputs. One of the major advantages of SBSC framework is that it accommodates various types of intermediate reasoning steps/sub-tasks, not just those producing immediate numerical or computational outputs. The “intermediate output printing protocol at each step” of SBSC framework serves multiple purposes such as: verification of step completion, as input for subsequent reasoning steps, execution error information for recovery and documentation of solution progress. **Due to the step-by-step nature of SBSC, it is better able to localise errors at the step level instantly and fix it at the site of occurrence. As a result, SBSC has superior step-level self-debugging abilities which prevents propagation of errors reducing the need to revisit older steps and minimizes the risk of going into restarting loops when the model restarts from step-1.** This can be seen in Figure 1 example. Hence due to these major advantages, SBSC achieves significant improvement over TIR-ToRA across 4 different benchmarks as can be seen in Table 1.
>
> Note: we are only referring to steps which return an erroneous output (invalid outputs, runtime error) as error steps for the study ahead.
>
> **Analysing occurrences of step-level error correction behaviour in SBSC’s trajectory:** One of the main novelties involved in SBSC is the stepwise solution paradigm for program generation based math reasoning. Further, the model is expected to correct erroneous intermediate steps immediately after it’s identification; minimising the need of revisiting or restarting the problem from scratch, unlike TIR-ToRA. This would help with:
> - Since error can be better localised at the intermediate step level itself, we can prevent propagation of errors and prevent the downstream steps from becoming ineffective.
> - Since some of the SBSC solutions can have as many as 6-7 steps( as shown A.1 in appendix), minimising revisiting previous steps/restarting from step 1 also helps in keeping the number of API calls/cost made per problem low.
>
> Note: Revisiting previous steps or restarting from step 1 is not completely avoidable and at times important. As we will look into the tables below, we will observe that SBSC does do that in few occasions when there is no other way. Also, in case of reasoning errors such as generating incorrect-subtask that doesn’t lead to any runtime or constraint violation type-errors could go ahead unidentified. Revisiting can be further explored with back-tracking mechanisms such as in tree-search methods for LLMs as done in Language Agent Tree Search [1], Tree search for language model agents [2].
>
> We present a study where we analyse in detail the various scenarios that may occur in case of an SBSC response. We consider three categories:
> Class 1: Response with no step-level error correction and no revisiting previous steps.
> Class 2: Response where the model does step-level error correction by fixing the erroneous step and doesn’t revisit the previous steps in any case.
> Class 3: Response where at least once: there is an error at a step and the model revisits the previous steps or restarts from step 1.
> For TIR-ToRA, there is always a single-step/code solutions and re-attempts, so only Classes 1 and 3(essentially revisiting step 1 again) are valid.
>
> ANSWER CONTINUED .....

---

> > ### Author Response · Authors · 2024-11-27
> > **Author Response to Official Review by Reviewer f1zY (8/N)**
> >
> > > [Questions] What is the advantage of this step-by-step approach compared to other similar methods, particularly in cases where the model is unable to revisit and revise a problematic step (e.g., Step 1) once it has progressed to later steps?
> >
> > ### PART 2 ANSWER ###
> > **Analysing occurrences of step-level error correction behaviour in SBSC’ trajectory: Continued...**
> >
> > We can see in the table below, majority of the problem solutions fall in the first two classes itself for SBSC, while TIR-ToRA has a fair proportion of Class 3 scenarios as it tries to reattempt. This can be attributed to the fact that SBSC first attempts to fix errors at step level and therefore minimizes visiting previous steps and minimizes restarting from step 1 unless it identifies a different approach altogether for solving the problem as the possible fix.
> >
> > | Dataset         | Class 1 Occurrences (TIR-ToRA) | Class 1 Occurrences (SBSC) | Class 2 Occurrences (TIR-ToRA) | Class 2 Occurrences (SBSC) | Class 3 Occurrences (TIR-ToRA) | Class 3 Occurrences (SBSC) |
> > |-----------------|--------------------------------|-----------------------------|--------------------------------|-----------------------------|--------------------------------|-----------------------------|
> > | AMC (475 Qs)    | 388                            | 321                         | -                              | 122                         | 87                             | 32                          |
> > | AIME (330 Qs)   | 246                            | 172                         | -                              | 104                         | 84                             | 54                          |
> > | Odyssey (148 Qs) | 108                            | 84                          | -                              | 59                          | 40                             | 7                           |
> >
> > Further, in the table below, we present a comparison of accuracies of both SBSC and TIR-ToRA under 2 distinct scenarios:
> > •	When the solution doesn’t face any errors in any of its code blocks (or the single block in case of TIR-ToRA) and reaches an answer, i.e. no rectification of any sort is needed
> > •	When either the entire code (TIR-ToRA) or any intermediate code (SBSC) fails to return a valid output, i.e. some rectification is needed in the overall or intermediate level solution
> >
> > | Dataset  | % Accuracy Class 2 or 3 (TIR-ToRA) | % Accuracy Class 2 or 3 (SBSC) | % Accuracy Class 1 (TIR-ToRA) | % Accuracy Class 1 (SBSC) |
> > |----------|------------------------------------|---------------------------------|-------------------------------|----------------------------|
> > | AMC      | 25.29                             | 32.47                          | 42.27                        | 58.57                     |
> > | AIME     | 14.29                             | 24.68                          | 26.83                        | 45.35                     |
> > | Odyssey  | 17.50                             | 34.38                          | 30.56                        | 44.05                     |
> >
> > We show that across all datasets, the SBSC paradigm proves to be more effective than TIR-ToRA. The step-by-step methodology has a better accuracy at solving problems with no errors and hence there is no step level errors to be rectified. Also, in scenarios when that is not the case, our localised step rectification works much better.
> >
> > **Detailed discussion on overall advantages:** Previous program generation-based methods have followed a static approach and hence were better suited for simpler problems. These methods typically attempt to solve the problem in a single, large code block or with minimal iterations(in case of TIR-ToRA). But all the previous methods lack producing/exploring any intermediate steps or arriving at the solution in step-by-step progression.
> >
> > SBSC decomposes complex problems into a sequence of manageable sub-tasks, each addressed individually and each building on top of previous ones. This fine-grained approach allows the model to concentrate on one aspect of the problem at a time. Turn by turn SBSC helps discover intermediate sub-tasks and focuses on each one of them.
> >
> > Given the multi-turn nature, SBSC produces intermediate outputs at each step. One of the major advantages of SBSC framework is that it accommodates various types of intermediate reasoning steps/sub-tasks, not just those producing immediate numerical or computational outputs. Here's a detailed breakdown on the kind of steps:
> > - *Computational Steps:* Direct calculations producing numerical results, Data transformations with clear outputs
> > - *Preparatory Steps:* Problem decomposition decisions, Variable and constant declarations, Function definitions or Data structure initialization for later use, Algorithm setup and configuration
> > - *Verification/Error Correction Steps:* Verification of intermediate or results, Rectifying an erroneous intermediate step
> >
> > CONTINUED..

---

> ### Author Response · Authors · 2024-11-27
> **Author Response to Official Review by Reviewer f1zY (9/N)**
>
> > [Questions] What is the advantage of this step-by-step approach compared to other similar methods, particularly in cases where the model is unable to revisit and revise a problematic step (e.g., Step 1) once it has progressed to later steps?
>
> ### PART 3 ANSWER ###
> **Detailed discussion on overall advantages: Continued...**
>
> The “intermediate output printing protocol at each step” of SBSC framework serves multiple purposes such as: verification of step completion, as input for subsequent reasoning steps, error information for recovery and documentation of solution progress. Its robustness is particularly evident in complex problem-solving scenarios where traditional monolithic approaches might fail to capture intermediate state changes or error conditions effectively. This systematic approach to output management ensures that the LLM maintains a clear understanding of the solution's progress, regardless of the step's nature, while providing necessary feedback for both the model and the user about the solution's development.
>
> **What SBSC could potentially unlock due to its granular nature?** As mentioned in “Future Work”, we believe that due to this very nature of structure output protocol for intermediate states/answers generated, we believe SBSC reasoning format could be highly useful for guided decoding strategies such as in Outcome-Supervised Value Model [3], AlphaMATH [4], Q* framework [5]. We plan to cover the down-stream benefits of SBSC in the next iteration of our paper.
>
>
> [1] Zhou et al., “Language Agent Tree Search Unifies Reasoning Acting and Planning in Language Models”    arXiv:2310.04406.
> [2] Koh et al., “Tree Search for Language Model Agents”        arXiv:2407.01476.
> [3] Yu et al., “Outcome-supervised value models for planning in mathematical reasoning”    arXiv:2311.09724.
> [4] Chen et al., “Alphamath almost zero: process supervision without process”    arXiv:2405.03553.
> [5] Wang et al., “Q*: Improving multi-step reasoning for llms with deliberative planning”     arXiv:2406.14283.

---

> > ### Author Response · Authors · 2024-11-30
> > **Gentle Reminder**
> >
> > Dear reviewer f1zY,
> >
> > As the discussion period is approaching the closing date, we want to check in on whether our responses have addressed your questions. We would be happy to discuss further.
> >
> > Authors

---

> ### Author Response · Authors · 2024-12-02
> **Summary of Rebuttals & Revisions**
>
> Dear Reviewer f1zY,
>
> As the discussion period is coming to an end in less than 24 hours, we wanted to check if you were able to go through our rebuttal answers to your queries. We also attempt to summarize the above replies:
>
> * **Clarified the exemplar curation and crafting process** for SBSC and expanded our pool of exemplars to **provide additional sensitivity analysis results** on AIME and Math-Odyssey. (Replies 1/N and 2/N, *added Appendix A.5 in paper*)
> * Provided additional benchmarking results **using the same AIME exemplars two new datasets: OMNI-Math and JEE-BENCH** to demonstrate the generalization ability of our method irrespective of the dataset and refute claims of overfitting. (Reply 2/N, *added Appendix A.2 in paper*)
> * Elaborately **explained the strong, positive correlation** between the underlying language model's code generation capabilities and corresponding SBC performance. (Reply 3/N)
> * Conducted a study to understand the **effect of using/not using Python libraries like SymPy** and compared it with another program-generation based method TIR-ToRA. (Reply 4/N, *added Appendix A.7 in paper*)
> * We provide **details regarding the required context length** for SBSC and its counterparts. (Reply 4/N, added *Appendix A.4 in paper*)
> * Supplemented our comparative **results under equivalent token costs in Section 4** by providing details on **average cost per question** using different methods, under both prompt caching and non-caching settings. (Replies 5/N and 6/N, *added Appendix A.4 in paper*)
> * Added thorough comparative results using the **open-source LLM DeepSeekCoder-2.5** to demonstrate the generalization ability of our method. (Reply 6/N, *added Appendix A.3 in paper*)
> * **Revised the "Related Work" section** of our paper to include a more detailed comparison. (*updated section 6 in paper*)
> * Enhanced **explanation of the overall advantages of our step-by-step workflow** compared to existing static program generation-based methods. (Replies 7/N and 9/N)
> * Added **detailed analysis of various possible scenarios** that may or may not involve errors at intermediate steps or may require revisiting the problem to correctly solve it. (Reply 8/N)
> * Lastly, we have rectified the typographical errors and presentation issues in our paper.
>
> If all concerns have been addressed to your satisfaction, we would be truly grateful if you could consider upgrading your score.
>
> Authors

---

> > ### Comment · Reviewer_f1zY · 2024-12-03
> >
> > Thank you for your detailed responses.
> >
> > I still believe that exemplar curation and crafting in this case is a manual and intensive process.
> >
> > My concern about heavily relying on SymPy still stands, especially given that, as you pointed out, 207 out of 300 questions depend on it.
> >
> > I appreciate your efforts. I am raising the presentation score to 4.

---

> > > ### Author Response · Authors · 2024-12-03
> > > **Clarifications regarding Exemplar Crafting and SymPy usage**
> > >
> > > We thank the reviewer for the reply and we truly appreciate the increase in presentation score. To the point of manual prompt curation and sympy usage we would like to re-highlight the following points:
> > >
> > >
> > > > I still believe that exemplar curation and crafting in this case is a manual and intensive process.
> > >
> > > • **Crafting SBSC exemplars require little to no manual effort**: We first used 0-shot prompting (only system prompt) to generate responses for questions in the SBSC format. Thus, our initial pool of SBSC exemplars was created without using any manual effort at all. For enhancing the exemplars further where necessary, we made use of GPT-4o to add helpful comments within the code.
> > >
> > > • **Previous methods have probably leveraged manual efforts**: PAL authors have mentioned the following in their paper in the section “Crafting prompts for PAL”: "In all cases, we augmented the free-form text prompts into PAL-styled prompts, leveraging programming constructs such as for loops and dictionaries when needed. Generally, writing PAL prompts is easy and quick. We also ensure that variable names in the prompt meaningfully reflect their roles."
> > >
> > > • Moreover, **authors of ToRA mention this**:
> > > “Prompt Curation: We compose instructions along with diverse few-shot examples, utilizing an interleaved format as depicted in Fig 2 (c). These examples showcase interactive tool usage trajectories, incorporating descriptive variable names and combined program outputs.”**.
> > >
> > >
> > >
> > > > My concern about heavily relying on SymPy still stands, especially given that, as you pointed out, 207 out of 300 questions depend on it.
> > >
> > > • **Consistent Use of SymPy library across tool-integrated reasoning methods**: The usage of SymPy library has shown to be useful for program-based math problem solving in existing literature. TIR-ToRA, which is the current SOTA in math reasoning on challenging benchmarks, explicitly instruct the model to make use of SymPy library in their system prompt [https://github.com/microsoft/ToRA/blob/main/src/prompts/tora/math.md]:
> > > *“Utilize the pi symbol and `Rational`` from Sympy for π and fractions, and simplify all fractions and square roots without converting them to decimal values.”*
> > >
> > > • In fact, the **TIR-ToRA few-shot exemplars clearly demonstrate the importing of SymPy library** at the beginning of the program and using it to solve the problem.
> > >
> > > • **Similar Proportion of SymPy usage by SBSC**: As we have already shown in the Table in reply 4/N and in Appendix A.7 of our paper, SymPy is used in SBSC only for solving 1-2% more questions (207 vs 200 in AIME, 306 vs 299 in AMC), compared to TIR-ToRA, while attaining a far higher accuracy. While SBSC already improves usage of sympy libraries, effective usage of such/more computational libraries can be one of the important future focus areas for program generation based math problem solving.

---

### Author Response · Authors · 2024-12-04
**Summary of Discussion Period Part 1**

We sincerely thank the reviewers for their valuable feedback. The discussion phase has been very productive, and we were able to greatly enhance the clarity of our evaluation results and deliver our findings in a more convincing manner through additional experiments & empirical observations as well as clarifications on some parts of our methodology.

We thank them for appreciating the novelties and strengths of our proposed methods, which are summarized as follows:

- SBSC leverages inference time compute to **break down complex problems into manageable sub-tasks and this structure allows the model to build on the results of previous steps**, using the output/feedback to condition its generation of the next step and sub-task. (*f1zY*)
- The step-by-step process with intermediate feedback provides the model with a chance to **detect and correct errors or rewrite parts of the code.** This improves the overall performance of the model. (*f1zY*)
- The concept of step-by-step, multi-turn coding is both intuitive and innovative. Unlike previous approaches that focus mainly on validation after the final generation or self-debugging, **SBSC’s use of intermediate results from an executor provides more actionable and informative feedback.** This method effectively **combines task decomposition from CoT with the program-assisted capabilities of PAL, offering a promising approach**. (*uQkW*)
- The empirical results demonstrate that **SBSC achieves superior performance over state-of-the-art methods**, suggesting that its iterative nature allows for more accurate problem-solving in the context of advanced math problems.  (*Ew8i, BqFn, uQkW*)
- The **experiments cover a variety of math competitions**, indicating that SBSC is not tailored to a specific type of problem but is instead **broadly applicable to Olympiad-level mathematics**. (*BqFn*)
- The **ablation and analysis section offers a deep analysis of the results, providing insights into why SBSC performs better** than existing methods and under what circumstances it might be most beneficial. (*BqFn*)

---

> ### Author Response · Authors · 2024-12-04
> **Summary of Discussion Period Part 2**
>
> Further, we take this opportunity to summarize the rebuttals and revisions made in the paper by us in response to the queries and concerns raised by the reviewers:
>
> - Provided additional benchmarking results using the **same AIME exemplars on two new datasets: OMNI-Math and JEE-BENCH** to demonstrate the generalization ability of our method irrespective of the dataset and refute claims of overfitting.  (added Appendix A.2 in paper)
> -  Added thorough comparative **results using the open-source LLM DeepSeekCoder-2.5** to demonstrate the generalization ability of our method. (added Appendix A.3 in paper)
> - We provide details regarding the **required context length for SBSC** and its counterparts as well as supplemented our **comparative results under equivalent token costs** in Section 4 of our paper by providing details on average cost per question using different methods, under both prompt caching and non-caching settings. (added Appendix A.4 in paper)
> - **Clarified the exemplar curation and crafting process** for SBSC and expanded our pool of exemplars to provide **additional sensitivity analysis results** on AIME and Math-Odyssey. (added Appendix A.5 in paper)
> - Conducted a study to understand the **effect of using/not using Python libraries like SymPy** and compared it with another program-generation based method TIR-ToRA. (added Appendix A.7 in paper)
> - **Added COT scores to the table comparing the topic-wise results** to demonstrate that SBSC outperforms not just program-generation based methods like TIR-ToRA but also natural language-based ones across all topics on all datasets.  (revised Section 5.2 in paper)
> - **Clarified the flexible and dynamic nature of sub-task generation** in SBSC that leverages the explorations of previous sub-tasks and enables granular step-level correction if required.
> - Provided a **detailed explanation of various types of intermediate reasoning steps/sub-tasks** that may be Computational/Preparatory/Error-Correction or Verification based, all of which provide important feedback to the model and aids in subsequent solution development. Also, presented a detailed **analysis of various possible scenarios that may or may not involve errors at intermediate steps or may require revisiting the problem** to correctly solve it
> - **Enhanced explanation of the overall advantages of our step-by-step workflow** compared to existing static program generation-based methods.
> - Conducted a study to understand the potential of SBSC in solving challenging problems by comparing our **method’s upper bound using pass @k metric** with SOTA OpenAI o1 models on last 3 years’ AIME problems.
> - Performed **quantitative and qualitative error analysis over different math topics* to demonstrate that our method is indeed capable of solving most Olympiad-level problems, irrespective of the domain.
> - Added multiple **sample SBSC responses that mimic human-like reasoning**, only the logic is codified instead of being written in a natural language form. [https://anonymous.4open.science/r/SBSC-69D3/example_responses/]
>
> We hope that we were able to satisfactorily address all the queries of the reviwers . It was indeed a rather fruitful and enriching discussion period, which allowed us to provide deeper insights  regarding our paper to the reviewers and readers alike.
>
> Authors

---

### Meta-Review · Area_Chair_p5HF · 2024-12-21

**Metareview:**

This paper presents Step-by-Step Coding (SBSC), a novel framework that enables LLMs to solve complex mathematical problems through sequential program generation. SBSC breaks down problems into sub-tasks and leverages code execution feedback to guide subsequent steps. The key strengths are the strong empirical results showing significant improvements over SOTA methods across multiple math competition datasets, thorough ablation studies, and broad applicability across different math topics. While there are concerns about manual prompt engineering effort and heavy reliance on SymPy, the authors adequately addressed these through additional experiments and analyses. The paper makes a valuable contribution to mathematical reasoning with LLMs by demonstrating the effectiveness of granular, multi-turn program generation. For the camera-ready version, the authors should ensure all figures are vectorized for improved readability.

**Additional Comments On Reviewer Discussion:**

During the rebuttal period, reviewers raised concerns about prompt engineering effort (f1zY), SymPy dependency (f1zY), and whether programming is suitable for advanced math problems (uQkW). The authors provided extensive experiments showing SBSC's effectiveness across additional datasets, comparative analysis with OpenAI's o1 models, and detailed error analysis across math topics. While some reviewers remained concerned about the approach's applicability to very challenging problems, the thorough responses and strong empirical results support acceptance. The discussion helped establish SBSC as a promising direction for advancing program-aided mathematical reasoning.

---

### Decision · Program_Chairs · 2025-01-22

Accept (Poster)